# PET-based tracking of CAR T cells and viral gene transfer using a cell surface reporter that binds to lanthanide complexes

Volker Morath [1,9], Katja Fritschle[1,9], Linda Warmuth[2], Markus Anneser[3], Sarah Dötsch[2], Milica Živanić [1], Luisa Krumwiede [1], Philipp Bösl[1], Tarik Bozoglu [4,5], Stephanie Robu[1], Silvana Libertini [6], Susanne Kossatz[1], Christian Kupatt[4], Markus Schwaiger[1], Katja Steiger [7], Dirk H. Busch [2], Arne Skerra [3] & Wolfgang A. Weber [1,8] ✉

The clinical translation of cell- and gene-based therapies is limited by the lack of non-invasive, quantitative and specific whole-body imaging tools. Here we present a positron emission tomography reporter system based on a membrane-anchored anticalin protein that binds a fluorine-18-labelled lanthanide complex with picomolar affinity via a bio-orthogonal interaction. The reporter was introduced into therapeutic cells, including CAR T cells and adeno-associated virus-transduced cells. In vitro, reporter expression conferred >800-fold higher radioligand binding versus controls. In mice, the radioligand demonstrated rapid renal clearance, showed no off-target accumulation and enabled high-contrast detection of as few as 1,200 CAR T cells in the bone marrow. Longitudinal positron emission tomography imaging over 4 weeks revealed precise tracking of CAR T cell expansion and migration, with signal intensity correlating linearly with flow cytometry data. The system also enabled the quantitative imaging of in vivo gene transfer using an adeno-associated viral vector. This depth-independent whole-body imaging platform offers a powerful tool for monitoring therapeutic cell dynamics and gene delivery in preclinical and potentially clinical settings.

Recent regulatory approvals of cell and gene therapies[1,2] have shown that advanced therapy medicinal products (ATMPs) can be used in clinical practice and have the potential to cure life-threatening diseases. However, the development of cell- and virus-based ATMPs remains hampered by limited knowledge about the biodistribution and persistence of these live drugs[3]. Technologies to non-invasively and quantitatively monitor the distribution of ATMPs in vivo, such as chimeric antigen receptor (CAR) T cells, could greatly improve our understanding of their trafficking, therapeutic efficacy and off-target toxicity[4]. Likewise, adeno-associated virus (AAV)-based gene therapies would benefit from quantifying the location, magnitude and duration of transgene expression.

Direct ex vivo labelling of cells (as with iron oxide particles[5]) is a sensitive approach for studying the distribution of transplanted cells early after injection. Although sensitive, this approach is limited at later timepoints because the label is diluted by each cell division, which

[1]Department of Nuclear Medicine, TUM University Hospital, School of Medicine and Health, Technical University of Munich, Munich, Germany. [2]Institute for Medical Microbiology, Immunology and Hygiene, School of Medicine and Health, Technical University of Munich, Munich, Germany. [3]Lehrstuhl für Biologische Chemie, School of Life Sciences, Technical University of Munich, Freising, Germany. [4]Deutsches Zentrum für Herz-Kreislaufforschung, Munich, Germany. [5]Medizinische Klinik I, Klinikum rechts der Isar, School of Medicine and Health, Technical University of Munich, Munich, Germany. [6]Novartis Biomedical Research, Basel, Switzerland. [7]Comparative Experimental Pathology, School of Medicine and Health, Technical University of Munich, Munich, Germany. [8]Bavarian Cancer Research Center, Munich, Germany. [9]These authors contributed equally: Volker Morath, Katja Fritschle. ✉e-mail: w.weber@tum.de

prevents the assessment of cell proliferation[6]. Furthermore, the label persists even after the cells have died or were phagocytosed[6]. This limits the utility of direct labelling for CAR T cell tracking, where a small fraction of the graft massively expands after activation[6].

As an alternative, cell graft proliferation and viability can be imaged by an indirect labelling strategy. This approach relies on either endogenous biomarkers or reporter genes (synthetic biomarkers) that can be detected using spectrometric methods[7]. Various luciferase enzymes and fluorescent proteins have been used as reporter genes to study transplanted cells in mice by means of optical imaging techniques[8]. However, light is strongly attenuated and scattered within biological tissues, which limits imaging depth and quantification in optical imaging[9], particularly in larger animals or humans[8].

A viable modality for reporter gene imaging should allow whole-body imaging, be highly sensitive and provide a quantitative, depth-independent signal[7]. Furthermore, imaging should be cross-sectional to enable precise anatomical localization. Positron emission tomography (PET) fulfils all of these requirements, thus enabling the spatiotemporal imaging of transgenes on a whole-body scale while being widely available for preclinical and clinical imaging[4]. The sensitivity of current clinical scanners lies in the picomolar range[6] and has been further increased by a factor of 40 with the recent introduction of total-body PET scanners[10].

The ideal reporter gene or protein should be small, bio-orthogonal—that is, not expressed endogenously and not affecting cell function—and non-immunogenic. For PET imaging, a straightforward radiolabelling procedure with a commonly available radioisotope such as fluorine-18 would be desirable. The small-molecule probe should show low non-specific binding to cells and undergo rapid kidney clearance but remain in circulation sufficiently long to ensure quantitative labelling of cells expressing the reporter gene. Notably, despite numerous efforts over three decades[11], no reporter gene system fulfils these requirements in a clinical setting. Herpes simplex thymidine kinase (HSV-tk)[12] has been extensively studied as a reporter gene, but this viral protein is highly immunogenic with a high seroprevalence. Meanwhile, systems like the sodium–iodide symporter (NIS)[13], somatostatin receptor 2 (SSTR2)[14] and (truncated) prostate-specific membrane antigen ((t)PSMA)[15,16] are not immunogenic but are expressed endogenously by various tissues, which hampers their application for the specific whole-body imaging of grafted cells.

To address the unmet medical need to track ATMPs, we developed and characterized a reporter gene and radioligand system that combines high sensitivity with high specificity. Our approach is based on an anticalin[17,18] that is expressed on the cell surface and binds a bio-orthogonal [18]F-labelled probe. Anticalins are engineered binding proteins based on the lipocalin protein family[19], which are endogenous plasma proteins comprising a single polypeptide with a robust β-barrel fold[20]. Here, we have extensively studied this reporter system in vivo and demonstrated its suitability to monitor CAR T cells and AAV-mediated gene transfer in mice.

## Results

### Design of the reporter protein and radioligand

The PET reporter protein has been designed as an artificial cell surface protein exhibiting a highly specific extracellular binding site for a small-molecule ligand (Fig. 1a). The protein construct comprises the lipocalin-2 signal peptide, an engineered lipocalin binding protein (anticalin[19]), a peptide linker containing the V5-tag (serving as a multifunctional molecular handle for staining of ATMPs)[21] and the α-helical human cluster of differentiation (CD)4 transmembrane domain. This fusion protein comprises only 257 amino acid residues (aa) and is encoded in a 774-base-pair (bp) open reading frame (see sequences in the Supplementary Information). Finally, for multimodal detection, an optional fluorescent protein was added to the C-terminal cytoplasmic end, such as mRuby3[22] or miRFP720[23] (Fig. 1b). Two anticalins were

chosen to construct different reporter proteins with two ligand specificities: (1) the anticalin CL31d, which binds CHX-A″-diethylene triamine pentaacetic acid (DTPA)•metal complexes with a dissociation constant ($K_D$) of ~500 pM (refs. 17,18) was used to construct the DTPA reporter (DTPA-R) and (2) the anticalin D6.4(Q77E), which binds colchicine with a $K_D$ of ~20 pM (ref. 24), was used to generate the Colchi reporter (Colchi-R). An in silico prediction (https://iedb.org/, Supplementary Fig. 1, Supplementary Table 1 and Supplementary Discussion 1) revealed that only two peptide fragments of DTPA-R (peptides 'rank 15' and 'rank 17' in Supplementary Fig. 1) were putative ligands of representative human leukocyte antigens. This prediction is in line with results from clinical trials in which so far nine anticalin drug candidates, including some fusion proteins, have been studied without notable signs of immunogenicity (https://clinicaltrials.gov/ and ref. 25). The cognate reporter probe features CHX-A″-DTPA•metal (Fig. 1c) or colchicine (Fig. 1d), respectively, a polyethylene glycol (PEG)₄-linker to facilitate access to the anticalin binding pocket without steric hindrance, optionally hydrophilic groups to improve pharmacokinetics and, finally, a labelling group for efficient incorporation of the [18]F isotope to enable PET detection (Fig. 1c).

### Analysis of reporter protein variants

To compare the two anticalin–ligand pairs, we generated cell lines expressing the respective reporter genes by retroviral transduction of the human T cell line Jurkat, human embryonic kidney cells (HEK293T) and the prostate carcinoma cell line PC3. Initially, cell surface expression of the DTPA-R or Colchi-R reporter proteins on PC3 cells was confirmed by fluorescence microscopy using an anti-V5-tag antibody (Fig. 1e). Membrane localization of mRuby3 in HEK293T cells expressing DTPA-R–mRuby3 indicated efficient transport of the reporter protein through the secretory pathway (Extended Data Fig. 1). Absolute reporter protein numbers per cell were measured by flow cytometry using an anti-V5-tag antibody and molecules of equivalent soluble fluorochrome (MESF) calibration beads. Expression levels on Jurkat cells ranged from ~16,000 for DTPA-R–miRFP720 to around 1 million receptors for DTPA-R (Fig. 1f) if assuming two receptors being bound by one antibody, which might overestimate receptor numbers. Omitting the optional fluorescent protein increased expression, for example, DTPA-R showed 8.3-fold ($P = 0.0008$) higher expression than DTPA-R–mRuby3 (Fig. 1f). Next, we studied the influence of T cell activation on reporter gene expression by comparing phorbol 12-myristate 13-acetate (PMA)–ionomycin-activated with untreated Jurkat cells, showing no significant differences (Fig. 1f and Extended Data Fig. 2a). Furthermore, the type of anticalin influenced the expression levels: a 4.9-fold ($P = 0.0015$) higher expression of DTPA-R was measured compared with Colchi-R (Fig. 1f, Extended Data Figs. 1 and 2a and Supplementary Fig. 2). We also compared the expression level of DTPA-R with a previously described reporter gene that uses the murine single-chain variable fragment (scFv) C825 or its humanized version huC825 (ref. 26), to bind tetraazacyclododecane-tetraacetic acid (DOTA)•metal complexes (Extended Data Fig. 2a). Using the same expression cassette and membrane anchor domain for these scFvs, DTPA-R showed a 5.8-fold (muC825) or 8.9-fold (huC825) higher median V5-AF488 signal after retroviral transduction. The number of genomically inserted expression cassettes per cell was quantified by droplet digital (dd)PCR and showed similar levels for the different constructs (Extended Data Fig. 2b). mRNA levels showed more variation between constructs, and longer mRNAs encoding fluorescent protein domains were expressed at lower levels (Extended Data Fig. 2c). While protein levels of anticalin-based reporters correlated with mRNA levels ($R^2 = 0.98$), scFv-based reporter genes produced high mRNA levels, which did not translate into elevated protein levels, indicating superior protein folding, stability and cell-surface presentation for anticalin-based reporter proteins (Extended Data Fig. 2d).

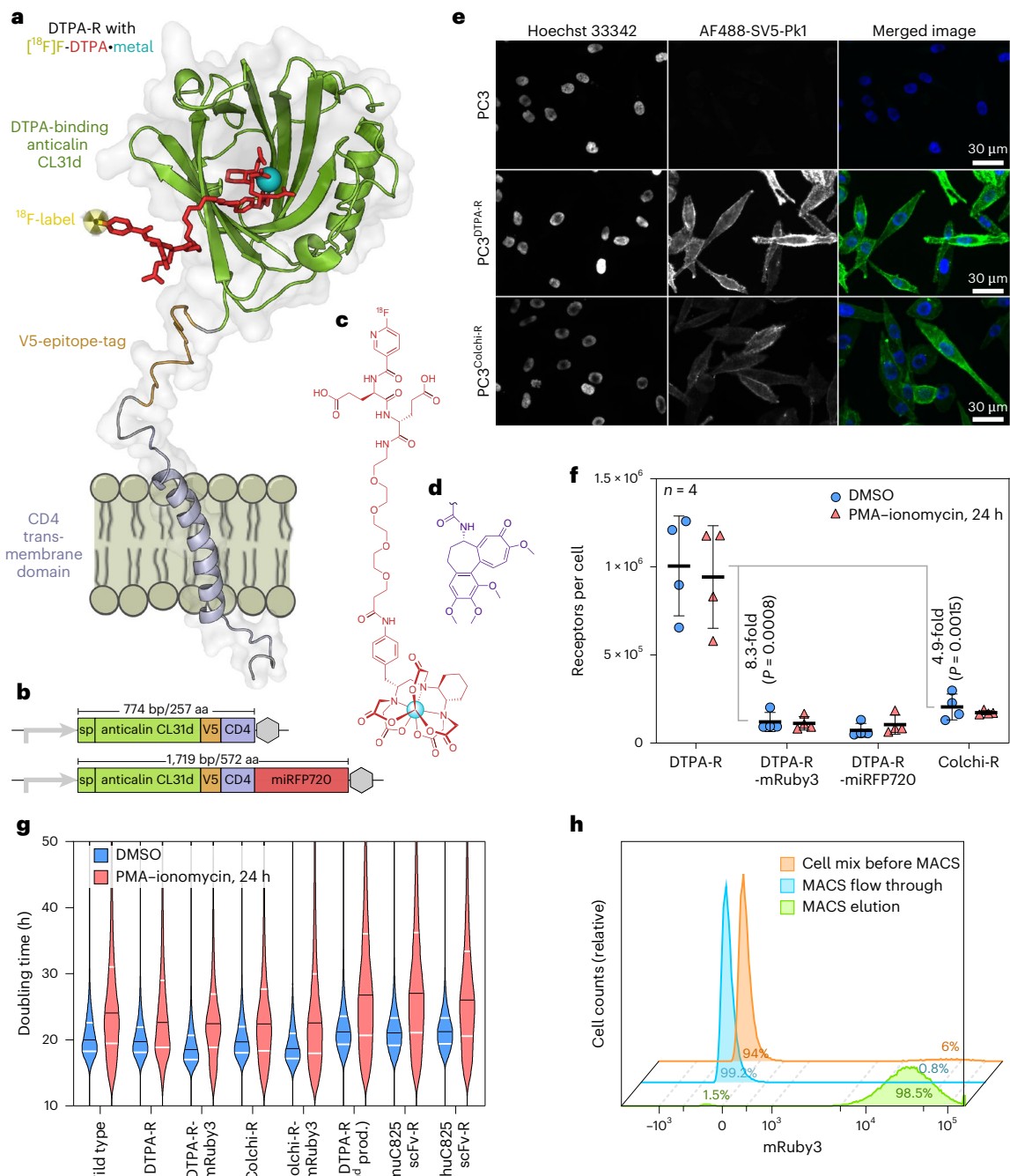

**Fig. 1 | Design and characterization of DTPA-R reporter protein.** A schematic representation of the DTPA-R reporter gene system composed of the reporter protein DTPA-R and the cognate PET reporter probe [18F]F-DTPA•metal. **a**, A molecular model (PyMol) based on the crystal structure of the anticalin/CHX-A″-DTPA•Y complex (PDB ID: 4IAX) and the NMR structure of the CD4 transmembrane domain (PDB ID: 2KLU). **b**, The design of the coding region for the anticalin-based reporter protein with a promoter, the Lcn2 signal peptide (sp), the mature anticalin, the V5-tag, the CD4 transmembrane domain and, optionally, a fluorescent protein. **c**,**d**, The chemical structure of the PET radioligand [18F]F-Nic-Glu₂-PEG₄-CHX-A″-DTPA•metal, dubbed [18F]F-DTPA (**c**), and the ligand moiety of [18F]F-colchicine (**d**). **e**, Fluorescence microscopy of PC3^DTPA-R and PC3^Colchi-R cells stained with Hoechst 33342 (cell nucleus) and an AlexaFluor488-conjugated anti-V5-tag antibody (reporter protein). **f**–**h**, Experiments conducted with Jurkat lines, created by retroviral transduction, and FACS isolation of the 10% highest-expressing clones: quantification of reporter protein surface densities by flow cytometry with MESF beads (four biological replicates; mean with s.d.; statistical analysis: multiple unpaired Student's *t*-test) (**f**); analysis of transduced Jurkat cell lines for their proliferation kinetics using the CFSE assay (median doubling time is shown in black bars and the quartiles as white lines) (**g**); isolation of transgenic Jurkat^DTPA-R cells from a 5:95 mixture with wild-type Jurkat cells using the anti-V5-tag antibody for MACS (**h**).

A potential metabolic burden of the reporter gene expression was assessed by measuring doubling times of non-activated or PMA–ionomycin-activated Jurkat cell lines using a carboxyfluorescein succinimidyl ester (CFSE)-based proliferation assay (Fig. 1g and Supplementary Fig. 3). The doubling time of wild-type Jurkat cells (non-activated: median 20.0 h/activated: median 24.1 h) was not substantially changed by reporter gene expression (DTPA-R: median 18.5 h/22.4 h).

Further applications of the V5-tag as part of the DTPA-R include the isolation of reporter gene expressing cells using magnetic-activated cell sorting (MACS). Using the V5-tag, Jurkat$^{DTPA-R}$ cells could be isolated by MACS[27] from a 5:95 cell mixture with high efficacy and purity (>98%) (Fig. 1h). Finally, the V5-tag also enabled the detection of the reporter protein at the cellular level via immunohistochemistry (IHC; Extended Data Fig. 3).

## Synthesis and characterization of [$^{18}$F]F-colchicine and [$^{18}$F]F-DTPA

First, we investigated the binding of $NH_2$-CHX-A″-DTPA in complex with $^{90}Y^{III}$ to Jurkat$^{DTPA-R}$ cells using a competitive assay, resulting in a half-maximal inhibitory concentration (IC$_{50}$) of 1.4 nM (Fig. 2a). However, fluorine-18 is the preferred PET radioisotope as it is widely available and almost every decay leads to the emission of a positron (ratio 96.9%) with desirably low energy ($E_{mean}$ = 250 keV, average positron range of 0.6 mm (ref. [28])). Meanwhile, positron-emitting radiometals have inferior physical characteristics, and their $NH_2$-CHX-A″-DTPA complexes often bind with lower affinity to DTPA-R than $Y^{III}$ or $Tb^{III}$ complexes[17,18]. For this reason, we designed a DTPA-based radioligand that can be charged with radioactive or non-radioactive metal ions and features a second group that can be easily radiolabelled with fluorine-18 (Fig. 2b). This radiolabelling approach uses a trimethyl-amine leaving group[29], similar to the clinically approved radioligand [$^{18}$F]PSMA-1007 (ref. [30]). For comparison, we synthesized analogous radioligand precursors for Colchi-R using different numbers of strongly polar Glu residues and a precursor with two Glu residues for DTPA-R (Supplementary Figs. 4 and 5). After quality control, both compounds were radio-fluorinated in a single step and purified by reverse-phase high-performance liquid chromatography (HPLC) to obtain highly pure radioligands (Fig. 2c and Extended Data Fig. 4a,b). For [$^{18}$F]F-Nic-D-Glu$_2$-PEG$_4$-CHX-A″-DTPA•Tb$^{III}$ (dubbed [$^{18}$F]F-DTPA) a radiochemical yield of ~20% and a radiochemical purity of >98% were achieved. Subsequently, the $^{18}$F-labelled radioligands were tested in ligand binding assays with DTPA-R- and Colchi-R-expressing PC3 cells (Fig. 2d and Extended Data Figs. 4c and 5a). The binding of both radioligands was highly specific to cells expressing the corresponding receptor (~1,000-fold for DTPA-R and ~500-fold for Colchi-R) and could be efficiently blocked by competition with non-radioactive ligand (~100-fold for DTPA-R and ~400-fold for Colchi-R). As DTPA-R does not comprise the CD4 internalization sequence, the majority of the [$^{18}$F]F-DTPA bound to DTPA-R remained on the cell surface, which was confirmed by enzymatic cleavage (Fig. 2d). The binding affinities of different CHX-A″-DTPA ligands with the anticalin domain were further characterized by kinetic measurements on living cells using real-time interaction cytometry (RT-IC), as well as in vitro by surface plasmon resonance (SPR) and competitive binding assays (IC$_{50}$ assay) (Fig. 2e). All measurements consistently resulted in high affinities in the 200–600 pM range, irrespective of the ligand used (cf. Extended Data Fig. 5 and Supplementary Table 3 for details).

## Dynamic PET imaging of mice bearing xenograft tumours

As a proof of principle for reporter gene imaging, the respective radioligands were injected into CD1-nude mice carrying subcutaneous PC3$^{DTPA-R}$ and PC3$^{Colchi-R}$ xenograft tumours above the right and left shoulder, respectively (Fig. 3a,b and Extended Data Fig. 4d). Dynamic [$^{18}$F]F-DTPA PET imaging showed fast, almost exclusively renal, clearance, accompanied by increasing signals in the kidneys, ureter and urinary bladder, with no retention in other tissues (Fig. 3a). The radioligand showed rapid accumulation in the PC3$^{DTPA-R}$ xenograft tumour, resulting in a stable activity concentration of 22.8 ± 0.6% injected dose (ID)$_{max}$ per gram at $t$ = 30–90 min post-injection (p.i.). While both xenograft tumours were clearly visible on magnetic resonance (MR) images (Fig. 3b), only the PC3$^{DTPA-R}$ tumour could be delineated in PET. The maximum activity concentration in the PC3$^{Colchi-R}$ xenograft (0.18%

ID$_{max}$ per gram) was 125-fold lower than for the cognate PC3$^{DTPA-R}$ xenograft. [$^{18}$F]F-DTPA in the blood pool was quickly cleared with an α-phase distribution of 0.5–1.5 min, which resulted in a tumour-to-blood ratio of 261 after 90 min. Furthermore, in a second static PET scan 6 h p.i., still 48% of the initial PC3$^{DTPA-R}$ signal was retained (Fig. 3c,d). Overall, these results indicate an exquisite specificity of the DTPA-R reporter gene system in vivo.

The biodistribution of [$^{18}$F]F-DTPA at 90 min p.i. in healthy female and male mice was analysed ex vivo to study sex-specific differences ($n$ = 5 female and 5 male). Very low retention of below 0.2% ID per gram was measured for most tissues (Fig. 3e), except for the kidneys (0.9% ID per gram) and the hepato-biliary excretion route (small intestine: 0.2% ID per gram; large intestine: 0.1% ID per gram). No significant sex-specific differences in biodistribution were observed (Fig. 3e).

In vivo stability of [$^{18}$F]F-DTPA was analysed after intravenous (i.v.) injection using size-exclusion chromatography (SEC) to separate intact radioligand from fragments with lower molecular weight. [$^{18}$F]F-DTPA within urine samples collected from two mice ~3 h p.i. was >97% intact, indicating serum stability (Fig. 3f).

For comparison, a dynamic PET study was conducted for the radioligand [$^{18}$F]F-Nic-Glu$_2$-PEG$_4$-colchicine ([$^{18}$F]F-colchicine). This radioligand showed not only renal (19.8% ID) but also substantial hepato-biliary excretion (69.7% ID), which is undesirable due to the strong and variable background PET signals in the gastrointestinal tract (Extended Data Fig. 4d,e). The elevated hepato-biliary excretion of [$^{18}$F]F-colchicine is probably explained by its lower hydrophilicity, as reflected by the octanol/phosphate buffered saline (PBS, pH 7.4) partitioning coefficient (log$D_{7.4}$) of −2.97 versus −3.58 for [$^{18}$F]F-DTPA. Taken together, these pharmacokinetic data, as well as the higher expression levels of the DTPA-R on transduced cells and the much higher radioligand uptake in the xenograft tumour, prompted us to focus on the further development of the DTPA-R reporter system.

## Design and evaluation of CAR T cells expressing DTPA-R

To investigate the application of DTPA-R in the context of human CAR T cell therapy, we assessed in an animal study whether the migration, proliferation and tissue infiltration of CAR T$^{DTPA-R}$ cells can be imaged. To this end, we used a well-characterized CD19-targeting second-generation CAR based on the scFv FMC63 (Fig. 4a)[31]. The corresponding expression cassette comprised the CAR followed by a 2A self-cleaving sequence separating a second membrane protein, the truncated epidermal growth factor receptor (EGFRt)[32]. EGFRt lacks normal EGFR function but is still recognized by cetuximab (Erbitux) for in vivo cell ablation[33]. To generate CAR T$^{DTPA-R}$ cells, we replaced the EGFRt in the retroviral expression cassette with DTPA-R (Fig. 4b). This enables PET imaging of the CAR T cells and optionally serves for in vivo cell ablation by CHX-A″-DTPA based ligands charged with therapeutic radiometal ions, such as $^{161}Tb^{III}$. Both CAR T vectors were used to transduce human peripheral blood mononuclear cells (PBMCs), resulting in a surface expression of approximately 100,000 DTPA-R molecules per CAR T cell (Fig. 4c).

There was no substantial difference in CD3, CD4 and CXCR3 expression levels in CAR T$^{EGFRt}$ and CAR T$^{DTPA-R}$ cells (Fig. 4c). CAR T cell function was assessed in a real-time killing assay of HEK$^{CD19}$ target cells (xCELLigence; Fig. 4d) as well as a $^{51}$Cr-release assay on NALM6 and Raji cells (Fig. 4e). Specific lysis was comparable between the αCD19-CAR T$^{EGFRt}$ and αCD19-CAR T$^{DTPA-R}$ cell-treated groups in both in vitro assays. Primary CAR T cells showed comparable transduction efficacies as well as CD4/CD8 and T cell subset ratios (Fig. 4f–h). The therapeutic efficacy of both αCD19-CAR T cells was studied in non-obese diabetic severe combined immunodeficiency gamma chain null (NSG) mice engrafted intravenously with NALM6$^{GFP-fLuc}$ lymphoma cells ($0.5 × 10^6$), followed by αCD19-CAR T$^{DTPA-R}$, αCD19-CAR T$^{EGFRt}$ or untransduced (mock) T cells ($10 × 10^6$ each). The initial clearance of the lymphoma was visualized using bioluminescence imaging (days 0, 7 and 12; Fig. 4i,j),

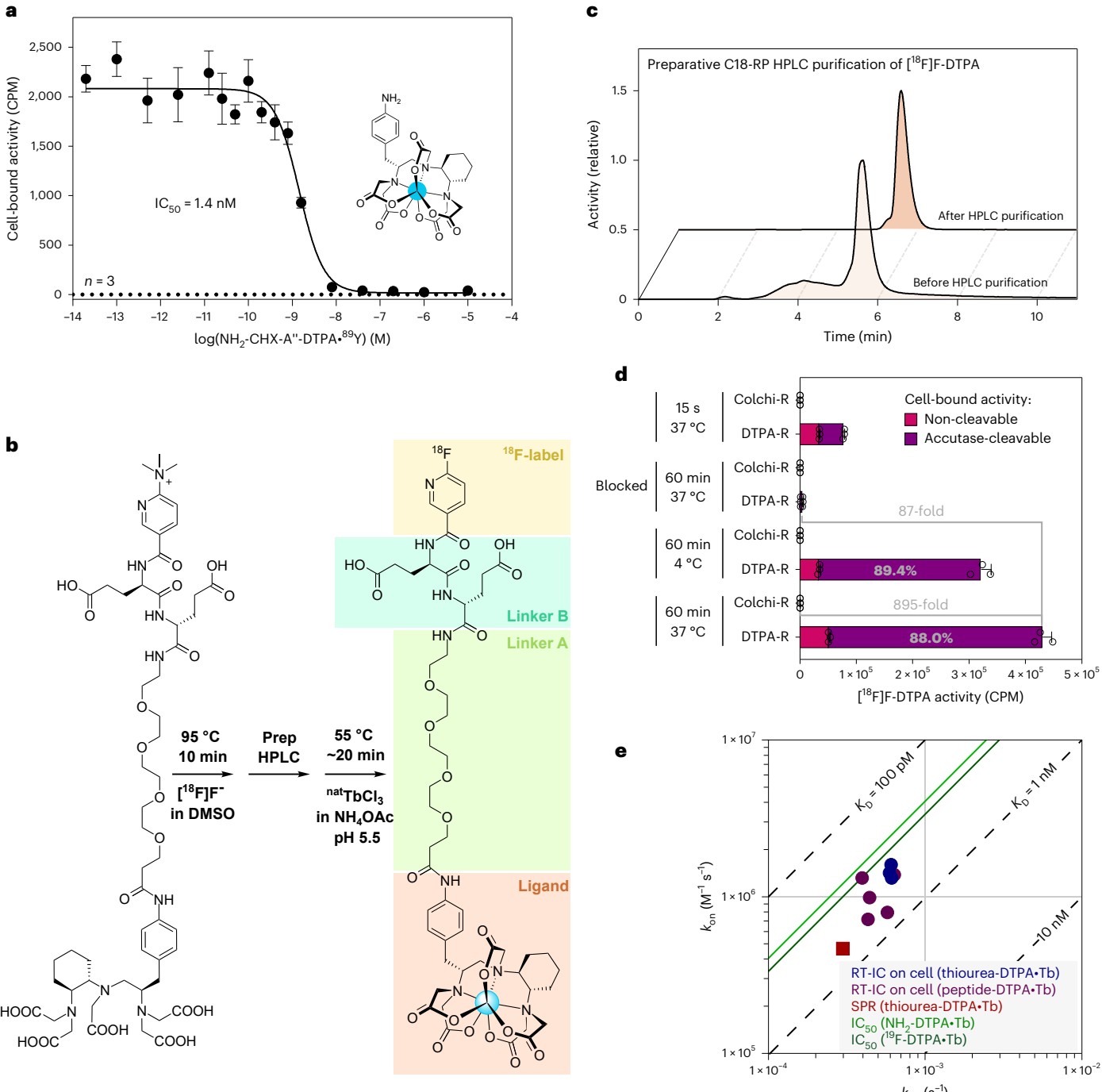

**Fig. 2 | Preparation and characterization of the [18F]F-DTPA radioligand.**
**a**, Binding of NH₂-CHX-A″-DTPA•⁹⁰Y competed by NH₂-CHX-A″-DTPA•⁸⁹Y to Jurkat cells expressing DTPA-R–mRuby3 (mean of triplicates with s.d. of one experiment). **b**, The radiosynthesis procedure including radiofluorination, preparative HPLC and charging of the CHX-A″-DTPA chelator with natTbIII. **c**, Quality control by analytical HPLC of the radiosynthesis product before and after preparative HPLC purification. **d**, Analysis of the internalization of [18F]F-DTPA

upon binding to PC3DTPA-R/Colchi-R cells by analysing the Accutase-cleavable and the non-cleavable fraction (mean of triplicates with s.d. of one experiment). Accutase efficiently cleaves the DTPA-R ectodomain and, thus, allows the identification of bound radioligand that is accessible on the cell surface in counts per minute (CPM). **e**, A $k_{off}$–$k_{on}$ plot summarizing results from RT-IC, SPR spectroscopy and competitive binding assays (IC₅₀). For detailed information on affinity measurements, see Extended Data Fig. 5.

and, subsequently, body weight trajectories (Fig. 4k) and survival were assessed (Fig. 4l), confirming comparable in vivo functionality of the CAR TDTPA-R and CAR TEGFRt cells.

## PET imaging of CAR T cell therapy

To demonstrate PET-based therapy imaging, CAR T cells were used to treat NSG mice engrafted with a systemic Raji tumour, which is

known to primarily home to the bone marrow[34]. After 7 days of Raji engraftment, $2 \times 10^6$ αCD19-CAR TDTPA-R or reference αCD19-CAR TEGFRt cells were infused. This experimental setting ensured constant CAR T expansion due to non-complete tumour eradication, which was imaged with [18F]F-DTPA on days 4, 8, 15, 22 and 30 (Fig. 5a). PET signals indicated the presence of CAR TDTPA-R cells on day 4 in the spleen and from day 8 onwards, with increasing intensity, in the bone marrow of the

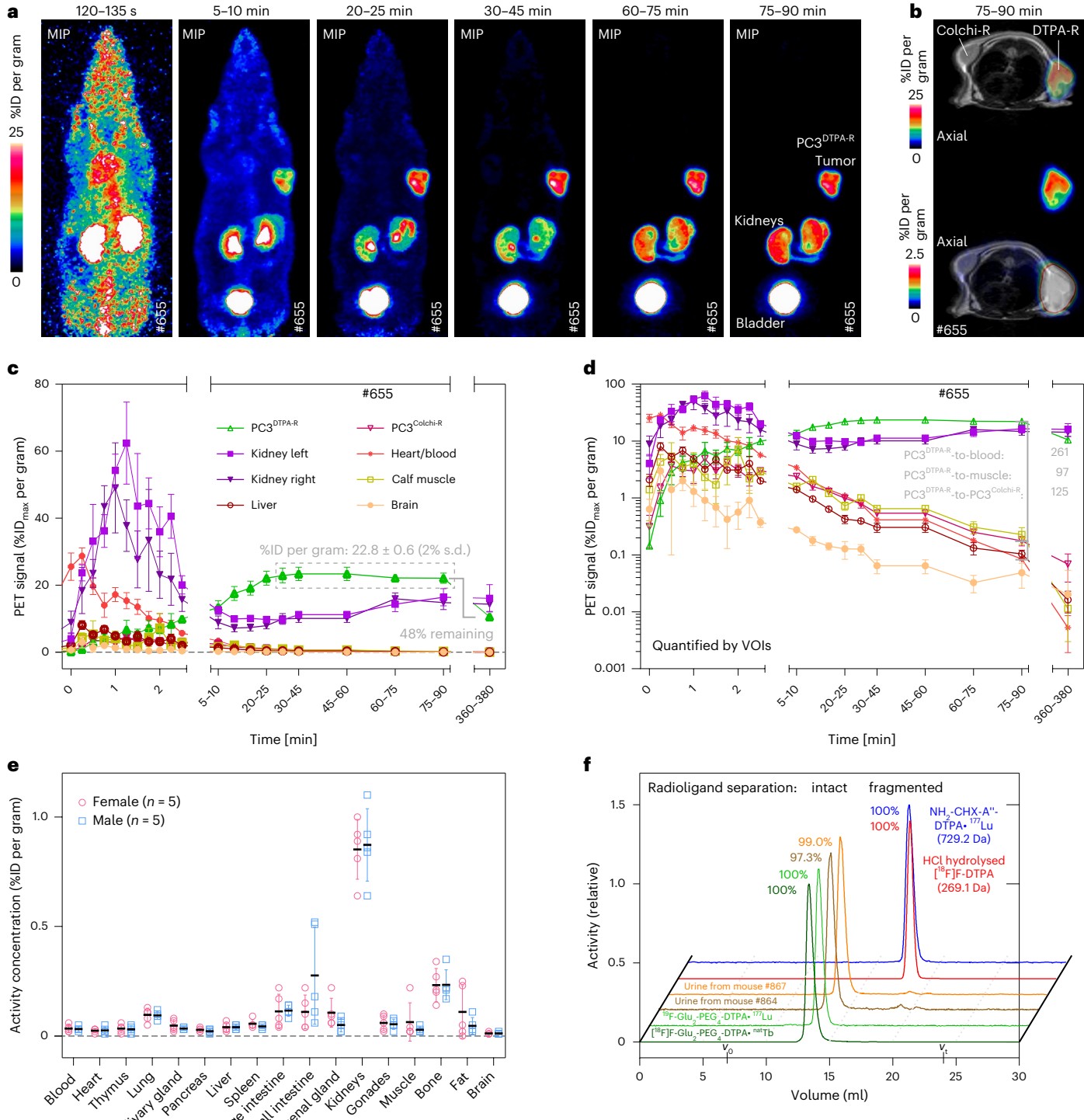

**Fig. 3 | Pharmacokinetic and stability of [¹⁸F]F-DTPA in vivo. a–d**, A dynamic PET scan of CD1-nude mice carrying subcutaneous PC3^DTPA-R (right shoulder) and PC3^Colchi-R (left shoulder) xenograft tumours: maximum intensity projections (MIPs) of a dynamic PET scan between 2 min and 90 min p.i. of 11.4 MBq [¹⁸F]F-DTPA (**a**) and an axial PET plane through both tumours at 0–25% ID per gram and 0–2.5% ID per gram (**b**), showing selective accumulation in the PC3^DTPA-R tumour but no elevated signal in the PC3^Colchi-R tumour; linear (**c**) and logarithmic (**d**) representation of the time–activity curve derived from the dynamic PET scan. VOI quantification using ten pixel spheres revealed increasing signal-to-background ratios over time. Mean with s.d. of one spherical VOI.

**e**, Ex vivo biodistribution analysis at $t = 90$ min after [¹⁸F]F-DTPA i.v. injection into male and female C57BL/6 mice. Please note that mice were awake between injection and $t = 90$ min, and not under anaesthesia as for the dynamic PET scan, causing different pharmacokinetic profiles. No significant differences were found (statistical analysis: multiple unpaired Student's $t$-test, Holm–Šidák correction for multiple comparison, mean with s.d., biological replicates). **f**, SEC of the intact [¹⁸F]F-DTPA radioligand (in complex with ^natTb), its hydrolysis fragments (¹⁸F-nicotinic acid) and NH₂-CHX-A''-DTPA•^177Lu, and urine samples collected from mice injected beforehand with [¹⁸F]F-DTPA.

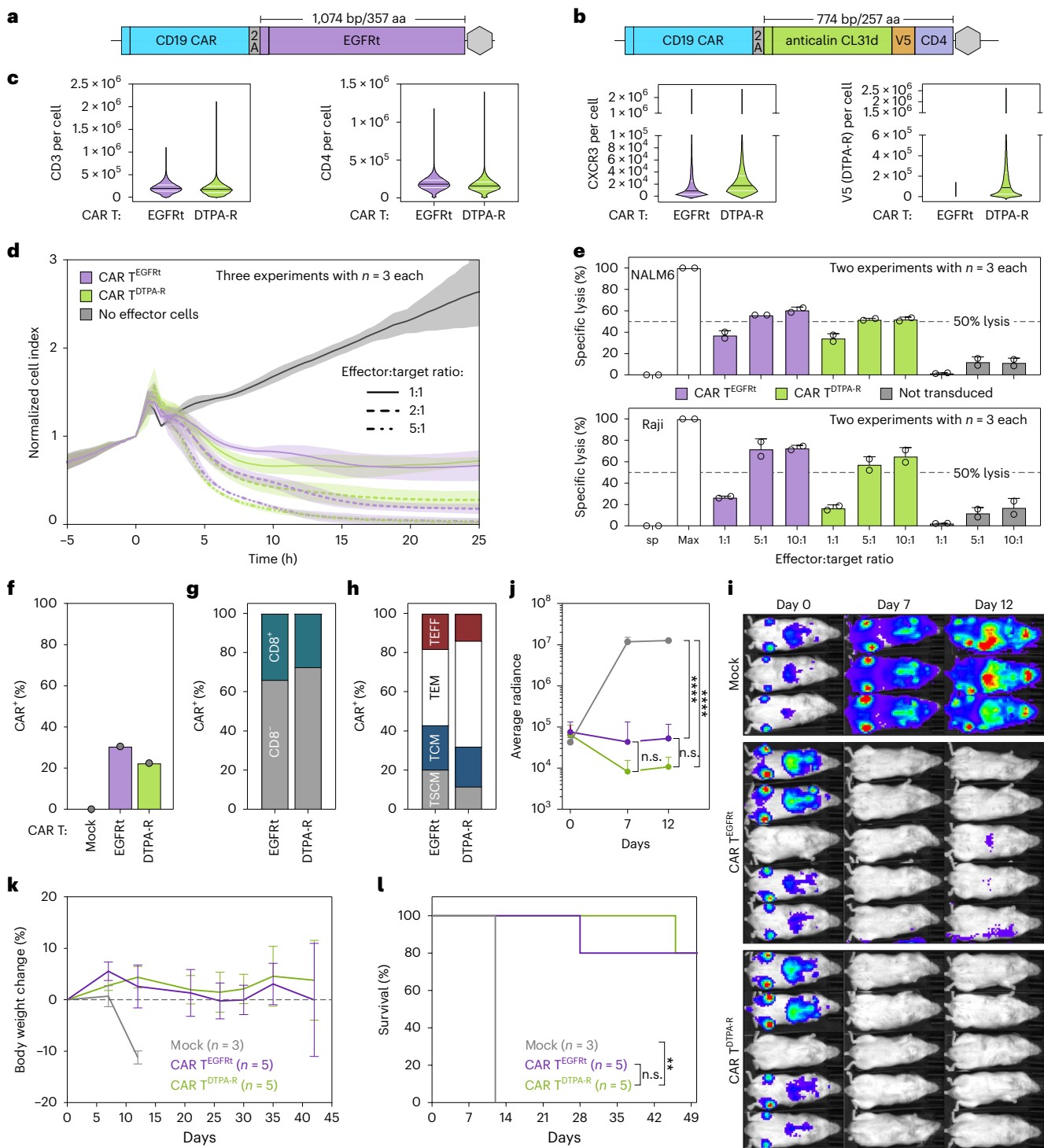

**Fig. 4 | Characterization of CAR T cells expressing EGFRt or DTPA-R. a,b,** Based on an established retroviral vector encoding a CD19 CAR and the truncated EGFR (EGFRt) (**a**), a new plasmid in which EGFRt was replaced by the DTPA-R gene (**b**) was constructed. **c–e,** These two plasmids were used for retroviral gene transfer into human PBMCs, and these CAR T cells were then subjected to different assays: the surface expression of important T cell surface markers was quantified by flow cytometry using MESF beads, including the T cell receptor (CD3), the co-stimulatory receptor CD4 and a chemokine receptor CXCR3 as well as the DTPA-R reporter gene (V5-tag) (the median receptor number is shown in black bars and the quartiles as white lines; for EGFRt CAR T cells, the median and quartiles for V5 (DTPA-R) are ~zero and therefore not visible) (**c**); the effect of different CAR T cells on HEK293^CD19 target cell confluence was measured in real time by changes in electric current (xCELLigence; mean as lines; s.d. as shaded area, biological repeats) (**d**); cell lysis of CD19-positive tumour cell lines NALM6 and

Raji was investigated by loading the tumour cells with radioactive chromium-51 and quantifying radioactivity in the supernatant after a 4-h incubation in the presence of respective CAR T cells (CAR T^EGFRt and CAR T^DTPA-R showed no significant differences between same conditions; statistical analysis: one-way ANOVA, Tukey's correction for multiple comparison, mean with s.d., biological replicates; sp, spontaneous release) (**e**). **f–l,** Functional comparison of CAR T^EGFRt and CAR T^DTPA-R cells, including transduction rate (**f**), CD4/CD8 ratio (**g**), T cell subtype analysis (**h**), bioluminescence imaging of NALM6 tumour cells in NSG mice treated at t = 0 with CAR T cell therapies (**i**) and the respective biostatistical analysis (**j**; statistical analysis: two-way ANOVA, Tukey's correction for multiple comparisons, mean with s.d., n = 5 or n = 3 (mock), biological replicates; n.s., not significant); long-term analysis of therapeutic efficacy assessment by body weight trajectories (mean with s.d., biological replicates) (**k**) and Kaplan–Meier plot (**l**; statistical analysis: log-rank test, mock versus DTPA-R P = 0.0082).

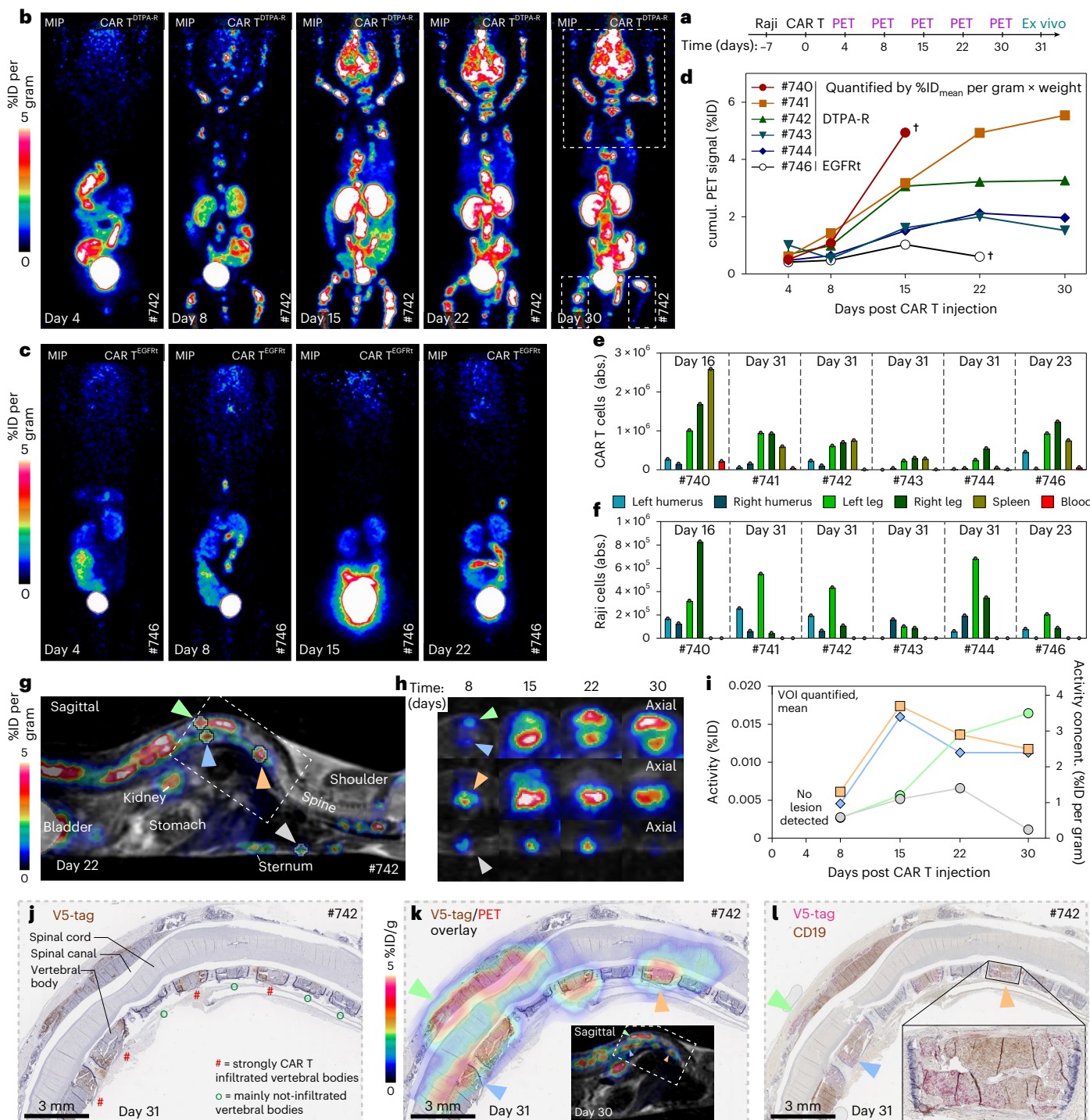

**Fig. 5 | Longitudinal PET imaging study of DTPA-R-expressing CAR T cells.**
**a**, The longitudinal CAR T cell study design involved NSG mice injected intravenously with Raji tumour cells ($0.5 \times 10^6$) and, after 7 days, i.v. injection of $2 \times 10^6$ DTPA-R- or EGFRt-expressing anti-CD19 CAR T cells. PET scans at $t = 90$ min p.i. were recorded during treatment on days 4, 8, 15, 22 and 30. **b,c**, MIP of an exemplary αCD19-CAR T$^{DTPA-R}$ (**b**) and αCD19-CAR T$^{EGFRt}$ (**c**) mouse, each depicted for the respective days. Note that [$^{18}$F]F-DTPA bound to a small amount of shedded DTPA-R leads to increased kidney signals. **d**, The cumulative (cumul.) specific signal from every [$^{18}$F]F-DTPA scan was quantified for the upper body (above the gall bladder) and the hind legs (excluding kidneys, guts and urinary bladder). This quantification allowed monitoring of global CAR T cell expansion and/or decline over time. **e,f**, FACS analysis of collected absolute (abs.) cell numbers of Raji tumour cells (**e**) and CAR T cells (**f**) in different extremities, spleen (normalized to 50 mg) and blood (normalized to 100 μl). **g–j**, Quantification of local CAR T infiltrations over the course of the treatment: a sagittal PET–MR scan on day 22 revealed different bone marrow infiltrations (**g**); axial PET–MR enabled the quantitative assessment of local CAR T cell infiltration over time (**h**); longitudinal VOI quantification (**i**) allowed the differentiation between tumour lesions experiencing an intermediate CAR T expansion and subsequent decline (grey, blue and orange arrowheads in **g** and **h**) and lesions with constant CAR T infiltration and/or expansion (green arrowhead in **g** and **h**); concent. indicates concentration; IHC of sagittal spine sections containing (green, blue and orange) lesions with infiltrating CAR T cells (V5-tag, brown) (**j**). **k**, An overlay of PET signals on the V5-IHC. **l**, Double staining of a consecutive tissue section for CAR T cells (V5-tag, red) and Raji lymphoma cells (CD19, brown).

extremities as well as spine, skull and axillary lymph nodes (Fig. 5b or Extended Data Fig. 6 for all animals). By contrast, PET signals were absent in mice injected with αCD19-CAR T$^{EGFRt}$ cells, demonstrating the specificity of [$^{18}$F]F-DTPA (Fig. 5c).

Quantification of the cumulated specific PET signals allowed the assessment of CAR T cell expansion over time (Fig. 5d). Furthermore, we used flow cytometry at the end of the study, or when the humane endpoint was reached, to quantify the numbers of CAR T (Fig. 5e) and Raji cells (Fig. 5f). This ex vivo analysis showed differences in CAR T cell numbers, confirming the high variability between animals and the need for imaging of these therapies. In addition, three-dimensional (3D) cross-sectional images of the [$^{18}$F]F-DTPA PET–MR enabled the monitoring of local CAR T infiltration (Fig. 5g–i). Longitudinal PET analysis revealed different trajectories of CAR T infiltrated lesions (Fig. 5h,i). Ex vivo analysis by V5-IHC of sagittal spine sections allowed CAR T$^{DTPA-R}$ identification on the cellular level, which confirmed CAR T infiltration into individual vertebra bodies (~1.5 × 0.5 mm) and, occasionally, invasion through the vertebral bone into the spinal canal (Fig. 5j). Importantly, the pattern of CAR T infiltration determined by IHC was very well matched by the signals in PET–MR (Fig. 5k). Furthermore, the co-localization of CAR T$^{DTPA-R}$ with Raji cells was confirmed by IHC co-staining of CD19 and the V5-tag (Fig. 5l). Interestingly, lesions with a CAR T cell plateau in longitudinal PET imaging (Fig. 5i, blue and orange) showed lower tumour cell density, which indicated better tumour clearance by CAR T cells compared with neoplasms with constantly increasing PET signal (Fig. 5i, green, and Extended Data Fig. 7a). Other organs such as the spleen, kidneys, liver and lung showed only few CAR T$^{DTPA-R}$ cells in V5-IHC (Extended Data Fig. 7b).

Correlation between the actual number of CAR T$^{DTPA-R}$ cells detected by flow cytometry after extraction from a hollow bone and the cumulated PET signal obtained for the same extremity (Fig. 6a and Extended Data Fig. 8a, b) was determined for mice euthanized on day 8 or 15 (Fig. 6b; for gating, see Extended Data Fig. 8c) and on day 16 or 31 (Extended Data Fig. 8d). A linear relationship ($R^2 = 0.92$) indicated that PET imaging of DTPA-R can quantitatively monitor tissue-specific CAR T cell infiltration. Utilizing the resulting regression line, we calculated a detection limit of around 1,200 CAR T$^{DTPA-R}$ cells in delineated lesions in the bone (Fig. 6c). To further characterize the relationship between the PET signal and the number of DTPA-R-expressing cells, we measured the signal of defined numbers of Jurkat$^{DTPA-R}$ cells in a PET phantom in vitro. For cells labelled with [$^{18}$F]F-DTPA, we observed a linear correlation ($R^2 = 0.999$) with a detection limit of 500 cells (Fig. 6d). In the next step, we determined the detection limit in tissue with low perfusion. Therefore, we injected 4,000–64,000 Jurkat$^{DTPA-R}$ cells, or CAR T$^{DTPA-R}$ cells, in the dorsal subcutis, immediately followed by i.v. injection of [$^{18}$F]F-DTPA (Fig. 6e,f). Here, 8,000 cells were clearly detectable by PET. Again, a linear correlation between cell number and PET signal was observed ($R^2 = 0.97$ and $R^2 = 0.99$, respectively; Fig. 6e,f).

## PET imaging of gene transfer mediated by AAV9 viral vectors

In vivo gene therapy represents another relevant application scenario for reporter gene imaging, where PET imaging can help to assess the roles of administration routes, dosing regimens, pharmacokinetics of capsid-modified vectors and endurance of gene expression. As a common example for gene therapy, we have selected vectors based on the AAV[35]. In particular, we have focused on the serotype AAV9, which sparked great interest for transducing muscle and neuronal tissue[35] and provided the basis for the Food and Drug Administration-approved drug Zolgensma[36]. Systemic AAV9 administration is described to cause strong transduction of the liver, different muscles (including the heart) and, to a lesser degree, the lung and brain[36]. We constructed a transfer plasmid flanked by AAV2 inverted terminal repeats and used it to produce AAV2/9 viral vectors encoding the DTPA-R reporter gene under the control of the CMV promoter (Fig. 7a). First, we investigated the reporter gene expression in mice after systemic injection of AAV9$^{DTPA-R}$.

Immunocompetent C57BL/6 (Fig. 7b) and immunocompromised CD1-nude (Fig. 7c) mice ($n = 3$ per group) were dosed with $2.5 × 10^{12}$ viral genomes (vg) per mouse, and on days 14, 21 and 28 [$^{18}$F]F-DTPA PET scans were recorded (Fig. 7b,c). Although cohorts were treated in parallel using the same viral vector batch, there were marked differences between the transduction patterns obtained in C57BL/6 and CD1-nude mice. Isocontour segmentation of the heart yielded a value of 15.3 ± 3.8% ID$_{max}$ per gram for C57BL/6 and 28.7 ± 5.4% ID$_{max}$ per gram for CD1-nude mice. Within each cohort, animals showed comparable transduction patterns (Extended Data Fig. 9). Moreover, we observed consistent trajectories of the repeated PET measurements, indicating stability of the vector expression over time (Fig. 7d).

We then intravenously injected four different titres, ranging from $1 × 10^{11}$ to $2.5 × 10^{12}$ vg per mouse, into C57BL/6 mice ($n = 3$ per group). After 7 days, [$^{18}$F]F-DTPA PET–MR scans were recorded for all cohorts and an untreated control cohort (Fig. 7e and Extended Data Fig. 10 for all animals). The transduction patterns were similar to the previous experiment; for example, mice with the highest applied titre showed clear PET signals in dorsal brown adipose tissue (BAT; 7.7 ± 1.5% ID$_{max}$ per gram), heart muscle (6.1 ± 3.6% ID$_{max}$ per gram), liver (11.3 ± 1.4% ID$_{max}$ per gram) and adrenals (29.5 ± 7.0% ID$_{max}$ per gram), whereas the background uptake in the upper body of the control mice was ~0.5 ± 0.06% ID$_{max}$ per gram. Of note, the combination of PET with high-resolution MR imaging allowed the precise analysis of transduction events (Fig. 7f), which was far beyond previous results obtained by bioluminescence imaging[36]. Lower titres resulted in gradually reduced PET signals within the respective tissues. Importantly, PET signals (%ID$_{max}$ per gram) showed a linear correlation with the administered dose in BAT ($R^2 = 0.79$), heart muscle ($R^2 = 0.61$), liver ($R^2 = 0.95$) and adrenals ($R^2 = 0.90$), which underlines the quantitative character of DTPA-R PET imaging (Fig. 7g). Well-perfused organs showed a steeper increase of the PET signal with increasing AAV titre. This observation could be due to either a higher transduction rate in these organs or a better local delivery of [$^{18}$F]F-DTPA. To test this hypothesis, we compared PET data for different tissues with perfusion values from the literature[37,38]. Indeed, a linear correlation between tissue perfusion and the slope of PET signal increase was seen (Fig. 7h). Next, we stained tissues with V5-IHC and observed transduction of individual cells, for example, in the cortex of adrenal glands (Fig. 7i). Quantification using automated positive cell detection showed 2% positive cells for animals transduced with the lowest, 33% for the low and 72.2% for the high dose, which yielded a perfectly linear relation with the injected vector titre ($R^2 = 1$; Fig. 7j). Also, the relationship between the measured PET signal and the injected AAV titre (Fig. 7k), or the transduction levels from histology (Fig. 7l), indicated a linear correlation ($R^2 = 0.99$ and 0.96, respectively). Transduction levels were further quantified by ddPCR, revealing a linear correlation between the number of vg and the PET signals in the liver ($R^2 = 0.97$; Fig. 7m). A multivariate analysis correlating the measured PET signals with injected AAV9 doses, mRNA levels and vg in the liver revealed a linear relationship ($R^2 = 1$; Fig. 7n).

## Discussion

We developed a reporter gene system based on a membrane-anchored anticalin that specifically binds a small-molecule radioligand, enabling quantitative and longitudinal PET imaging of ATMPs with remarkable specificity and sensitivity. Based on PET with [$^{18}$F]F-DTPA, we have demonstrated the suitability of DTPA-R in relevant use cases such as CAR T cell therapy of CD19 lymphoma[39] and AAV9 gene therapy[35]. CAR T cell movement and tumour homing in a mouse model of systemic lymphoma could be visualized and quantitatively tracked in vivo over 30 days. Furthermore, the transduction of AAV9 vectors in distinct tissues could be quantitatively analysed by molecular imaging. This theranostic approach, combining cell and gene therapies with a quantitative imaging modality using a universal reporter system that potentially can bridge preclinical development and clinical evaluation,

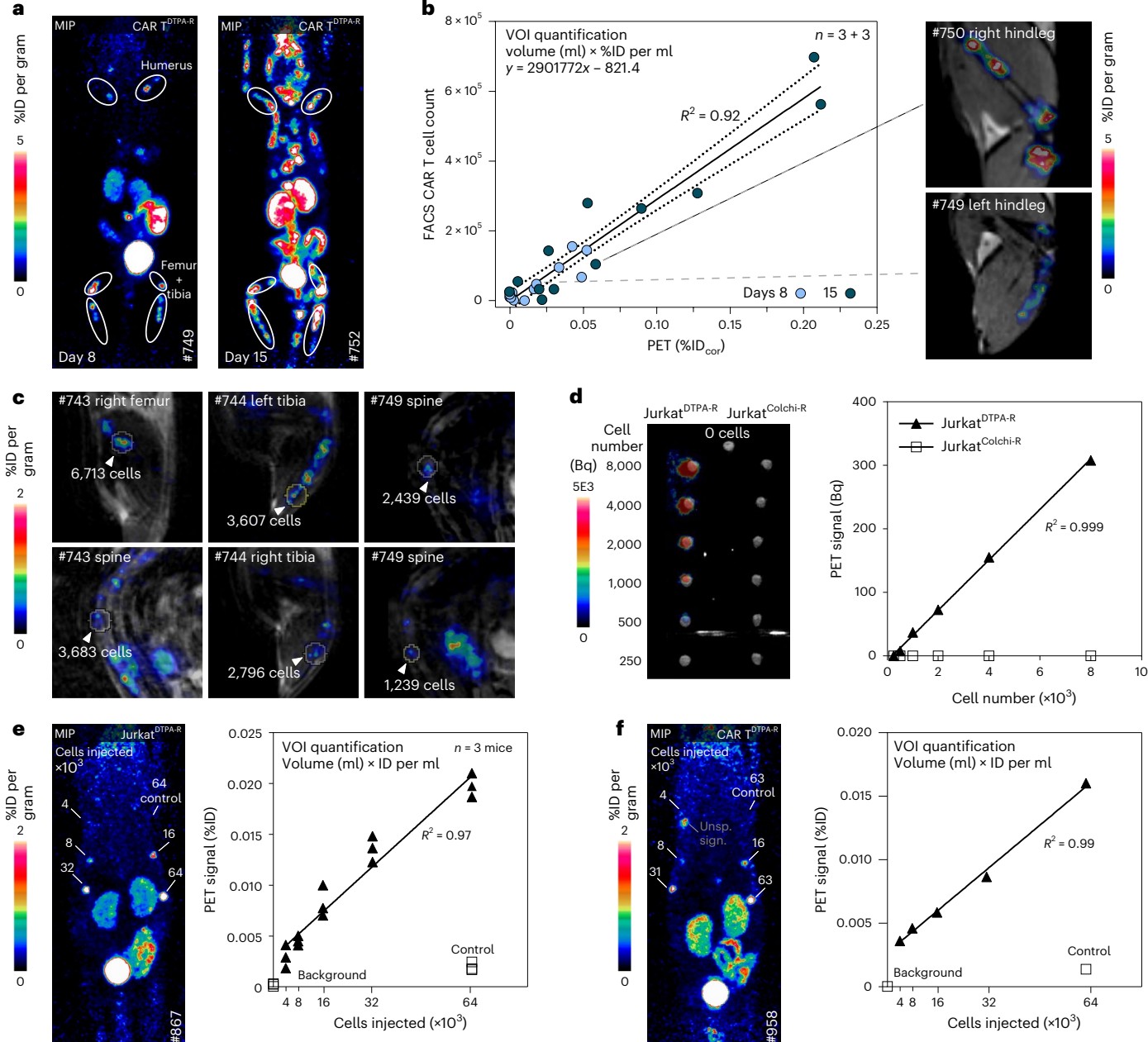

**Fig. 6 | Quantification and detection limit for DTPA-R-labelled lymphocytes. a,b,** Flow cytometry quantification of CAR T^DTPA-R cells in the bone marrow of mice after PET imaging: **a**, cells within hollow bones were collected for ex vivo analysis; correlation of CAR T cell numbers was found by flow cytometry with the PET signal (corrected %ID (%IDcor)) obtained from hollow bones (with exemplary PET−MR cross-section images) (**b**). Linear regression with 95% confidence interval. **c**, Individual lesions in the bone marrow were quantified, and CAR T^DTPA-R cell numbers were calculated by interpolation using the regression line from **b**. **d**, Maximum signal capacity was assessed by staining Jurkat^DTPA-R cells in vitro with

[^18F]F-DTPA. PET phantom study: Jurkat^DTPA-R or Jurkat^Colchi-R cells were incubated with [^18F]F-DTPA, washed three times and counted, and 10 μl of the suspension was transferred into PCR tubes. PET signals were recorded for 60 min, and activity was quantified. **e,f,** Spot assay: Jurkat^DTPA-R cells (**e**) or αCD19-CAR T^DTPA-R cells (**f**) were injected subcutaneously into the back of mice. Five samples of a 1:2 dilution series of the DTPA-R-expressing cells and of a control cell line were used. 90 min after i.v. [^18F]F-DTPA injection, PET scans were recorded for 20 min. The %ID of the spots was quantified by VOIs and plotted against the number of injected cells.

should facilitate the clinical translation of ATMPs[40]. In addition, the non-invasive longitudinal monitoring of ATMP-based therapies in vivo can reduce the number of animals required per study, highlighting its utility for the implementation of the 3R principle (replacement, reduction and refinement). To this end, our PET reporter system offers promising functional features, which are described below.

High expression level of the reporter protein determines the strength of the signal, which has been demonstrated for Jurkat^DTPA-R cells displaying ~1 × 10^6 receptors per cell without measurable

negative effects on cellular fitness or T cell function. This number is approximately 10-fold higher compared with well-known lymphocyte receptors, such as the B cell receptor ($1.2 × 10^5$ copies[41]) or CD4 (~1 × 10^5 copies[42]). Direct comparison of the anticalin CL31d specific for [^18F]F-DTPA with the scFv huC825, which binds DOTA•metal, resulted in an 8.9-fold higher expression of DTPA-R (Extended Data Fig. 2a). Low surface densities for huC825-based reporter proteins are also reflected by transfected HEK293T cells expressing only ~1.5 × 10^4 receptors per cell[26]. In contrast to scFv antibody fragments, known for their

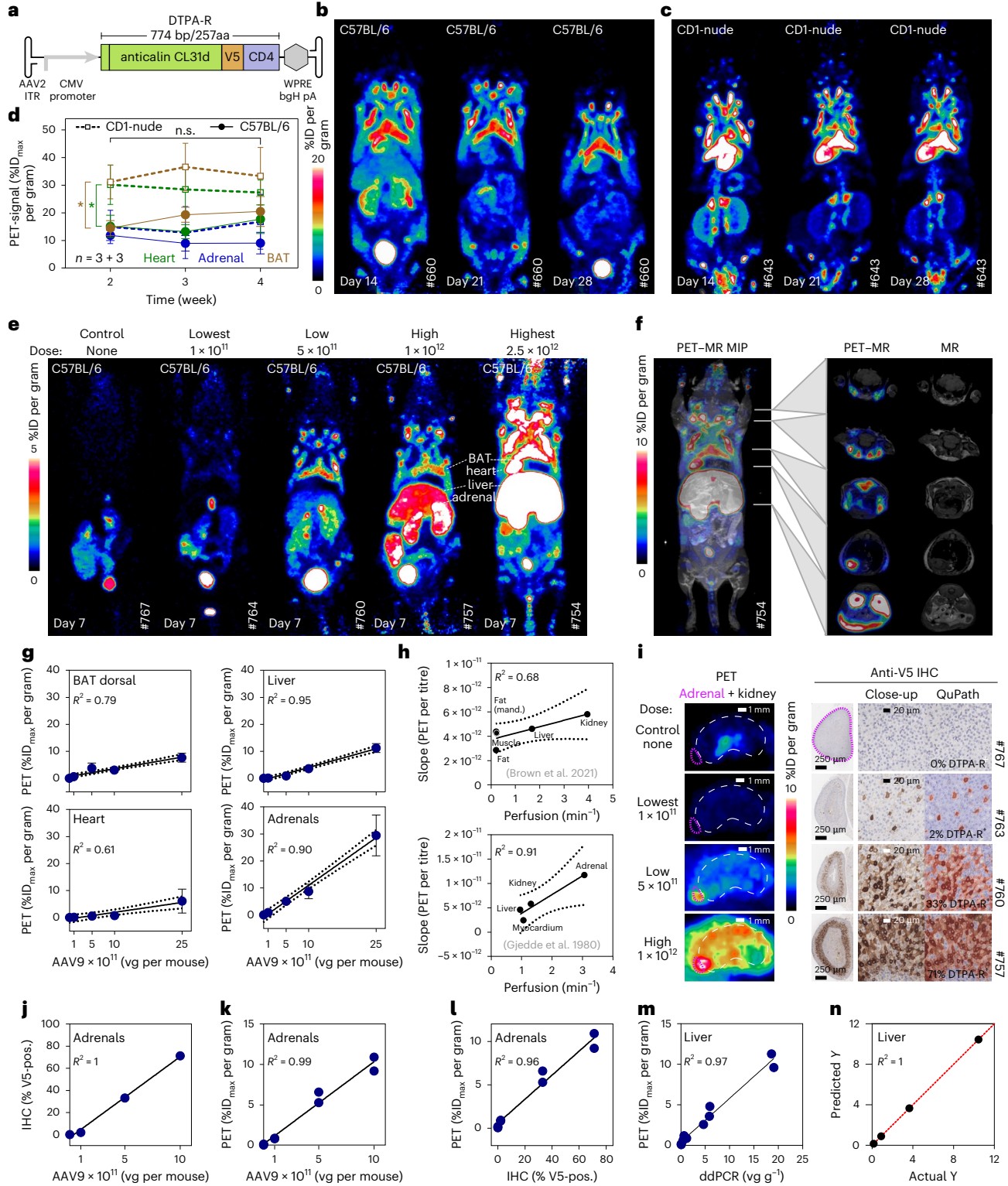

**Fig. 7 | PET imaging of AAV9 gene therapy via DTPA-R. a**, C57BL/6 or CD1-nude mice were injected intravenously with AAV9/2 viral vectors encoding the DTPA-R. **b,c**, MIPs (90 min p.i. [$^{18}$F]F-DTPA) are shown for a longitudinal study of immunocompetent C57BL/6 mice (**b**) and immunocompromised CD1-nude mice (**c**) at $t$ = 14, 21 and 28 days after injection of $2.5 \times 10^{12}$ vg per mouse. **d**, Quantification of PET signals by VOI spheres in selected organs throughout the study (mean with s.d., biological replicates, statistical analysis: unpaired Student's $t$-test, two-tailed; heart $P$ = 0.034, BAT $P$ = 0.012). **e**, MIP of C57BL/6 mice transduced with different titres ($1 \times 10^{11}$, $5 \times 10^{11}$, $1 \times 10^{12}$ and $2.5 \times 10^{12}$ vg per mouse) as well as untreated control mice imaged after 7 days. **f**, MIP and axial PET–MR overlays allow the identification of anatomical structures accumulating

[$^{18}$F]F-DTPA. **g**, Correlation of injected AAV9 titre and quantified PET signals for BAT, liver, heart and adrenal glands ($n$ = 3 per titre, biological replicates, mean with s.d.). **h**, The slope of the increase in PET signals was correlated with the perfusion of the respective organ (obtained from literature)[37,38]. **i**, Sagittal PET and V5-IHC of the adrenal gland with quantification of positive cells in the zona fasciculata of the adrenal gland cortex. **j–l**, Correlations of positive cell count (**j**) and PET signal (**k**) with the injected viral vector dose, as well as with each other (**l**) ($n$ = 1 per titre). **m**, Correlation of detected vector genomes and PET signal. **n**, Multivariate analysis correlating the PET signal (actual $Y$) with AAV9 dose, mRNA and vg (predicted $Y$) in the liver.

oligomerization and aggregation tendencies[43], anticalins possess a robust fold, are composed of a single polypeptide chain and can be expressed as recombinant proteins at high levels[19].

Minimal gene size is of importance due to the limited packaging capacities of viral and non-viral gene shuttles and met by DTPA-R, which is encoded by just 774 bp (~26.4 kDa for the mature fusion protein), in contrast to most other PET reporter proteins, including HSV1-tk (1,131 bp)[12], NIS (1,932 bp)[13,44], tPSMA$^{(N9del)}$ (2,226 bp)[16], SSTR2 (1,110 bp)[14], DAbR1-2A-GFP (2,280 bp)[45], SNAPtag (846 bp)[46], eDHFR (480 bp)[47] and EGFRt (1,074 bp)[32].

Functional inertness, including lack of biological activity, interference and toxicity of both the reporter gene and probe, is a crucial factor as the capability of in vivo imaging is second to the therapeutic function of an ATMP. We have investigated the proliferation kinetics, activation status, receptor expression and cellular toxicity of CAR T cells and found no difference due to the DTPA-R expression compared with EGFRt. Given the binding activity of DTPA-R for an exogenous hapten, interferences with cellular processes are much less likely than for reporter proteins that lead to ion transport over the cell membrane (NIS), represent tumour-associated surface antigens (tPSMA and SSTR2) or possess catalytic activities (HSV-tk, tPSMA, SNAPtag and eDHFR).

However, immunogenicity of the human-derived DTPA-R protein in immunocompetent research animals is likely and must be considered when designing preclinical studies, as is known for other reporter proteins. Nevertheless, our data indicate that DTPA-R can track human CAR T cells in immunocompromised mice and monitor AAV-mediated gene transfer in immunocompromised and immunocompetent mice over several weeks. The DTPA-R system was developed with a focus on translational research applications. With respect to human studies, it is encouraging that DTPA-R is based on human lipocalins and that other anticalins have already undergone clinical testing[25]. A definitive assessment of the immunogenicity of DTPA-R in humans will require human studies, as appropriate preclinical approaches to accurately predict tolerance to engineered proteins in humans are lacking. If immunogenicity should become a concern in humans, it is possible to design a hypo-immunogenic DTPA-R version (Supplementary Fig. 1).

Stability of the PET signal over time is important to allow reproducible imaging, which relies mainly on the receptor–ligand affinity. While the soluble recombinant CL31d protein previously showed a $K_D$ value of 543 pM for the complex with CHX-A″-DTPA•Y[18], the $IC_{50}$ of $^{19}$F-Glu$_2$-PEG$_4$-CHX-A″-DTPA•$^{nat}$Tb for DTPA-R expressed on the cell surface was 299 ± 148 pM, which was comparable to the $K_D$ of 507 ± 143 pM measured by RT-IC using fluorescent CHX-A″-DTPA. Stability of the [$^{18}$F]F-DTPA signal within the tumour was demonstrated in dynamic PET scans. This is not the case for the NIS reporter gene, for example, owing to the naturally occurring efflux of the radioactive iodide[44]. Potential internalization of DTPA-R is very low due to the absence of the cytosolic CD4 domain, which triggers the internalization of CD4[48]. In addition, DTPA-R expression levels are independent of T cell activation, which is commonly known to heavily influence expression profiles in T cells[49].

A linear relation between the PET signal and the number of DTPA-R-labelled cells is mandatory to move beyond qualitative imaging. The DTPA-R system allows a strong correlation of the number of CAR T cells in the bone marrow or the applied AAV9 viral vector titre with the corresponding PET signal, thus enabling non-invasive quantification. The amount of bound [$^{18}$F]F-DTPA ligand is governed only by the local concentration of DTPA-R and the law of mass action, contrasting with the much more complex and unpredictable relationships for transporter- or enzyme-type reporter proteins. After cell death, debris containing the DTPA-R may retain some [$^{18}$F]F-DTPA binding capacity until removed by physiological mechanisms. This delayed signal loss upon cell death by genetic reporters, in the range of minutes to a few hours, is in marked contrast to direct labelling approaches

(for example, beads for MR imaging detection) where the signal is not abolished, even after phagocytosis[50].

Criteria for a corresponding radioligand[51] include its chemical composition as well as the choice of the radioisotope for PET imaging. The choice of $^{18}$F, with its ideal physical half-life, high positron yield and low positron energy, allows detection with high sensitivity and resolution, especially compared with isotopes such as $^{111}$In or $^{86}$Y, used for DAbR1 or C825 reporter probes[26,45]. Other advantages of $^{18}$F are the relatively short half-life of 109 min, which enables repeated serial imaging in a daily interval, while the half-life is long enough to allow multiple PET scans from a single batch of radioligand. In silico radiation dose estimations using the MIRDcell software[52] resulted in radiation doses that are far below doses that are known to affect CAR T function (Supplementary Fig. 6 and Supplementary Discussion 2)[53]. Thus, no functional impairment of the ATMP is expected, even for repeated diagnostic [$^{18}$F]F-DTPA scans. The stability of [$^{18}$F]F-DTPA in blood was demonstrated, together with a stable PC3$^{DTPA-R}$ signal in the time–activity curve. Nevertheless, achieving high specific activity remains a challenge for [$^{18}$F]F-DTPA synthesis, as well as for other $^{18}$F-based radioligands, and constitutes a current area of improvement.

The absence of endogenous binding activity was impressively demonstrated by the very low background accumulation of [$^{18}$F]F-DTPA in various organs. This is in marked contrast to all fully human reporter proteins described so far, including NIS with its thyroidal, gastric, mucosal, salivary and lactating mammary gland expression[54]. Furthermore, the biodistribution analysis of [$^{18}$F]F-DTPA in female and male mice showed no sex-specific differences. Excretion of [$^{18}$F]F-DTPA was seen mainly via the renal route, which led to defined signals in the kidneys and bladder. By contrast, the radioligand for the C825-scFv-based reporter gene system caused a diffuse signal in the abdomen at $t$ = 30 min p.i., and only ~80% of the $^{86}$Y-DOTA-Bn was cleared via the renal route[26]. Of note, increased specific DTPA-R signals also caused elevated kidney signals in a dose-dependent manner (Supplementary Fig. 7 and Supplementary Discussion 3). This can probably be attributed to ectodomain shedding of a small fraction of the reporter protein (Supplementary Fig. 7). The observed degree of renal uptake was smaller than that of many radiopharmaceuticals in widespread clinical use, even in animals harbouring a very large number of DTPA-R-expressing cells, and is therefore not expected to represent a relevant limitation of the reporter gene system (Supplementary Fig. 7). Furthermore, the ease of probe preparation and the price per imaging day are of importance both in biomedical research and in a clinical setting. In this regard, the DTPA-R system is attractive as the chemical synthesis of the ligand precursor is straightforward. Furthermore, radiolabelling involves only a few steps, and the supply of [$^{18}$F]F$^-$ is neither limiting nor expensive.

Finally, the sensitivity of signal detection is a crucial aspect, which depends on the number of binding sites, radioligand affinity, physical decay characteristics of the isotope and the specific activity of the radioligand. Using a PET phantom, we determined a detection limit of 500 cells, while a standard [$^{18}$F]F-DTPA imaging protocol allowed the clear detection of as few as $8 × 10^3$ Jurkat$^{DTPA-R}$ or αCD19-CAR T$^{DTPA-R}$ cells in vivo. The reported sensitivity for DTPA-R is comparable to or even higher than that of other reporter systems[54]. In a comparative study with primary T cells transfected with human reporter genes[55], only the norepinephrine transporter with [$^{18}$F]F-MFBG was able to detect $3–4 × 10^4$ cells, while for HSV-TK/[$^{18}$F]F-FEAU and NIS/[$^{124}$I]-iodide a sensitivity of ~$3 × 10^5$ and ~$1 × 10^6$ was reported, respectively. In direct comparison with those reporter systems DTPA-R/[$^{18}$F]F-DTPA not only achieved a higher sensitivity but also shows a superior excretion profile, which leads to lower background signals[55].

In summary, the DTPA-R and [$^{18}$F]F-DTPA system meets all relevant design and functional requirements for a universal reporter gene system, which may boost future ATMPs by making them PET traceable.

# Methods

## Cloning of plasmids
Construction of plasmids, for example, for the production of retrovirus- and AAV-based vectors, was accomplished by standard cloning techniques. Sequence maps for DTPA-R and Colchi-R can be found in the Supplementary Information (pages 2 and 3, respectively). Schematic gene maps can be found in Figs. 1b, 4a,b and 7a. Genes were synthesized by Twist Bioscience or Eurofins Genomics. Correctness was confirmed by restriction digestion and by Sanger sequencing (Eurofins Genomics).

## Cell culture
Eucaryotic cells were cultured at 37 °C in a humidified 5% $CO_2$ atmosphere and were regularly tested by PCR for potential mycoplasma contamination. Cell lines from academic sources were authenticated by single-nucleotide polymorphism profiling (Multiplexion). Cell lines were cryo-preserved in recovery freezing medium (Gibco, Thermo Fisher Scientific) and stored in the vapour phase of liquid nitrogen. The Jurkat T cell line (obtained from Prof. Bernhard Küster, TU Munich (American Type Culture Collection (ATCC), TIB-152), identity confirmed by Multiplex human Cell line Authentication Test on 20 August 2023), Prostate carcinoma cell line PC3 (obtained from the ATCC; cat. no. CRL-1435) and Raji-GFP-fLuc and NALM6-GFP-fLuc cells (expressing green fluorescent protein (GFP) and firefly luciferase (fLuc); obtained from Prof. Stanley Riddell, Fred Hutchinson Cancer Center Seattle, ATCC: CCL-86 transduced with GFP-fLuc[56], identity confirmed by Multiplex human Cell line Authentication Test on 20 August 2023, and NALM6: RL-3273 and transduced with GFP-flLuc) were cultured in Roswell Park Memorial Institute (RPMI) 1640 medium with GlutaMAX supplement, 10% (v/v) foetal bovine serum (FBS) and 1% (v/v) penicillin–streptomycin (pen–strep) stock solution (10,000 U ml⁻¹, 10 mg ml⁻¹; all from Gibco). Human embryonic kidney (HEK293T) cells (obtained from Prof. Gil Westmeyer, TU Munich (Sigma-Aldrich: ECACC 12022001), identity confirmed by Multiplex human Cell line Authentication Test on 20 August 2023) and HEK293[CD19] cells (obtained from Prof. Stanley Riddell, ATCC: CRL-1573 transduced with CD19, identity confirmed by Multiplex human Cell line Authentication Test on 20 August 2023) were cultured in Dulbecco's modified Eagle medium with 10% FBS and 1% pen–strep. Adherent cells were detached by washing with Dulbecco's balanced salt solution without calcium, magnesium, or phenol red (DPBS) and incubated in 0.25% trypsin–EDTA (both from Gibco) for 5–10 min at 37 °C. Cells were sedimented by centrifugation at 300g. For primary CAR T cells, the RPMI medium with GlutaMAX was supplemented with 10% FBS and 5% (v/v) SC⁺-stock solution (stock concentration: 100 mM HEPES, 20% pen–strep, 1 g l⁻¹ gentamycin and 1 mM β-mercaptoethanol) and with 200 U ml⁻¹ interleukin-2 (IL-2; PreproTech) for expansion or 80 U ml⁻¹ for maintenance.

## Generation of AAV vectors
AAV vector production was performed as described previously[57]. In brief, HEK293T packaging cells were triple transfected using PEI MAX (Polysciences). Producer cells containing AAV9[DTPA-R] viral vectors were lysed by three freeze–thaw cycles, and subsequently, vectors were purified by iodixanol gradient centrifugation and a final SEC. Titering of AAV9 vectors was performed by a set of primers annealing in the inverted terminal repeats[58].

## Gene transfer by retroviral transduction and cell line creation
Human cell lines (Jurkat, HEK293T and PC3) stably expressing different reporter gene variants, and primary human T cells expressing a CAR expression cassette were generated by retroviral gene transfer. The sequence of the plasmid backbone is proprietary but largely similar to GeneBank MW079339.1. It contains 5' long terminal repeats, an intron and the coding region of the transgene (reporter gene variant or CAR-2A-reporter gene), followed by 3' long terminal repeats. These plasmids were used to transiently transfected RD114 cells using

the $CaCl_2$-precipitation method. After 48 h, the supernatant containing the viral particles was collected. For each experiment, PBMCs from a healthy donor (German Red Cross Blood Donor Service) were isolated via density gradient centrifugation using Pancoll human (PanBiotech; density 1.077 g ml⁻¹) and activated with 360 U ml⁻¹ IL-2 (PeproTech) and 2.25 µl ml⁻¹ CD3/CD28 Expamer[59] (Juno Therapeutics, BMS). After 48 h, human cell lines and PBMCs were transduced via spinoculation. Transduced cell lines were expanded for 14 days and subsequently stained with anti-V5-tag-AF488 (Alexa Fluor 488) antibody for fluorescence-activated cell sorting (FACS) using a FACSAria Fusion Sorter (Becton Dickinson). Transduced PBMCs were expanded for 14 days and, if applicable, stained with Streptavidin-efluor450 and either anti-human EGFR(t)-PE (clone AY13) or anti-V5-tag-PE (clone TCM5) for FACS using MoFlo Astrios Sorter (Beckman Coulter). For the cell lines Jurkat and HEK293T, the 10% highest-expressing clones were isolated; the cell line PC3 was sorted twice for the 10% highest-expressing clones. For CAR T cells, all positive clones were isolated by cell sorting. Transduction levels for Jurkat cells were at ~5% and for PBMCs at ~25%. The absence of viral vectors in the resulting cell line was confirmed by reverse transcription-PCR with appropriate primers or a HIV-1 p24 ELISA kit (XB-1000; XpressBio).

## Flow cytometry of cell culture samples
Surface marker expression was measured using flow cytometry. To this end, cells were stained with a 1:1,000 dilution of Zombie Violet viability stain (BioLegend), followed by washing with FACS buffer (5% FBS in DPBS). Subsequently, cells were stained with the following antibodies for 1 h on ice: anti-human CD3-AF488 clone HIT3a (1:20), CD4-AF488 clone OCT4 (1:20), CXCR3-AF488 clone G025H7 (1:20), CD69-APC clone FN50 (1:50) and streptavidin-FITC (1:400) for CAR detection (all BioLegend) and anti-V5-tag antibody clone SV5-Pk1 (Bio-Rad) conjugated to AF488-NHS (Lumiprobe) in house (3.1 µg ml⁻¹). After that, cells were washed three times and resuspended in 100 µl FACS buffer. Flow cytometry analysis was done on a LSR-Fortessa flow cytometer (Becton Dickinson) using excitation lasers at 405, 488, 561 and 640 nm and bandpass filters for FITC (530/30 nm), BV421 (450/40 nm), PE-Texas-Red (610/20 nm), APC (670/14 nm) and Alexa 700 (730/45 nm). For the quantification of the number of fluorescent molecules per cell, Quantum MESF kits (Bangs Laboratories) were used. Alexa Fluor 488 MESF beads (for AF488-labelled antibodies) or FITC-5 MESF beads (for Streptavidin-FITC) were analysed in the flow cytometer on the same day in FACS buffer. Results were analysed using FlowJo software (ver. 10.8.1; Becton Dickinson).

## Flow cytometry of mouse samples
Blood samples were collected from mice, and coagulation was prevented by adding 10 µl heparin (Heparin-Natrium-25,000; Ratiopharm) per 50 µl blood. Mice were euthanized, and tissue samples were directly collected and kept on ice in RPMI medium. Cells from mouse spleen were isolated by passing the dissected organ through a 70-µm cell strainer (Corning). For the analysis of CAR T cells found in hollow bones of mice, the joints were removed mechanically, and subsequently, the bone marrow was gently flushed out using a 30 G cannula attached to a 1-ml syringe filled with RPMI medium. For all samples, red blood cells were lysed by incubation in ammonium chloride (ACT) lysis buffer (0.17 M $NH_4Cl$ and 0.17 M Tris–HCl, pH 7.5) at room temperature. Upon incubation, the reaction was stopped by adding an equal amount of cold RPMI medium, and samples were centrifuged at 1,500 rpm and 4 °C. Spleen samples were incubated once in 5 ml ACT buffer for 5 min. Bone marrow samples were lysed in 3 ml ACT buffer for 3 min. Blood samples were lysed once in 10 ml for 10 min and a second time in 3 ml for 5 min in ACT buffer. Upon red blood cell lysis, the samples were resuspended in 100 µl FACS buffer (2% bovine serum albumin (BSA) in PBS), cell numbers were determined, and a maximum of $1 \times 10^7$ cells were used for antibody staining.

Cell suspensions prepared from tissues and organs were mixed with 10 µl of counting beads (Thermo Fisher Science 123count eBeads counting beads, with 1,009,000 eBeads ml$^{-1}$) to extrapolate cell numbers of the whole sample. Samples were washed and incubated in 1:400 diluted Fc-Block (BioLegend, purified anti-mouse CD16/32 clone 93) for 20 min on ice. After washing, samples were resuspended in antibody master mix and incubated for 20 min on ice in the dark. The antibody master mix contained the following antibodies: anti-human EGFR(t)-PE clone AY13 (1:2,000) or anti-V5-tag-PE clone TCM5 (1:500), anti-human CD3-APC clone UCHT1 (1:200), anti-human CD8-APC-efluor780 clone OKT8 (1:100), anti-human CD45-krome orange clone J33 (1:50) and streptavidin-efluor450 (1:50). For compensation, non-transduced PBMCs were stained with different anti-CD8-antibodies conjugated to fluorescent dyes: PE clone OKT8 (1:50), APC clone RPA-T8 (1:200), APC-efluor780 clone OKT8 (1:100), pacific orange clone 3B5 (1:50), efluor450 clone OKT8 (1:100). Upon incubation, cells were pelleted and resuspended in FACS buffer containing propidium iodide (1:100), centrifuged, resuspended, filtered through a 40-µm cell strainer (Corning) and washed in FACS buffer. Flow cytometry analysis was done on a CytoFLEX S flow cytometer (Beckman Coulter) using excitation lasers at 405, 488, 561 and 638 nm and bandpass filters for FITC (525/40 nm), BV421 (450/45 nm), PE-Texas-Red (610/20 nm), APC (660/20 nm) and Alexa 700 (780/60 nm). Results were analysed using FlowJo software (ver. 10.8.0; Becton Dickinson). A representative gating strategy for lymphocyte gating is shown in Extended Data Fig. 8c.

## Activation of T cells

For activation, T cells were seeded into six-well plates at a density of $0.5 \times 10^6$ cells ml$^{-1}$ in activation medium (cell culture medium containing 2.5 µg ml$^{-1}$ (3.3 nM) Ionomycin solved in 10% dimethyl sulfoxide (DMSO; BioGems) and 0.5 µg ml$^{-1}$ (0.8 nM) PMA (InvivoGen, solved in DMSO)) or control medium (cell culture medium with equal DMSO content (0.035% v/v)). After 24 h, the medium was removed, and cells were washed with DPBS and cultured in the same volume of cell culture medium for another 48 h before they were analysed by flow cytometry.

## CFSE proliferation assay

Proliferation was analysed using the CFSE cell division tracker kit (BioLegend) according to the manufacturer's instructions. In brief, $1 \times 10^7$ cells were washed with DPBS and resuspended in 333 µl CFSE working solution. After staining for 20 min at 37 °C in the dark, cells were washed once with medium and treated as stated in 'Activation of T cells' section. After 1 h ($t_0$), $0.5 \times 10^6$ cells of each cell line (non-activated) were fixed with neutral-buffered 4% formaldehyde solution (Otto Fischar) for 10 min, washed twice with DPBS and stored till the final evaluation in DPBS at 4 °C. After cultivation for 3 days, cells were stained with an anti-CD69-APC antibody (T cell activation marker), fixed as described above and analysed by flow cytometry.

The doubling time (see equation (1), where $d$ is the time difference between $t_0$ and $t_1$) was calculated for every individual cell after 3 days of culture using the fluorescence signal (FITC-A $t_1$) and the median fluorescence intensity (MFI $t_0$) from the reference population fixed after 1 h.

$$t = \frac{d}{\log_2\left(\frac{\text{MFI}t_0}{\text{FITC-}At_1}\right)}. \tag{1}$$

## Quantification of absolute receptor numbers by flow cytometry

The absolute number of receptors per cell was calculated using the Quantum MESF kit Alexa Fluor 488 (Bangs Laboratories). MESF beads allowed the correlation of fluorescent signal measured in the flow cytometer and known numbers of fluorophores in the reference particles. Median fluorescence of wild-type Jurkat cells or CAR T$^{\text{EGFRt}}$ cells (V5-tag negative) stained with the V5-antibody was subtracted from the

fluorescent signals measured, and calculation of the median MESF was done using the provided evaluation template. The degree of labelling (DOL) for each antibody was provided by BioLegend or, in the case of the anti-V5-antibody (clone SV5-Pk1), measured using a Nanophotometer NP80 (Implen). One antibody was assumed to bind two epitopes for calculating antibodies per cell. The calculation is shown in equation (2).

$$\text{Receptors/cell} = \frac{\text{MESF}}{\text{DOL}} \times 2. \tag{2}$$

## MACS

MACS was done using a MiniMACS starting kit with MS columns and anti-mouse IgG MicroBeads (both Miltenyi Biotec). Approximately $1 \times 10^7$ cells were labelled with 1 µg ml$^{-1}$ mouse anti-V5-tag antibody in 2 ml (clone SV5-Pk1, Bio-Rad) for 1 h at 4 °C and washed twice with FACS buffer. The MACS sorting was done according to the manufacturer's instructions, and purity was analysed via flow cytometry.

## Fluorescence microscopy

Black 96-well µ-plates (ibidi) were coated with 6.6 ng ml$^{-1}$ poly-D-lysine (Gibco) for 1 h at room temperature and washed with PBS three times before drying. PC3 or HEK293T cells were detached using trypsin–EDTA (0.25%), and 10,000 cells per well were seeded in 200 µl respective medium and cultured overnight. Cells were fixed with 4% paraformaldehyde (Sigma-Aldrich) in PBS for 10 min at 4 °C, and after two washing steps with PBS, blocking was done with PBS containing 3% BSA for 1 h at room temperature. Subsequently, the blocking buffer was replaced by primary anti-V5-tag antibody SV5-Pk1 (Bio-Rad) diluted 1:500 (1 mg ml$^{-1}$ stock solution) in PBS with 3% BSA and 0.1% Tween-20 (Carl Roth) in a volume of 100 µl. After 1 h at room temperature, the wells were washed with PBS, and the secondary antibody (1:20,000 F(ab')$_2$-fragment of rabbit-anti-mouse IgG conjugated to AF488; Invitrogen) in PBS with 3% BSA and 0.1% Tween-20 was added for another hour at room temperature. Cell nuclei were stained using Hoechst 33342 solution (1:20 in PBS; Thermo Fischer Scientific) for 10 min. Before the acquisition, the wells were washed again with PBS. Immunofluorescence microscopy images were acquired using an EVOS M7000 system using the DAPI (excitation (Ex) 357/44 nm, emission (Em) 447/60 nm), GFP (Ex 482/25 nm, Em 524/24 nm) and TexasRed (Ex 585/29 nm, Em 628/32 nm) fluorescent channels and an EVOS 40× plan fluor objective (AMEP 4699, all Thermo Fisher Scientific).

## Western blot analysis

Protein expression was analysed by semi-dry fluorescent western blot analysis. Jurkat cells ($1 \times 10^7$ cells) were collected and lysed in 500 µl RIPA-buffer (Thermo Fisher Scientific) containing protease inhibitors (1 tablet per 10 ml; cOmplete Mini Protease Inhibitor Cocktail, Roche) for 15 min under mild agitation on ice. After centrifugation at 13,200 rpm for 15 min, 400 µl of the supernatant was transferred in a fresh microcentrifuge tube and snap-frozen in liquid nitrogen.

Bradford assay (Pierce Coomassie Protein-Assay-Kit, Thermo Fisher Scientific) was used to quantify the total protein amount. For PNGase F (New England Biolabs) digestion 67.5 µg of cell lysates was denatured by incubation at 98 °C for 10 min in denaturing buffer (New England Biolabs). In a total of 20 µl, 2 µl GlycoBuffer 2, 2 µl of 10% NP-40 and 1 µl PNGase F (all from New England Biolabs) were added, and the reaction was incubated at 37 °C for 1 h. Samples were prepared by adding reducing Laemmli buffer followed by 5 min incubation at 95 °C, 2 min centrifugation at 13,000 rpm. The supernatant was stored at −20 °C. Five microlitres of Chameleon Duo prestained protein ladder (LI-COR) was used as the protein ladder. A 4–20% sodium dodecyl sulfate (SDS)–PAGE precast gel (GenScript Biotech) was loaded with 35 µg of protein for each cell line and separated at 120 V for 100 min. An Immobilon-P PVDF membrane (Merck Millipore) was activated in methanol for 5 min and equilibrated for at least 30 min in transfer buffer

**Table 1 | Primer pairs used for ddPCR**

| | Forward | Reverse | Probe | Provider | Catalogue number |
|---|---|---|---|---|---|
| **V5-CD4** | TCA ACC CAT GGC TCT GAT CG | CAC CGC ACG CAA AAG AAG AT | /56-FAM /TGG CGG AGT /ZEN/ TGC TGG ACT GC /3IABkFQ/ | IDT | N/A |
| **bGHpA** | GCC AGC CAT CTG TTG T | GGA GTG GCA CCT TCC A | /56-FAM /TCC CCC GTG /ZEN / CCT TCC TTG ACC /3IABkFQ/ | IDT | N/A |
| **RPP30** | AGA TTT GGA CCT GCG AGC G | GAG CGG CTG TCT CCA CAA GT | HEX -TTC TGA CCT GAA GGC TCT GCG CG-BHQ-1 | Microsynth | N/A |
| **DTPA** | AAC CGC GAG TAC TTC AGC AT | ACG ATG TGA TTC TCG GGC AG | /56-FAM /TCT CTG CTC /ZEN / GGC CGG ACC AA /3IABkFQ/ | IDT | N/A |
| **Tfrc** | N/A | N/A | Vic /TAMRA | Thermo | 4458366 |
| **18S** | N/A | N/A | FAM /MGB | Thermo | Hs99999901_s1 |

N/A, not applicable.

(25 mM Tris–HCl, 192 mM glycine, 0.1% SDS and 20% (v/v) MeOH, pH 8.3). After protein transfer in a semi-dry blotting chamber (Biometra) for 1 h at 300 mA, the membrane was washed with methanol and rinsed with water. After washing in TBS-T (10 mM Tris–HCl and 150 mM NaCl, pH 7.5 with 0.1% Tween-20) for 15 min, the membrane was blocked in TBS with 3% BSA for 1 h at room temperature under mild agitation. The primary anti-V5-tag antibody SV5-Pk1 (Bio-Rad) was diluted 1:2,000 in TBS-T with 3% BSA. For normalization, an anti-β-actin antibody conjugated to Dylight CW680 (clone AbD12141, Bio-Rad) was added at a dilution of 1:5,000. After incubating at 4 °C overnight with mild agitation, the membrane was washed three times with TBS-T for 15 min each at room temperature. The secondary IRDye 800CW-conjugated goat anti-mouse antibody (LI-COR) was diluted in TBS-T with 3% BSA and 0.1% SDS at a dilution of 1:20,000, and the membrane was incubated for 1 h at room temperature under mild agitation. After three washing steps in TBS-T, the fluorescence signals were detected using an Odyssey XF imaging system (LI-COR) in the 700-nm and 800-nm channels.

### DNA and RNA extraction from cells
Jurkat cells were washed with DPBS and counted. Aliquots containing $5 \times 10^6$ cells were pelleted and snap-frozen in liquid nitrogen after removing the supernatant. DNA was extracted using the DNeasy Blood & Tissue kit (#69506; Qiagen) according to the manufacturer's instructions. The optional RNase A digestion during proteinase K digestion was performed to eliminate RNA contaminants. DNA was eluted with 100 µl of buffer AE. RNA was extracted using RNeasy Plus Mini kit (74136; Qiagen) according to the manufacturer's instructions. Contaminating DNA was eliminated with the additional on-column DNase digestion step. RNA was eluted in 30 µl RNase-free water. Total DNA and RNA were quantified using a Nanodrop (Witec AG).

### RNA and DNA extraction from tissues
Tissue samples from mice were collected directly upon euthanasia, rinsed in water and pat-dried. Their weight was measured, and the samples were transferred into a cryovial and snap frozen in liquid nitrogen. Samples were stored at −80 °C. DNA was extracted from ~25 mg of tissue (except for spleen, where ~10 mg were used) with the DNeasy 96 Blood & Tissue kit (#69582; Qiagen) and according to the manufacturer's instructions. In brief, samples were transferred to Lysing Matrix D Tubes prefilled with 400 µl of 10% proteinase K in buffer ATL and mechanically fragmented by shaking with a Precellys Evolution Lyser (3 × 30 s at 7,600 rpm) before overnight incubation at 56 °C on a thermoshaker. Six microlitres of RNase A (100 mg ml$^{-1}$) was added to ensure RNA-free genomic DNA. The volume of buffer AL and ethanol was adapted according to the sample volume (~600 µl). The remaining steps were performed as per protocol. DNA was eluted with 100 µl of buffer AE. RNA was extracted from ~25 mg using the RNeasy Mini kit (#74106; Qiagen) with the additional on-column DNase

digestion step, according to the manufacturer's instructions. In brief, all tissues were transferred to Lysing Matrix D tubes prefilled with 600 µl of RLT buffer with 1% β-mercaptoethanol. Tubes were shaken with a Precellys Evolution Lyser (30 s at 7,600 rpm), cooled on ice and quickly centrifuged. Lysates were transferred in new Eppendorf tubes and centrifuged for 3 min at full speed. Supernatants were transferred in a new Eppendorf tube, and one volume (~600 µl) of 70% ethanol was added to the samples. The remaining steps were performed according to the instructions. RNA was eluted with 50 µl of RNase-free water (only 30 µl for heart and muscle). Total DNA and RNA were quantified using a Nanodrop nanophotometer (Witec AG).

### ddPCR for viral quantification
Viral genomes from human cells and murine tissues were quantified by ddPCR using, respectively, a probe-based assay for two sequence regions located in the V5-CD4 region of the reporter gene and the bovine growth hormone polyadenylation signal (bGHpA) regulatory element. Human ribonuclease P protein subunit p30 (RPP30, for cells) and murine transferrin receptor (Tfrc; for tissues) were used as reference genes (normalizer) for diploid genome calculation. Assay information is listed in Table 1.

The ddPCR was performed with a QX200 AutoDG Droplet Digital PCR system (Bio-Rad) according to the manufacturer's protocol. Each 25 µl ddPCR reaction contained 12.5 µl of 2× ddPCR SuperMix for probes (no dUTP; 1863024; Bio-Rad), 12.5 ng template DNA, 0.5× NotI-HF restriction enzyme (R3189L; New England Biolabs) and probe-based assays. Custom-made forward and reverse primers were used at a final concentration of 500 nM and labelled probes at 250 nM. Commercially available Tfrc was used 0.25×. Each target was run in duplicate. bGHpA and Tfrc were run in duplex. Reactions were prepared in a 96-well plate (12001925; Bio-Rad). After droplet generation on the AutoDG, the plate was sealed with a pierceable foil heat seal (1814040; Bio-Rad). PCR was performed with the following program: 95 °C for 10 min, 40 amplification cycles (94 °C for 30 s and 60 °C for 60 s), 10 min at 98 °C, followed by a cooling step at 4 °C until the plate was measured using a QX200 droplet reader (Bio-Rad). Thresholds were manually set for each sample by averaging the peaks for the positive and the negative droplets. Vector copy numbers (VCN) are calculated according to the formula VCN = 2 × vector copies/reference gene copies and reported as copies per cell.

### Reverse transcription quantitative PCR
cDNA was synthesized from 300 ng of total RNA using the High-Capacity cDNA Reverse Transcription kit (4368813; Thermo Fisher Scientific), including the optional RNase Inhibitor (N8080119; Thermo Fisher Scientific), in a final volume of 20 µl. Retrotranscription was performed for 10 min at 25 °C, followed by 120 min at 37 °C and 5 min at 85 °C. Quantitative PCR was performed using TaqMan Fast Advanced Master

Mix (4444557; Thermo Fisher Scientific) and 2 µl of cDNA per well in a final volume of 20 µl per well. Each target was analysed in triplicates. The standard program (50 °C for 2 min, 95 °C for 10 min, followed by 40 cycles at 95 °C for 15 s and 60 °C for 60 s) was performed on the 7900HT Fast Real-Time PCR system (Thermo Fisher Scientific). Custom-made probe-based assays for V5-CD4 and for DTPA-R were, respectively, used for transgene expression in cells and tissues (Table 1). A commercially available assay for 18S was used to normalize for total input RNA. For each target, the average ΔCt (Ct values for target − Ct values for the normalizer 18S) was used to calculate target expression in arbitrary units from $2^{-\Delta Ct}$.

## Chromium-51 release assay

The cytotoxic effector function of CAR T cells on the CD19 positive target cell line Raji-GFP-fLuc was investigated by the release of radio-active γ-emitting chromium-51. Target cells were labelled with 50 µCi $^{51}$Cr (sodium chromate in saline, $t_{1/2} = 27.71$ days, PerkinElmer) for 1 h at 37 °C in a humidified 5% $CO_2$ atmosphere. Cells were washed three times with medium, and 10,000 cells were plated in a 96-well V-bottom plate. Effector cells were added in different ratios in a final volume of 150 µl per well in triplicates. To assess spontaneous and maximum release, target cells were cultured in cell culture medium or medium with 2% SDS, respectively. After 4 h at 37 °C, the plate was centrifuged for 5 min at 300g at room temperature. The released $^{51}$Cr activity in the supernatant was quantified using a Wizard$^2$ automated gamma counter (PerkinElmer). Specific lysis was calculated using equation (3), and means were calculated for the triplicates.

$$\text{Specific lysis}(\%) = \frac{\text{release in sample} - \text{spontaneous release}}{\text{maximum release} - \text{spontaneous release}} \times 100. \tag{3}$$

## Real-time cell-killing assay

Target cell killing by αCD19-CAR T cells was analysed using an xCEL-Ligence real-time cell analysis system (ACEA Bioscience) according to the manufacturer's instructions. Transgenic HEK293T-CD19 target cells and HEK293T control cells were seeded into a 96-well glass electronic microplate (Agilent) in triplicates or quadruplicates (15,000 cells per well) for each condition. After a 24 h growth phase, different ratios of effector T cells were added to the respective wells. In addition, a control for target cell growth (with normal culture medium), and another control for total cell killing (4% SDS) were included. The growth of the target cells was measured continuously, and data were analysed using RTCA software pro (ver. 2.0.0.1301; ACEA Bioscience). Cell index was normalized to the last measurement before the addition of effector cells. Means of triplicates or quadruplicates for each experiment were calculated and are depicted in the figure.

## Chemical synthesis of radioligand precursors

Precursors for radiofluorination were prepared by standard chemical synthesis techniques (Supplementary Fig. 5); a detailed protocol of the optimized synthesis is currently in preparation. The non-radioactive precursors for [$^{18}$F]F-Nic-Glu$_2$-PEG$_4$-colchicine were prepared on 2-CTC-resin (100–200 mesh from Carbosynth), starting with Fmoc-NH-PEG$_4$-COOH (Iris Biotech) and Fmoc-D-Glu(OtBu)-OH followed by Boc-D-Glu(OtBu)-OH (both abcr) following standard solid-state synthesis protocols. D-Glu was used instead of the proteinogenic L-Glu building block to increase the stability against proteolytic cleavage in vivo. The reaction product was cleaved from the resin and conjugated to NH$_2$-CHX-A''-DTPA (Macrocyclics) or deacetylcolchicine, which was prepared from colchicine (BOC Sciences) according to Bagnato et al.[60]. The conjugation product was deprotected and conjugated to trimethylammonium-nicotinic-acid-tetrafluorophenol ester (as described by Zhou et al.[61]) yielding the final precursor molecules, which was purified by C18-HPLC. Quality control of the final

products included analytical HPLC as well as electrospray ionization–time-of-flight (ESI-TOF) mass spectrometry on a maXis mass spectrometer with an electrospray ionization source (Bruker Daltonics) and $^1$H/$^{13}$C-NMR on a 500 MHz NMR spectrometer (Bruker). The final product was lyophilized in individual portions, which were stored at −20 °C until use.

## Radiofluorination of [$^{18}$F]F-DTPA

For $^{18}$F radiosynthesis, the aqueous [$^{18}$F]F$^-$ produced via $^{18}$O(p,n)$^{18}$F reaction in a PETtrace 880 cyclotron (GE Healthcare) was passed through an anion-exchange Sep-Pak QMA carbonate Plus Light cartridge (46 mg; equilibrated with 10 ml deionized water ($_{di}$H$_2$O, Milli-Q; Waters). Radiolabelling was done manually or on a Modular-Lab Standard synthesis module (Eckert & Ziegler). In brief, the [$^{18}$F]F$^-$ anion (starting activity 3–6 GBq) was eluted with 700 µl 75 mM tetrabutylammonium hydroxide solution, and the water was evaporated at 95 °C for 5 min followed by azeotropic drying of [$^{18}$F]F$^-$ using anhydrous acetonitrile (0.001% H$_2$O max.; Merck Millipore) for 5 min twice. The labelling reaction was started by adding 0.5 mg TMA-Nic-D-Glu$_2$-PEG$_4$-CHX-A''-DTPA precursor, solved in 500 µl anhydrous DMSO (0.005% H$_2$O max.; VWR) to the reaction vial. After 10 min labelling at 95 °C, 2 ml $_{di}$H$_2$O was added, and the product was separated on a 250 × 10 mm C18 reversed-phase HPLC column (Multospher 100 RP 10-5µ; CS Chromatographie-Service) using an isocratic elution with 18% MeCN with 0.1% trifluoroacetic acid (TFA) flow or a 250 × 4.6 mm C18 reversed-phase HPLC column (ReproSil C18 Aq, 5 µm particle size; Dr A. Maisch, Ammerbuch, Germany) using an isocratic elution with 25% MeCN with 0.1% TFA as the mobile phase. The collected product was diluted with $_{di}$H$_2$O and passed through a preconditioned Sep-Pak C18 classic cartridge (Waters). The cartridge was flushed with 15 ml $_{di}$H$_2$O, and the product was eluted with 1 ml ethanol. To the product, 200 µl of a 0.15 M NH$_4$OAc buffer pH 5.5 with 20 mM terbium$^{III}$ chloride hexahydrate (AlfaAesar) and 200 µl $_{di}$H$_2$O were added, and complexation was completed at 55 °C for 15–30 min until the ethanol evaporated. After the labelling, 1 ml DPBS was added to precipitate the free terbium and the tube containing the product was briefly centrifuged in a tabletop centrifuge. The supernatant containing the [$^{18}$F]F-Nic-D-Glu$_2$-PEG$_4$-CHX-A''-DTPA•Tb radioligand ([$^{18}$F]F-DTPA) was used for further experiments.

## Determination of the log$D_{7.4}$

The octanol/PBS partitioning coefficient (log$D_{7.4}$) was determined by mixing 500 µl 1-octanol and radioligand (~0.5 MBq) in 500 µl PBS and shaking vigorously at 2,850 rpm for 5 min. After centrifugation at 13,000 rpm, a volume of 100 µl from both phases was transferred to a new tube, and the activity was measured in a Wizard$^2$ gamma counter (PerkinElmer). The log$D_{7.4}$ was calculated using equation (4).

$$\log D_{7.4} = \log \frac{\text{activity(1-octanol)}}{\text{activity(PBS)}}. \tag{4}$$

## HPLC

For preparative HPLC, a Shimadzu prominence system composed of two LC-20AP pumps, a SPD-M20A photodiode array detector and a CBM-20A communication module (all Shimadzu) was used. Separation was done using a ReproSil-Pur 120 C18-AQ column (250 × 30 mm, 5 µm particle size; Dr. Maisch). The liquid phase was water and acetonitrile (VWR) with 0.1% TFA.

Analytical HPLC was conducted using a system composed of two LC-30AD pumps, a SIL-30AC autosampler, a SPD-M20A photodiode array detector, a RF-20A fluorescence detector and a CTO-20AC column oven (all Shimadzu). Radioactivity was quantified using a Raytest GABI detector (Elysia-Raytest). The system is controlled using Chromeleon Chromatography Data System software version 6.80 (Dionex). For analysis, a Chromolith HighResolution RP-18e (100 × 4.6 mm; Merck kGaA) column was used.

## Binding assay

Cell binding assays were conducted with the respective radioligand and PC3 or Jurkat cells expressing DTPA-R-mRuby3, DTPA-R or Colchi-R, or not transduced wild-type controls. Binding assays were conducted in PBS with 2% bovine serum albumin (PBS$_{BSA}$). For binding studies with adherent PC3 cells, PBS$_{BSA}$ was complemented with Ca$^{2+}$ and Mg$^{2+}$ ions by adding 1/400 of a stock solution (10 g l$^{-1}$ CaCl$_2$ and 10 g l$^{-1}$ MgSO$_4$·6 H$_2$O) yielding PBS$_{BSA/Ca/Mg}$ to reduce undesired cell detachment during incubation and washing steps. At the end of the experiment, cell fractions were lysed in 1 M NaOH to obtain for each sample a solution with equal volume and homogeneous radioactivity distribution to ensure identical measurement geometries. Radioactivity was quantified in a Wizard$^2$ automated gamma counter (PerkinElmer) with appropriate detection windows.

To determine the IC$_{50}$ of CHX-A″-DTPA•$^{90}$Y, Jurkat$^{DTPA-R-mRuby3}$ cells were washed once with PBS and transferred to conical tubes (1 × 10$^6$ cells per tube). For competition, cells were incubated with increasing concentrations (each in quadruplicates) of not radioactive p-NH$_2$-Bn-CHX-A″-DTPA•$^{89}$Y for 1 h on ice (p-NH$_2$-Bn-CHX-A″-DTPA purchased from Macrocyclics). Subsequently, radioactive p-NH$_2$-Bn-CHX-A″-DTPA•$^{90}$Y ($^{90}$YCl$_3$ in 0.05 M HCl purchased from PerkinElmer) was added in a final concentration of 10 pM and incubated on ice for 1 h, and subsequently, cells were washed three times with PBS. From the cell-bound radioactivity, the IC$_{50}$ value was calculated by equation (5).

$$y = min + \frac{max - min}{1 + 10^{(\log(IC_{50} - x) \times slope)}}. \tag{5}$$

To determine the IC$_{50}$ of NH$_2$-CHX-A″-DTPA•$^{nat}$Tb and $^{19}$F-Nic-D-Glu$_2$-PEG$_4$-CHX-A″-DTPA•$^{nat}$Tb, 5,000 PC3$^{DTPA-R}$ and PC3$^{Colchi-R}$ cells were seeded in flat-bottom 96-well plates 2 days before the experiment. Cells were incubated with increasing concentrations (300 µl, each in quadruplicates) of not radioactive NH$_2$-CHX-A″-DTPA•$^{nat}$Tb and $^{19}$F-Nic-D-Glu$_2$-PEG$_4$-CHX-A″-DTPA•$^{nat}$Tb (each diluted in PBS$_{BSA/Ca/Mg}$) for 1 h at room temperature or 4 °C. Subsequently, 50 µl radioactive [$^{18}$F]F-DTPA•$^{nat}$Tb (0.5 MBq ml$^{-1}$ in PBS$_{BSA/Ca/Mg}$) was added and incubated for 1 h at room temperature or 4 °C. After four washing steps with PBS$_{BSA/Ca/Mg}$, cells were lysed with 1 M NaOH, and radioactivity was quantified using a Wizard$^2$ automated gamma counter (PerkinElmer). The values were fitted using a nonlinear fit function ([inhibitor] versus response with variable slope), and IC$_{50}$ values were calculated.

To study binding and internalization of the [$^{18}$F]F-DTPA radioligand, PC3$^{DTPA-R}$ and PC3$^{Colchi-R}$ cells were plated 2 days before the experiment onto six-well plates (0.4 × 10$^6$ cells per well). For competitive blocking, the cells were incubated in 1 ml of 100 µM NH$_2$-CHX-A″-DTPA•$^{nat}$Tb for 30 min at 37 °C or before the addition of the radioligand. The medium or NH$_2$-CHX-A″-DTPA•$^{nat}$Tb solution was removed, and cells were incubated with 0.5 MBq [$^{18}$F]F-DTPA in 1 ml PBS$_{BSA/Ca/Mg}$ for 15 s or 60 min at 4 °C or 37 °C as indicated (each in triplicates). After incubation, cells were washed twice with PBS$_{BSA/Ca/Mg}$ and twice with PBS. One triplicate of wells was lysed using NaOH, and the second triplicate was used to assess internalization. To separate the internalized and surface-bound fraction of [$^{18}$F]F-DTPA, cells were incubated with 1 ml Accutase (Gibco) for 30 min at 37 °C. Cells were subsequently centrifuged at 300g for 5 min and washed three times with PBS (cell-bound, non-cleavable activity), and all supernatant fractions were combined (surface bound, Accutase-cleavable activity).

## Kinetic affinity measurements on cells by RT-IC

RT-IC was performed on a heliX$^{cyto}$ instrument with heliX$^{cyto}$ M5 chips (both Dynamic Biosensors). The chip features five flow-permeable traps for cell-line-agnostic immobilization of individual cells on its measurement spot and an empty control spot in the same fluidic channel. Consecutive kinetic runs with three analyte concentrations each

were set up in automated assays in the heliOS (ver. 2024.1.0) software (Dynamic Biosensors). Excitation in the red channel was set to 0.6. Jurkat$^{DTPA-R}$ target cells were collected from culture and washed in calcium- and magnesium-free PBS and diluted to 3 × 10$^6$ cells ml$^{-1}$ before experiments. AlexaFluor647-PEG$_4$-[peptide]-CHX-A″-DTPA and AlexaFluor647-PEG$_4$-[thiourea]-CHX-A″-DTPA were synthesized and HPLC purified, and their concentration was determined by absorbance measurements using a Nanophotometer NP80. They were complexed with twofold molar excess of Tb(III)Cl$_3$·6H$_2$O (99.999%; Alfa Aesar) and diluted in Running Buffer 1 (RB 1: PBS + 0.01% Pluronic) to 0.2, 1 and 5 nM and placed along the cell sample into the temperature-controlled autosampler at 15 °C. The automated assay started by resuspending the cells in the autosampler and injecting 35 µl onto the measurement chip. The traps were filled with cells within seconds of injection and retained for subsequent fluidic steps. The kinetic measurement was performed at 25 °C by consecutive injections of increasing analyte concentrations at 25 µl min$^{-1}$ for 5 min each. Subsequently, dissociation was observed during continuous RB 1 flow at 50 µl min$^{-1}$ over the cells for 40–60 min. Finally, the chip was regenerated by reverse buffer flow that removes the trapped cells and prepares the chip for the next run. Data analysis was performed in heliOS (ver. 2024.1.0) software. Real-time background was automatically subtracted from data, and injection spikes were manually masked before fitting them when necessary. A maximum of three small signal drops in the continuous kinetic data during dissociation (due to cell fragmentation) were compensated. A global mono-exponential 1:1 binding kinetics fit model with free amplitudes (discontinuous) was applied to each dataset to extract $k_{on}$, $k_{off}$, $t_{1/2}$ and $K_D$.

## SPR spectroscopy

Real-time SPR spectroscopy was performed on a BIAcore X100 system (Cytiva) at 25 °C using HBS-T (20 mM HEPES–NaOH pH 7.5, 150 mM NaCl and 0.005% (v/v) Tween-20) as running buffer. The ectodomain of the DTPA-R (Anticalin-Avi-Strep) was produced in *Escherichia coli* co-transformed with an expression plasmid expressing the biotin ligase BirA. This ligand protein was immobilized (resonance units (ΔRU) ~240) on a streptavidin-functionalized (ΔRU ~2,200) sensor chip using the Biotin CAPture kit (Cytiva). Enhanced GFP conjugated to NCS-CHX-A″-DTPA and charged with $^{nat}$Tb was applied at 128 nM to the chip, and subsequently, the dissociation was measured for 60 min. Rate constants of association and dissociation were calculated from reference-corrected sensorgrams by fitting them to a global 1:1 Langmuir binding model.

## Stability assay of the radioligand

The stability of the [$^{18}$F]F-DTPA radioligand was tested by SEC using the analytical HPLC system described before. A Superdex 30 Increase (10/300 GL, Cytiva) column and PBS as the liquid phase at a 0.5 ml min$^{-1}$ flow rate was used. As a control for degraded radioligand, [$^{18}$F]F-DTPA was chemically hydrolysed in 6 M HCl at 90 °C for 1 h. NaOH was added to achieve a neutral pH before chromatography. Ten nanomoles of $^{19}$F-Glu$_2$-PEG$_4$-DTPA and NH$_2$-CHX-A″-DTPA were labelled with 1 MBq $^{177}$Lu in NH$_4$OAc buffer pH 5.5 for 30 min at room temperature and analysed by SEC. The produced radioligand and the urine from mice injected intravenously with [$^{18}$F]F-DTPA were directly used for analysis.

## Analysis of DTPA-R ectodomain shedding

Supernatant from in vitro cultivated PC3$^{DTPA-R}$ and PC3$^{Colchi-R}$ cells was concentrated ~10-fold using centrifugal filter units (molecular weight cut-off 10 kDa; Pall) at 4 °C and treated with 1× cOmplete Mini Protease Inhibitor Cocktail. Samples were analysed using SEC and western blot analysis. For chromatographic analysis, 200 µl of the cell culture supernatant was incubated with 1.7 MBq [$^{18}$F]F-DTPA for >1 h at room temperature to form complexes of the radioligand with potentially shedded DTPA-R ectodomains. The reaction solution was separated

by a Superdex 75 increase tricorn column (separation range 3–70 kDa; Cytiva) operated at 0.5 ml min⁻¹ using PBS as a mobile phase at the analytical HPLC. For the western blot, cell lysates were prepared by scraping PC3$^{DTPA-R}$ and PC3$^{Colchi-R}$ in PBS 2 mM EDTA from the cell culture flask. Cells were pelleted and lyses in RIPA buffer (500 µl per $1 \times 10^7$ cells) with cOmplete Mini Protease Inhibitor Cocktail. Samples were centrifuged at 13,200 rpm for 15 min, and the supernatant was used for sample preparation in reducing conditions. Samples from the supernatants were also prepared by addition of reducing Laemmli buffer and incubation at 95 °C. Ten microlitres of all samples were used for western blotting as described in 'Western blot analysis' section. The blot was stained using the anti-V5-tag antibody SV5-Pk1 (Bio-Rad) at a dilution of 1:2,000.

### In vivo models
Mice were purchased from Charles River Laboratories and were housed in a specific-pathogen-free environment in Sealsafe Next Greenline individually ventilated cages (Techniplast) at 45–60% humidity and 20–24 °C. Mouse strains for imaging experiments included C57BL/6 (C57BL/6NCrl, strain code 027), CD1-nude (Crl:CD1-$Foxn1^{nu}$, strain code 086) and NSG (NOD.Cg-Prkdc$^{SCID}$Il2rg$^{tm1Wjl}$/SzJ, strain code 614). Comparative mouse studies about the performance of CAR T cells in vivo were performed using NSG mice from an internal breeding colony. All animals were allowed a 1-week acclimatization period. Animal experiments were conducted in accordance with animal welfare regulations in Germany, with permission from the District Government of Upper Bavaria (approvals ROB-55.2-2532.Vet_216-15, Vet_21-127,Vet_02-21-41 and ROB- 55.2-2532.Vet_02-18-162) and in accordance with institutional guidelines. The animal protocol was reviewed by the commission defined by §15 of the German animal protection act and received approval. Humane endpoints were defined that included, among other criteria, loss of 10% body mass compared with the previous week. Results from animal experiments are trackable by unique institutional animal numbers (#xxx). Researchers were not blinded during the animal studies. Female animals were used for all experiments to decrease biological variation, except for the biodistribution study investigating the sex-specificity of [$^{18}$F]F-DTPA biodistribution. Typically, mice were between 6 and 10 weeks old when starting the experiments. Mice were kept under a 12-h day–night cycle and had access to ad libitum chow and water. The description within this publication follows the ARRIVE Guidelines for Reporting Animal Research[62].

Xenograft models of prostate carcinoma were engrafted by subcutaneous injection of $5 \times 10^6$ transgenic PC3 cells in a 1:1 mix with Matrigel Matrix (Phenol Red-Free; Corning) above the shoulder of CD1-nude mice. The maximum tumour size permitted with a mean diameter of 1.5 cm (equalling a 1,747 mm³ sphere) was not exceeded. [$^{18}$F]F-DTPA or [$^{18}$F]F-colchicine PET imaging was conducted around 20 days after tumour implantation when the tumours reached a diameter of 0.5–1 cm.

AAV9 viral vectors encoding an expression cassette for DTPA-R (AAV9$^{DTPA-R}$) were intravenously injected via the tail vein into CD1-nude or C57BL/6 mice. AAV doses used for experiments were inspired by the manufacturer's dosage guide for Zolgensma, where $1.1 \times 10^{14}$ vg kg⁻¹ are used clinically for i.v. injection into humans. For CD1-nude and C57BL/6 mice with a body weight of $24.3 \pm 1.1$ g at the beginning of the longitudinal AAV9-imaging cohort, this corresponds to a dose of ~$2.5 \times 10^{12}$ vg per mouse. In vivo transduction was imaged by [$^{18}$F]F-DTPA PET imaging either 7 days p.i. (comparison of different titres) or for the longitudinal evaluation on days 14, 21 and 28 p.i. On day 21, only four out of six animals were measured due to technical issues. For the AAV9 studies, animals were randomly assigned to the different cohorts, and no animals were excluded from the study. Organs from mice were collected and either fixed in neutral-buffered 4% formaldehyde solution for IHC or frozen for DNA and RNA extraction (see 'RNA and DNA extraction from tissues' section). Sectioning of paraffin blocks with kidneys and adrenals yielded only one central section of the adrenal out of $n = 3$

organs embedded (for correlation of IHC and PET). Outcome measures included imaging data and data from ex vivo analysis.

CAR T functional studies were done in NSG mice with systemic lymphoblastic leukaemia. A total of $0.5 \times 10^6$ NALM6-fLuc-GFP⁺ tumour cells were engrafted via the tail vein. After 7 days, $10 \times 10^6$ not sorted CAR T cells, expressing either the reporter protein DTPA-R or the EGFRt[32] or without transgene (mock), were intravenously injected. Bioluminescence imaging was done on days 0, 7 and 12 after CAR T administration by intraperitoneal injection of D-Luciferin-K-salt (150 mg kg⁻¹ body weight; PJK). Imaging was done 5 min after injection using the IVIS Lumina Imaging System (PerkinElmer, LAS), and signals were analysed by quantification of photons s⁻¹ cm⁻² sr⁻¹ with Living Image software (ver. 4.5).

CAR T cell distribution and proliferation was followed in NSG mice with a systemic B cell lymphoma. The tumour was engrafted by tail vein injection of $5 \times 10^5$ Raji-fLuc-GFP⁺ cells. All animals were included into the study and assigned randomly to the CAR T$^{DTPA-R}$ and CAR T$^{EGFRt}$ groups. After 7 days, $2 \times 10^6$ sorted CAR T cells, either expressing the reporter protein DTPA-R or as a control EGFRt[33], were intravenously injected. PET–MR imaging was done on days 4, 8, 15, 22 and 30 after CAR T cell administration (longitudinal cohort). Two animals reached the human endpoint, and the experiment was terminated at that timepoint. To correlate PET images and ex vivo analysis, mice were euthanized either directly after imaging (endpoint cohort on days 8 and 15) or one day later (longitudinal cohort on days 16 and 31). Bones (femur, tibia and humerus) and half of the spleen were used for flow cytometry analysis (see 'Flow cytometry of mouse samples' section) while the other organs were fixed in neutral-buffered 4% formaldehyde solution for IHC. Outcome measures included survival, imaging data and data from ex vivo analysis.

### PET–MR imaging
Animals were injected intravenously via the tail vein with 10–12 MBq of [$^{18}$F]F-DTPA or [$^{18}$F]F-colchicine diluted in DPBS and were either imaged dynamically over a 90 min period or kept awake for maximal renal excretion until static PET measurement started at $t = 90$ min. PET acquisition was conducted for 20 min using a nanoScan PET/MR system with 3 T field strength and two PET rings (Mediso Medical Imaging Solutions). The scanner was operated using the Nucline NanoScan software (ver. 3.04.025.0000; Mediso). For anatomical orientation, T1-weighted MR images were subsequently recorded using a 2D fast-spine-echo (FSE) sequence or a 3D gradient-recalled-echo (GRE) sequence with 0.25 mm isotropic resolution using a mouse body coil. Raw data were reconstructed using the Tera-Tomo 3D algorithm with normalization and correction for randoms, dead time and decay with no correction for attenuation or scatter, yielding a resolution of 0.7 mm. All reconstructed PET data were analysed using Inveon Research Workplace (ver. 4.2; Siemens Medical Solutions) by 3D isocontour set at 50% or 10% (for kidney signal) of the maximum intensity voxel, spheres with a diameter of 10–40 pixels or by 3D volume of interest (VOI) calculating %ID per gram or activity. PET signals caused by external contamination of the animal were not included in the evaluation (mouse #741).

### Detection limit and spot assay
The maximal signal capacity was determined using a phantom study for which $2 \times 10^5$ Jurkat cells were incubated with 1 MBq ml⁻¹ [$^{18}$F]F-DTPA in PBS$_{BSA}$ for 30 min, washed three times with PBS$_{BSA}$, counted and diluted to final concentrations. PET measurements were conducted for 60 min in a phantom consisting of two eight-well 0.2-ml PCR tube strips (VWR) filled with a volume of 10 µl each.

A spot assay was used to determine the number of DTPA-R⁺ T cells detectable if localized subcutaneously. Therefore 40 µl containing $4 \times 10^3$ up to $64 \times 10^3$ Jurkat$^{DTPA-R}$ cells or $4 \times 10^3$ up to $250 \times 10^3$ CAR T$^{DTPA-R}$ cells in DPBS were subcutaneously injected into CD1-nude mice at six different spots on the back of the animal. As a negative control, one

of the spots was injected with Jurkat cells not expressing the DTPA-R but Colchi-R with $64 \times 10^3$ or $250 \times 10^3$ cells, respectively. [$^{18}$F]F-DTPA was injected intravenously after the cell injection, and PET imaging was done after a 90-min excretion period for 20 min. A VOI for each spot was drawn, and the %ID was calculated without any background correction and assuming one millilitre equals one gram (equation (6)). The background in each animal was quantified by drawing five randomly placed spheres in reference tissue and calculating the mean for the obtained signals.

$$\%ID = \%ID_{mean} \text{ per ml (lesion)} \times \text{Volume(lesion (ml))}. \quad (6)$$

To calculate the detection limit in the CAR T model, a VOI was drawn over each hollow bone, including only the region used for ex vivo analysis (excluding signal originating from the joint heads), and the %ID per ml was calculated. Background signal in the extremities (forearms and legs separately) of the control mice (CAR T$^{EGFRt}$) was calculated by averaging the signal in the same regions over all six scans. This value was then subtracted individually for the front and hind legs from the signal in the mice with CAR T$^{DTPA-R}$ cells, and the %ID in the VOI was calculated (equation (7)).

$$\%ID_{cor} = [\%ID_{mean} \text{ per ml (lesion)} - \%ID_{mean} \text{ per ml (background)}]$$
$$\times \text{Volume(lesion (ml))}]. \quad (7)$$

The resulting signal was correlated with the cell number obtained by flow cytometry. The equation of this linear correlation was used to calculate the signals obtained from individual small lesions in animals treated with CAR$^{DTPA-R}$ to determine a detection limit in a therapeutic setting.

## Histology

Tissue samples from animal experiments were fixed in neutral-buffered 4% formaldehyde solution for 48 h at room temperature and subsequently transferred into PBS at 4 °C. Bones were decalcified in Osteosoft (Merck Millipore) for 4–20 days. Samples were dehydrated using an automated system (ASP300S; Leica Biosystems), followed by embedding in paraffin. Serial 2-μm sections were cut using a rotary microtome (HM355S; Thermo Fisher Scientific) and subjected to histological and immunohistological analysis. Haematoxylin and eosin staining was performed on deparaffinized sections with Eosin and Mayer's Haemalaun (Morphisto). IHC from formalin-fixed paraffin-embedded samples (IHC(P)) was performed using a Bond RXm system (Leica Biosystems) with primary antibodies against the V5-tag (clone SV5-Pk1; 1:500; Bio-Rad) and human CD19 (clone D4V4B; 1:600; Cell Signaling Technology). In brief, slides were deparaffinized using deparaffinization solution (Leica Biosystems), incubated with epitope retrieval solution 1 (corresponding to citrate buffer, pH 6) or with epitope retrieval solution 2 (corresponding to EDTA-based buffer, pH 9) for 30 min. The primary antibody was incubated at given dilutions for 15 min, and the bound antibody was detected with the polymer refine and/or refine red detection kit without post-primary reagent or with an intermediate rabbit anti-mouse secondary antibody (all Leica Biosystems). All IHC slides were counterstained using haematoxylin. Slides were scanned using an Aperio AT2 digital pathology slide scanner, and representative image regions were prepared using Aperio ImageScope (ver. 12.4) software (both Leica Biosystems). The positive cell fraction was analysed using QuPath[63] (ver. 0.3.2) software.

## Data analysis and figure preparation

The sample size (n) of experiments is given where appropriate. For in vitro experiments, distinct samples were measured and for no experiment samples were measured repeatedly. Statistical analysis and visualization was done using the Prism software (ver. 9.3.1; GraphPad). Error bars indicate standard deviation (s.d.) if not stated otherwise, and statistical significance was evaluated by unpaired Student's t-test

or analysis of variance (ANOVA). P values of <0.05 were considered as significant and are provided in the figures or figure legends if $P > 0.001$ (*$P < 0.05$, **$P < 0.01$, ***$P < 0.001$, ****$P < 0.0001$). Multivariate analysis of the AAV titre cohort was done by correlating the mean values of all variables (dose, vg and mRNA levels) for each group with the same dose with the mean PET signal obtained by multiple linear regression analysis. Image processing was done using Gimp (ver. 2.10.30; www.gimp.org). Chemical structures were drawn using ChemDraw (ver. 21.0.0). Protein structures were visualized using PyMol (ver. 2.5.2; Schrödinger). Figures were assembled using Inkscape (ver. 1.2.1; www.Inkscape.org).

## Statistics and reproducibility

Statistical analysis was done using the Prism software (ver. 9.3.1; Graph-Pad) as described in the figure legends. Comparisons were performed using unpaired two-tailed Student's t-test for two groups or using one-way or two-way ANOVA for multiple groups. Correlation of two variables was evaluated using linear regression. Correlation of multiple variables was done by multivariate analysis. Kaplan–Meier survival data were analysed using the log-rank (Mantel–Cox) test. Corrections for multiple analysis are indicated in the figure captions.

## Reporting summary

Further information on research design is available in the Nature Portfolio Reporting Summary linked to this article.

## Data availability

The authors declare that the main data supporting the findings of this study are available within the Article and its Supplementary Information. The corresponding author will make raw data and step-by-step protocols available upon request. Source data are provided with this paper.

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

## Acknowledgements

We thank M. Herz for producing F-18; M. Mittelhäuser, H. Rolbieski, S. Reder and N. Röder for PET–MR acquisition; F. Schilling and G. Topping for help with MR sequence optimization; O. Seelbach and M. Mielke for histology; A. Wolf for production of AAV9 vectors; R. Mishra for FACS sorting; A. Eichinger for help with the generation of the structural model; and B. Küster and G. Westmeyer for cell lines (all from TU Munich). Furthermore, we thank L. Bontadelli for quantitative PCR and ddPCR analysis; F. McBlane and M. Gutknecht for their help with in silico immunogenicity assessment (all from Novartis); and R. Strasser, N. Matscheko and A. Kratzert (all from Dynamic Biosensors) for making RT-IC affinity measurements available. V.M., K.S., D.H.B., A.S. and W.A.W. disclose support for the research described in this study from the National Centre for the 3Rs via CRACK IT Challenge 32 (grant nos. NC/C01905/1 and NC/C019202/1). GlaxoSmithKline and Novartis supported the project with in-kind contributions in the frame of the CRACK IT programme of the NC3Rs. The work was further supported by SFB-TRR 338/1 2021- 452881907 (project A01 (D.H.B.)) and the Bavarian Cancer Research Center (lighthouse project theranostics).

## Author contributions

V.M., K.F., D.H.B., A.S. and W.A.W. conceived the study and designed the experiments. V.M., K.F., L.W., M.A., S.D., M.Ž., L.K., P.B., T.B., S.R., S.L. and K.S. conducted the experiments and collected and analysed the data. S.K., C.K., M.S., D.H.B. and A.S. contributed experimental or analysis tools. V.M., K.F., A.S. and W.A.W. wrote the manuscript. All authors carefully reviewed and approved the manuscript.

## Funding

## Competing interests

V.M., K.F., A.S. and W.A.W. have applied for intellectual property rights on Anticalin-based reporter genes (WO 20022/101492 A1). A.S. is a co-founder and shareholder of Pieris Pharmaceuticals. The other authors declare no competing interests.

## Additional information

**Extended data** is available for this paper at https://doi.org/10.1038/s41551-025-01415-7.

**Correspondence and requests for materials** should be addressed to Wolfgang A. Weber.

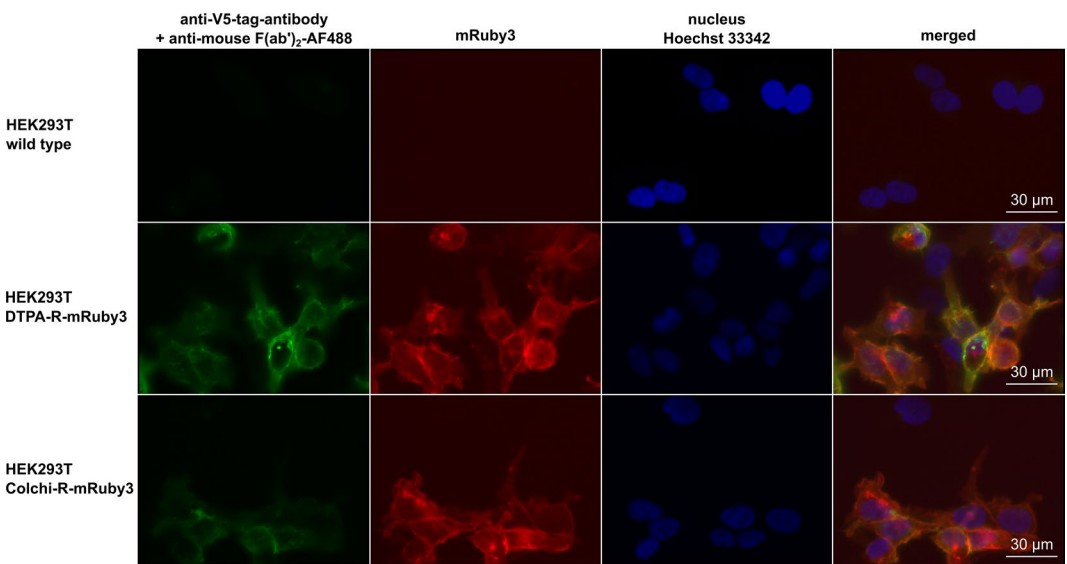

**Extended Data Fig. 1 | Fluorescence microscopy of HEK293T^DTPA-R-mRuby3 and HEK293T^Colchi-R-mRuby3.** HEK293T cells were transduced with retroviral vectors encoding the DTPA-R-mRuby3 or Colchi-R-mRuby3 reporter genes. Cells were stained with an anti-V5-tag antibody and an AlexaFluor488-conjugated secondary antibody and Hoechst 33342 and subsequently analysed by fluorescence microscopy.

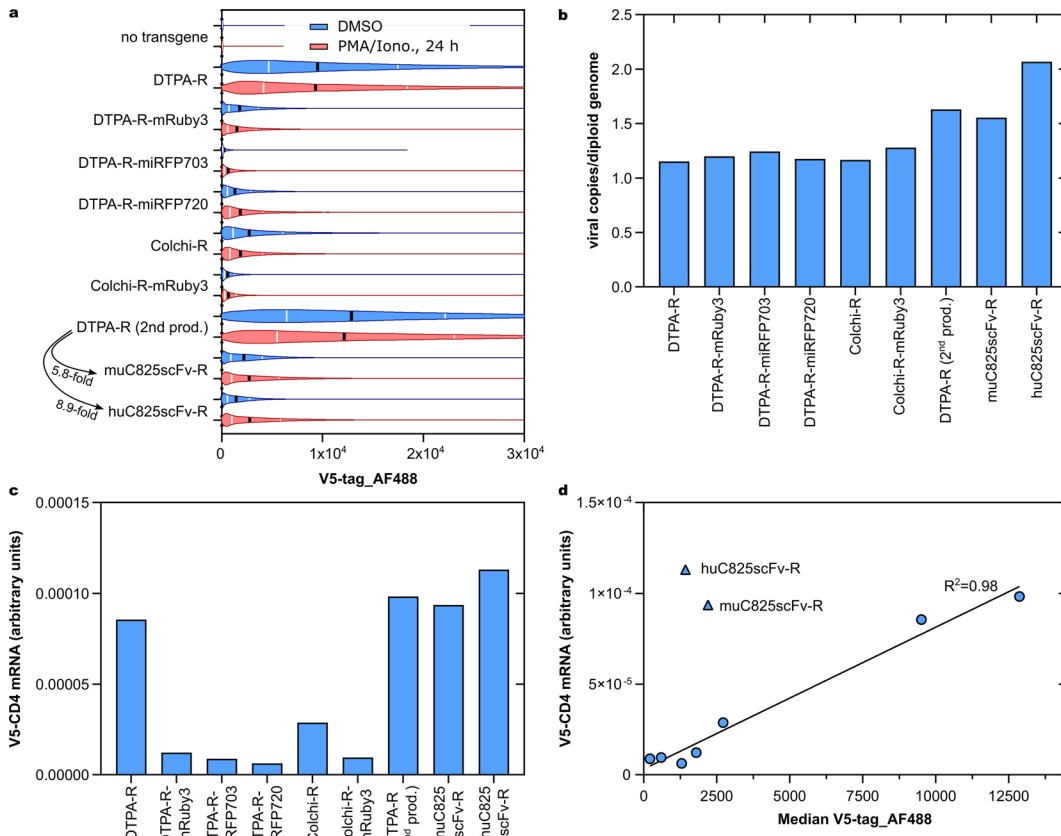

**Extended Data Fig. 2 | Expression analysis of anticalin vs. C825 scFv-based reporter proteins. a**, Jurkat cell lines transduced with reporter proteins containing the anticalins binding CHX-A''-DTPA•metal or colchicine, or the single chain variable fragment (scFv) binding DOTA•metal complexes were stained with AlexaFluor488-conjugated SV5-Pk1 anti-V5-tag antibody. Subsequently, the fluorescence was quantified by flow cytometry. The median is shown as black bars, quartiles as white bars, and 100,000 total events were analysed

each. **b**, Average number of genomic transgene insertions by retroviral gene transfer quantified by ddPCR. **c**, The C825 scFv-based reporter genes show similarly high mRNA levels in qPCR comparable to DTPA-R. **d**, While a linear correlation between mRNA level and surface-accessible protein levels was found for anticalin-based reporters (dots), the C825 scFv-based reporters showed low reporter protein densities on the cell surface (triangles), hinting towards a superior folding and/or stability of DTPA-R compared to scFv-Rs.

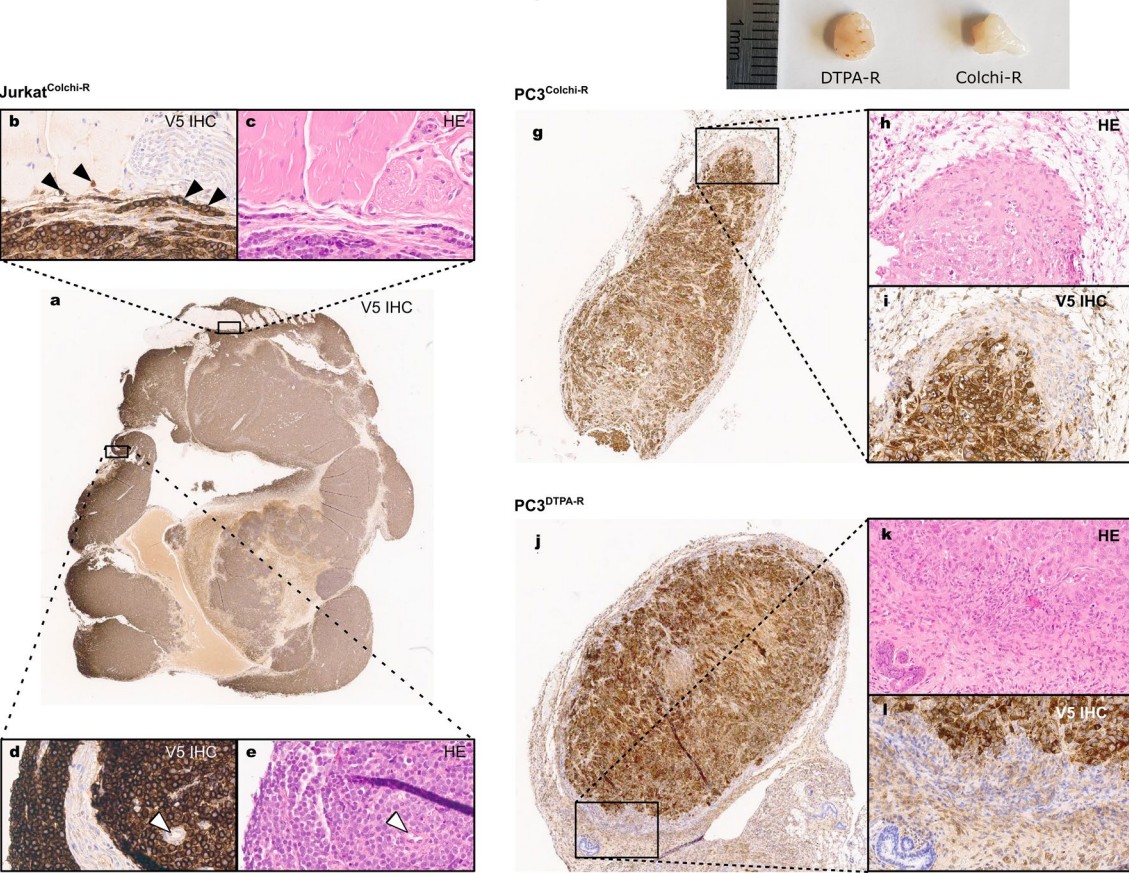

**Extended Data Fig. 3 | Immunohistochemistry of subcutaneous xenograft tumours using a V5-tag antibody.** Jurkat^Colchi-R (**a–e**), PC3^Colchi-R (**g–i**), and PC3^DTPA-R (**j–l**) cells were used to inoculate subcutaneous xenograft tumours, and after sacrificing the animals, the tumour tissue was subjected to immunohistochemical analysis. Consecutive sections were stained with anti-V5-tag Immunohistochemistry (**b,d,i,l**) and haematoxylin and eosin (HE; **c,e,h,k**).

Horseradish peroxidase (HRP) results in a strong chromogenic membranous stain for both tumour cell lines and both reporter proteins investigated. For the Jurkat T cell line, single-cell infiltrating in the murine tissue are clearly identifiable (black arrowheads in **b**), and blood vessels are visible through the absence of brown precipitate (white arrowheads in **e,f**).

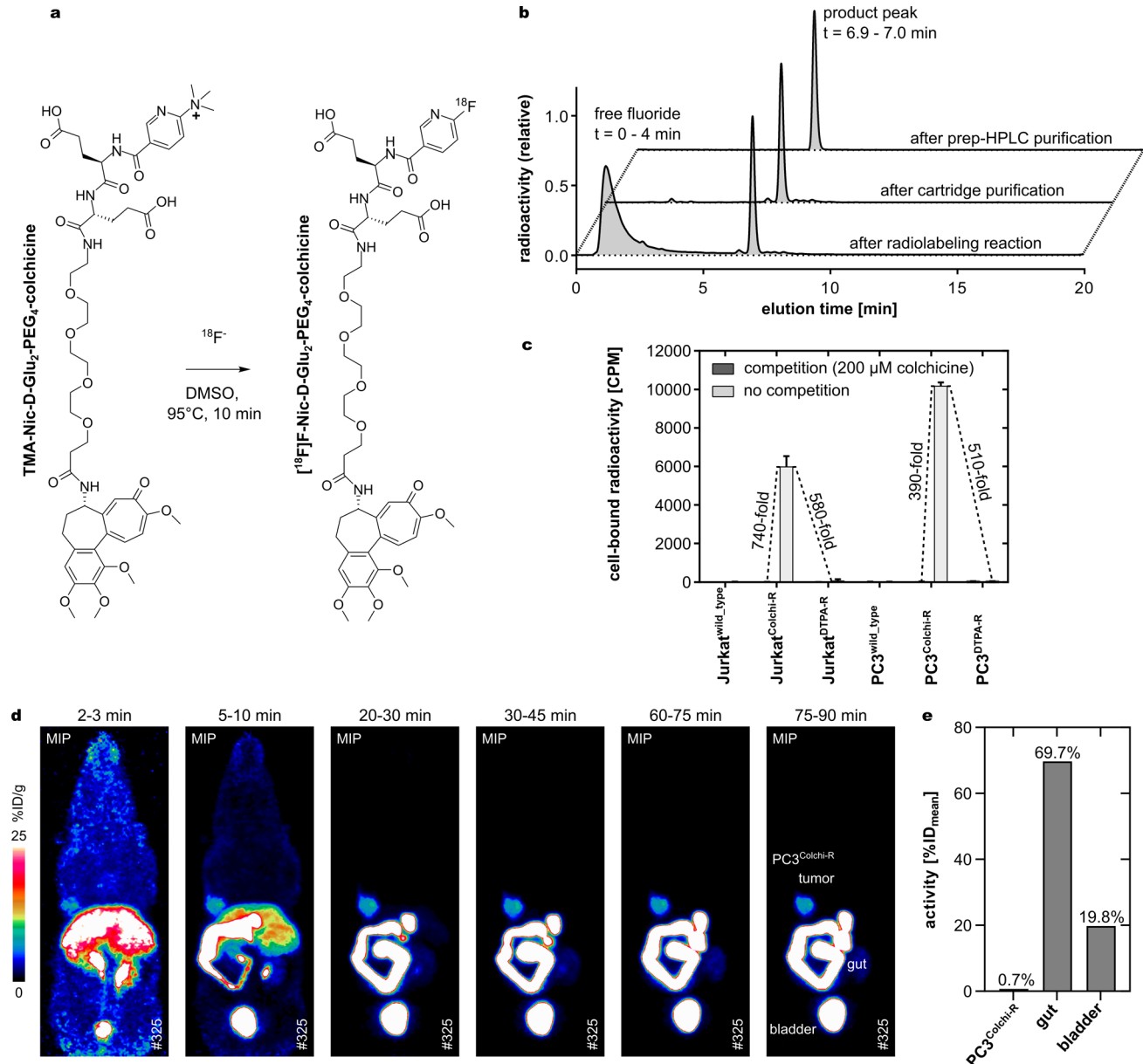

**Extended Data Fig. 4 | Preparation and analysis of [¹⁸F]F-Nic-Glu₂-PEG₄-colchicine. a**, The radioligand [¹⁸F]F-Nic-Glu₂-PEG₄-colchicine (=[¹⁸F]F-colchicine) was produced by direct radio-fluorination of the precursor molecule TMA-Nic-Glu₂-PEG4-colchicine, followed by PSH-H⁺ cartridge purification and subsequent preparative HPLC (250 × 10 mm C18 reversed-phase, isocratic), followed by a C18 cartridge purification. **b**, Quality control was performed by analytical HPLC using a Chromolith C18 reversed-phase column with a gradient elution (5–55% MeCN, 0.1% TFA). In the reaction product, free fluorine and a single fluorinated product were visible. Cartridge purification on PSH-matrix removed free fluorine, and the HPLC further increased purity by isolating the [¹⁸F]F-colchicine radioligand from minor impurities. **c**, Binding studies of [¹⁸F]F-colchicine were conducted on untransduced PC3 cells and sub-cell lines expressing Colchi-R and DTPA-R. The binding of [¹⁸F]F-colchicine was highly specific for PC3^Colchi-R cells when

compared to the PC3^DTPA-R control cells (510-fold), and the binding to the Colchi-R reporter protein on PC3 cells could be blocked by the previous addition of a high molar excess of 200 μM colchicine, which saturated binding sites (390-fold). **d,e**, Finally, the pharmacokinetic profile of [¹⁸F]F-colchicine was tested in a dynamic PET study. To this end, CD1-nude mice bearing a PC3^DTPA-R and a PC3^Colchi-R xenograft tumour above the shoulders, were injected i.v. with the radioligand. **d**, In the 2–3 min time frame [¹⁸F]F-colchicine is predominantly accumulated in the liver, indicating a mainly hepato-biliary excretion profile. **e**, VOI quantification of the last time frame of the dynamic PET scan (t = 75–90 min) revealed that 69.7 %ID was present in the gastrointestinal tract (gut), while only 0.7 %ID was taken up in the PC3^Colchi-R tumour and 19.8 %ID was excretion via the renal route and accumulated in the bladder.

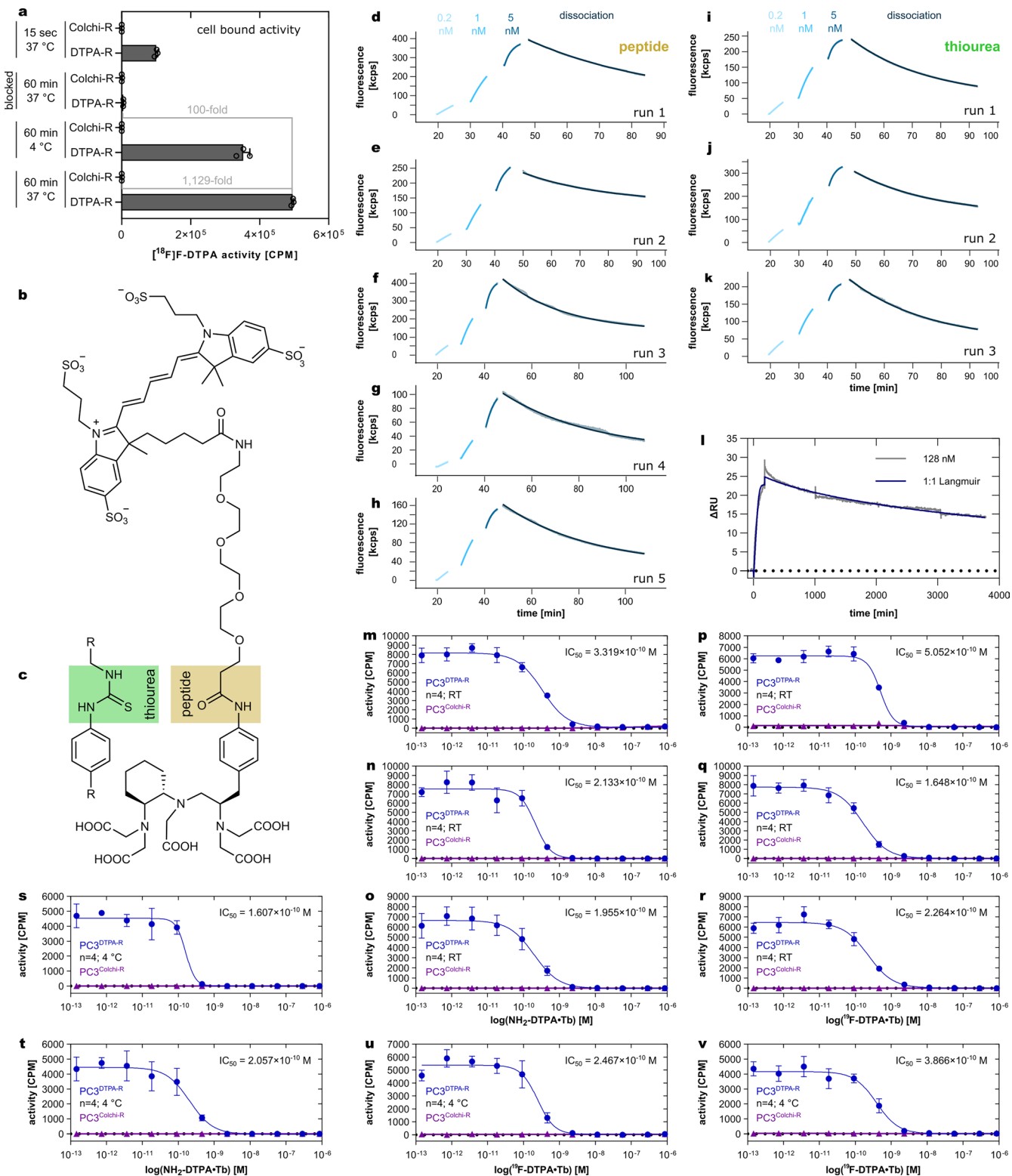

**Extended Data Fig. 5 | See next page for caption.**

**Extended Data Fig. 5 | Binding assays characterizing CHX-A"-DTPA•Tb binding to CL31d / DTPA-R. a**, Total cell binding of [$^{18}$F]F-DTPA to PC3$^{DTPA-R}$ cells was characterized by a binding assay comparable to the assay in Fig. 2d, but without the characterization of the subcellular localization of the radioligand. **b**–**k**, To assess the kinetic binding of CHX-A"-DTPA•$^{nat}$Tb based ligands to cells expressing DTPA-R, we employed real-time interaction cytometry (RT-IC), in which life cells are trapped in small cages within a microfluidic system and subsequently the association and dissociation of fluorescent ligands to these cells can be studied. **b**, To this end, two different ligands featuring a fluorescent dye were synthesized, purified, and charged with $^{nat}$Tb. One of the ligands comprises the dye, PEG$_4$, and CHX-DTPA linked by a peptide bond (AlexaFluor647-PEG$_4$-CHX-A"-DTPA) as it is the case for the [$^{18}$F]F-DTPA. However, on the other hand, we also studied (**c**) AlexaFluor647-PEG$_4$-thiourea-CHX-A"-DTPA, which resembles the situation when NCS-CHX-A"-DTPA is conjugated to a protein. Sensorgrams of repeated

RT-IC runs on target cells were recorded for the peptide (**d**–**h**) and the thiourea (**i**–**k**) compounds, each with three association concentrations of 0.2, 1, and 5 nM ligand concentration. **l**, Surface plasmon resonance (SPR) spectroscopy was employed as a complementary kinetic binding study. A recombinant DTPA-R ectodomain was produced in *E. coli*, purified, and immobilized on a CAP chip via a biotin specifically conjugated to a biotin acceptor peptide (Avi-tag). Enhanced green fluorescent protein (eGFP) conjugated to NCS-CHX-A"-DTPA was applied at 128 nM to the chip, and subsequently, the dissociation was measured for 60 min. **m**–**v**, Competitive binding assays (IC$_{50}$) were conducted for NH$_2$-CHX-A"-DTPA•$^{nat}$Tb (=NH$_2$-DTPA; **m**–**o**) and $^{19}$F-Glu$_2$-PEG$_4$-CHX-A"-DTPA•$^{nat}$Tb (≙ [$^{18}$F]F-DTPA; **p**–**r**) at room temperature and at 4 °C (**s**–**v**). Determined affinities are listed in Supplementary Table 3. **a,m-v**, All data points are means of triplicates (a) or quadruplicates (m-v) (technical repeats) with SD of one experiment.

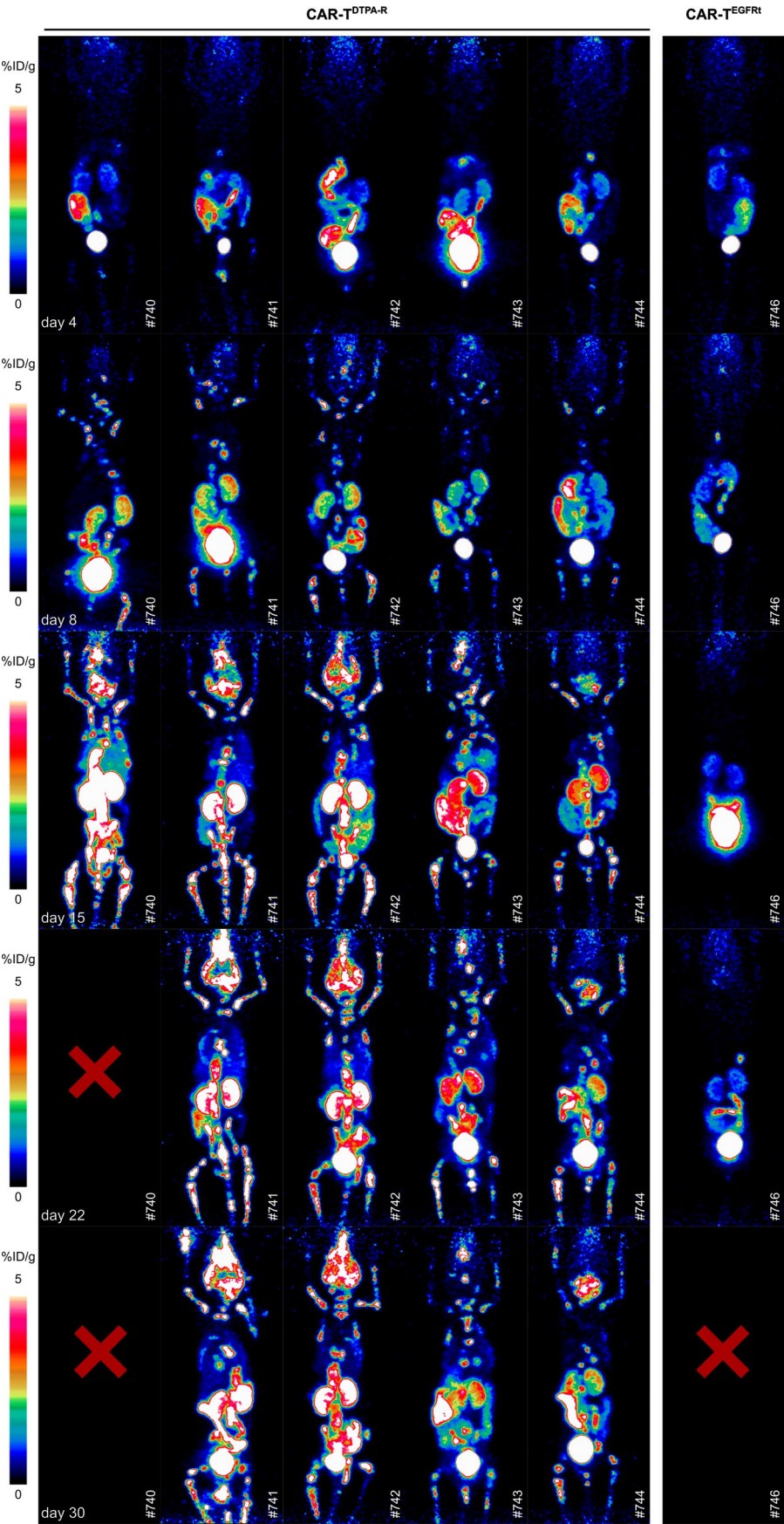

**Extended Data Fig. 6 | Complete cohort overview for longitudinal CAR T imaging.** A cohort of six NSG mice (♀) was injected i.v. with $5 \times 10^5$ Raji B cell lymphoma cells, which are known to home to the bone marrow. After 7 days, the mice were injected with $2 \times 10^6$ CAR-T$^{DTPA-R}$ cells (five animals) and one control animal with $2 \times 10^6$ CAR T$^{EGFRt}$ cells. To follow the proliferation and migration of the CAR-T$^{DTPA-R}$ cells, an [$^{18}$F]F-DTPA PET scan was recorded for 20 min on days 4, 8, 15, 22, and 30 after CAR-T cell injection.

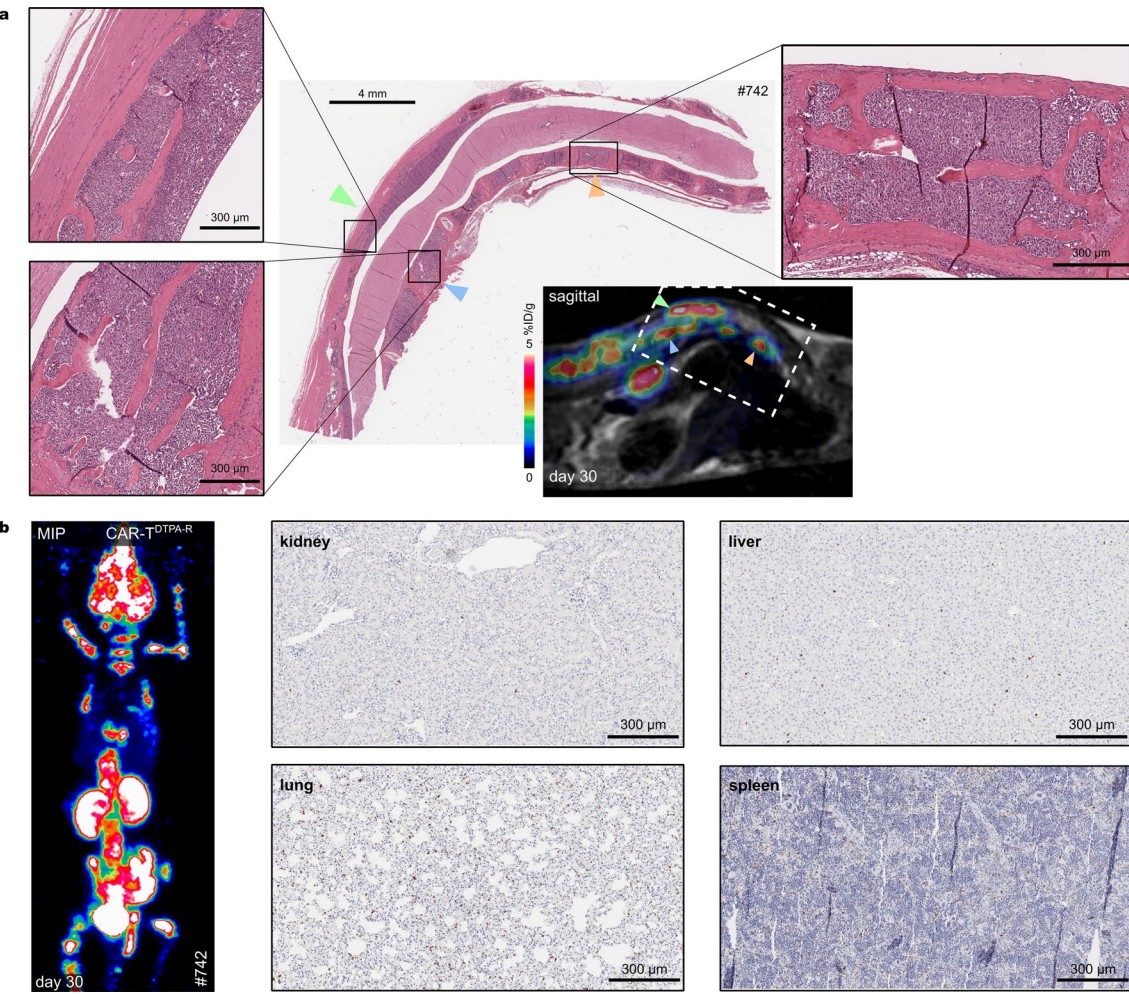

**Extended Data Fig. 7 | Histological analysis of a mouse from the longitudinal CAR-T study.** Additional histological analysis of mouse #742 (longitudinal CAR-T cohort injected with anti-CD19-CAR-T$^{DTPA-R}$ cells, see Fig. 9). **a**, HE stained section of the sagittal spine section with relevant regions magnified. **b**, MIP image at the end of the study (day 30). IHC detects V5-tag-positive cells (DTPA-R) in the lung and spleen but only a minor accumulation of CAR-T$^{DTPA-R}$ cells in kidney and liver.

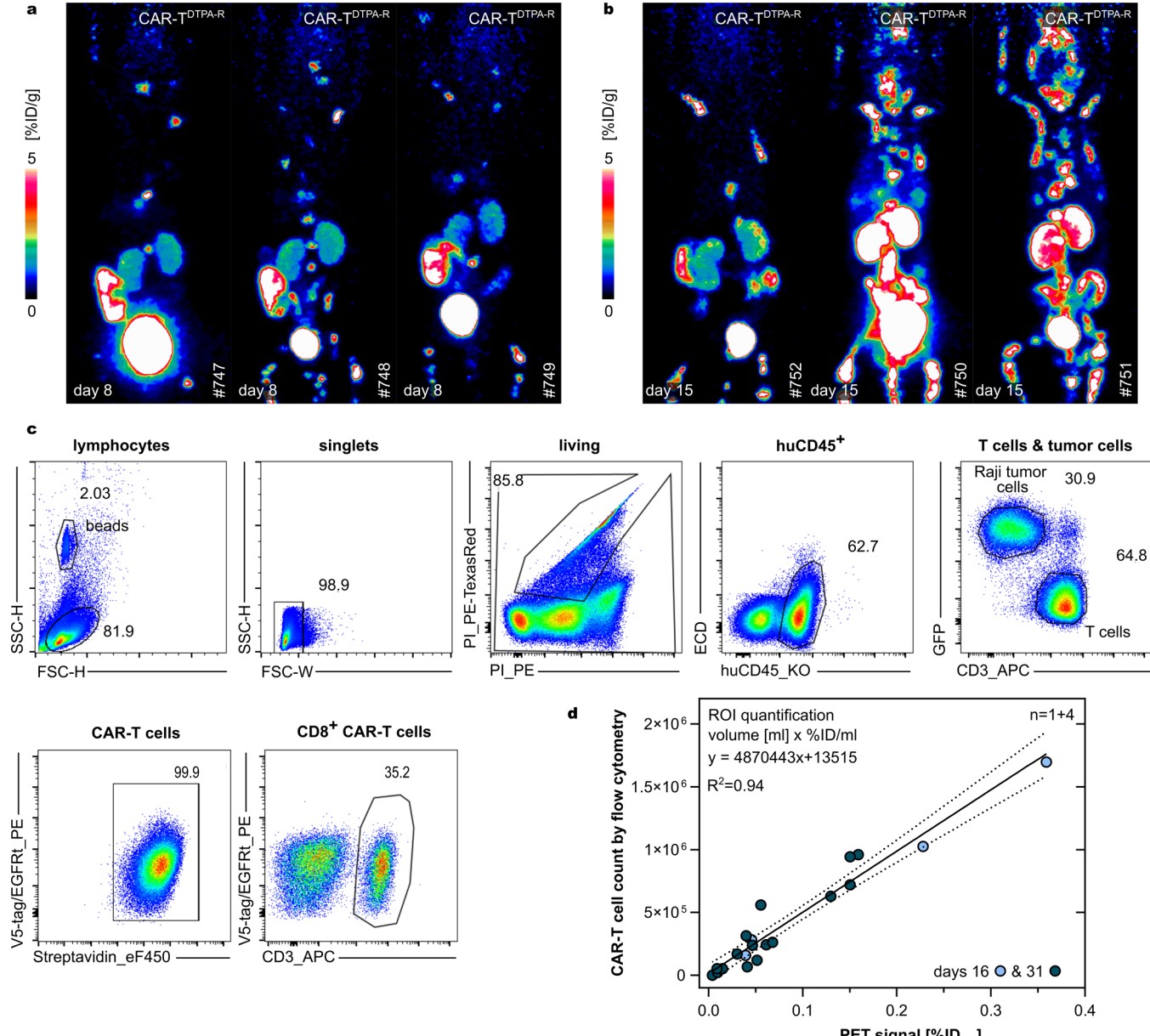

**Extended Data Fig. 8 | Cohort overview for CAR-T cell end-point study and flow cytometry analysis of mouse tissue samples.** Two cohorts of three NSG mice (♀) and one control mouse each were injected i.v. with $5 \times 10^5$ Raji B cell lymphoma cells, which are known to home to the bone marrow. After 7 days, the mice were injected with $2 \times 10^6$ αCD19-CAR-T$^{DTPA-R}$ or in case of the control mouse with $2 \times 10^6$ αCD19-CAR-T$^{EGFRt}$ cells. These cohorts conducted in parallel to the longitudinal CAR-T cell cohort allowed the *ex vivo* analysis of animals at a time point at which the longitudinal experiment was not completed yet. For this reason, (**a**) at day 8 or (**b**) day 15 after CAR-T cell injection, a [$^{18}$F]F-DTPA PET scan was recorded, and the animals were subsequently sacrificed. **c**, Flow cytometry gating of the bone marrow of the extremities of mice treated with αCD19-CAR-T cells (259,165 events analysed). **d**, Correlation of PET signal and number of CAR-T$^{DPTA-R}$ cells in the bone marrow of mice from the longitudinal cohort. Mice were sacrificed one day after the last PET imaging. Linear regression with 95% confidence interval.

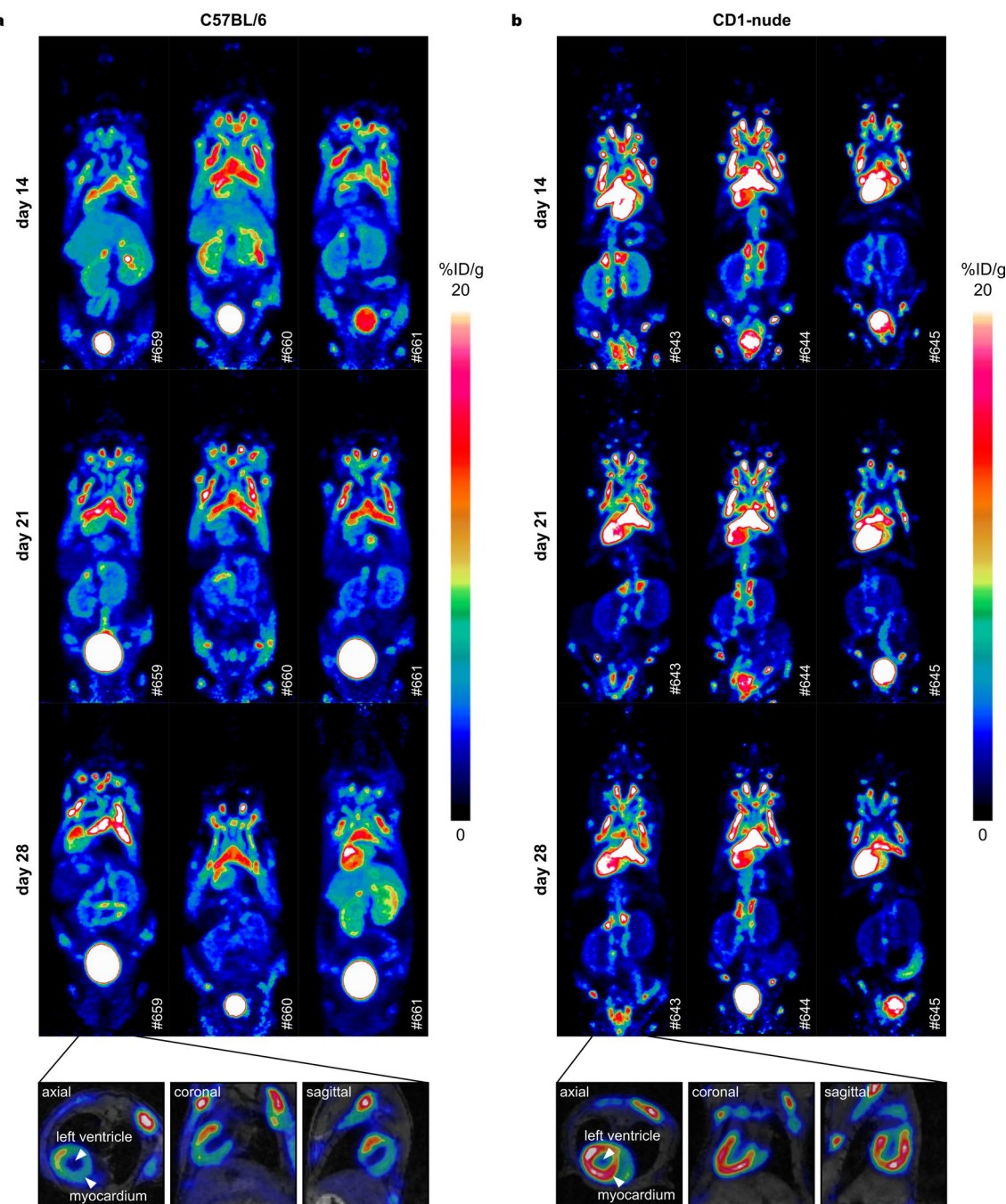

**Extended Data Fig. 9 | Complete cohort overview for AAV9 longitudinal study.** For the longitudinal AAV9 study, two cohorts of three ♀ C57BL/6 (**a**) and three CD1-nude (**b**) mice were injected i.v. with $2.5 \times 10^{12}$ vg/mouse. On days 14, 21, and 28 after AAV injection, an [$^{18}$F]F-DTPA PET scan was recorded for 20 min. For each mouse strain, cross-sectional images of the heart in three orientations are depicted, clearly showing the left ventricle.

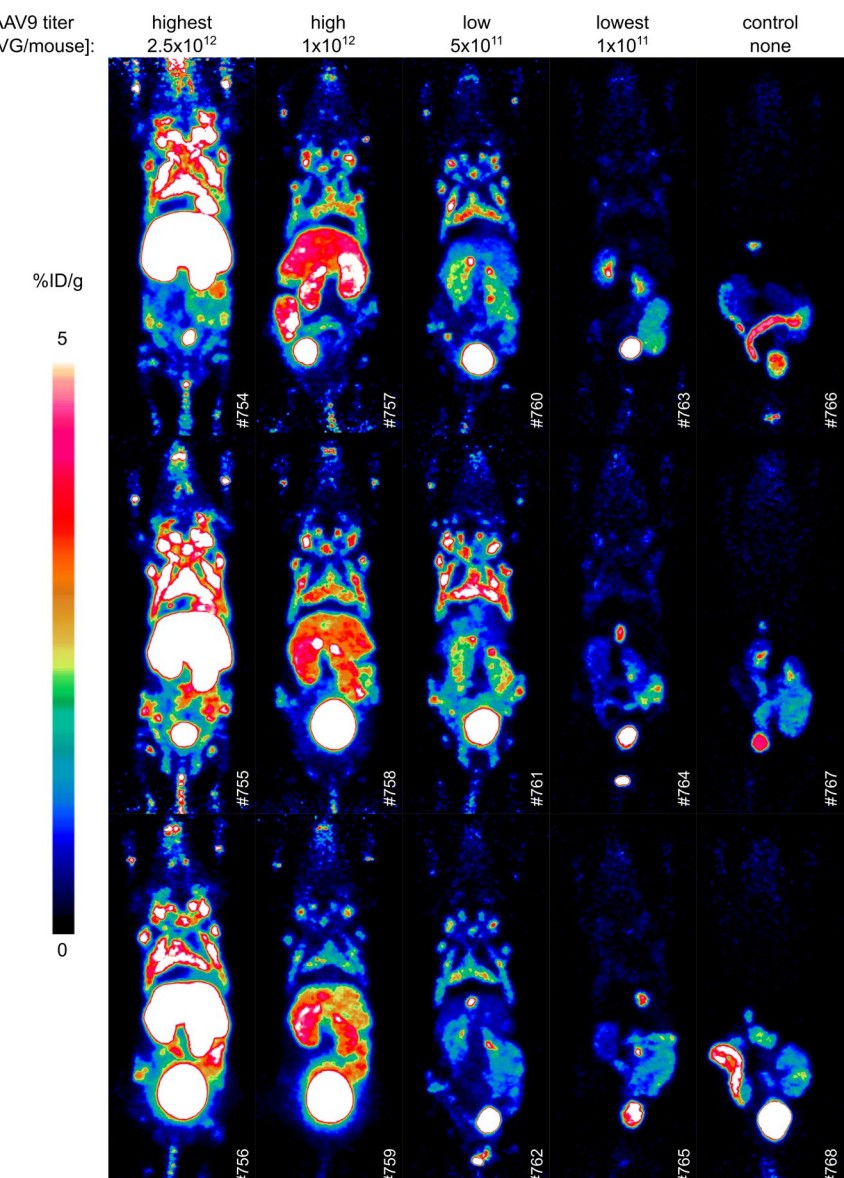

**Extended Data Fig. 10 | Complete cohort overview for AAV9 titre study.** Five cohorts of three ♀ C57BL/6 mice were injected i.v. with four different concentrations (highest: $2.5 \times 10^{12}$, high: $1 \times 10^{12}$, low: $5 \times 10^{11}$; lowest: $1 \times 10^{11}$ vg/mouse) or with PBS. On day 7, after AAV injection, an [$^{18}$F]F-DTPA PET scan was recorded for 20 min.

# Reporting Summary

## Statistics

For all statistical analyses, confirm that the following items are present in the figure legend, table legend, main text, or Methods section.

| n/a | Confirmed | |
|---|---|---|
| ☐ | ☒ | The exact sample size (*n*) for each experimental group/condition, given as a discrete number and unit of measurement |
| ☐ | ☒ | A statement on whether measurements were taken from distinct samples or whether the same sample was measured repeatedly |
| ☐ | ☒ | The statistical test(s) used AND whether they are one- or two-sided *Only common tests should be described solely by name; describe more complex techniques in the Methods section.* |
| ☒ | ☐ | A description of all covariates tested |
| ☐ | ☒ | A description of any assumptions or corrections, such as tests of normality and adjustment for multiple comparisons |
| ☐ | ☒ | A full description of the statistical parameters including central tendency (e.g. means) or other basic estimates (e.g. regression coefficient) AND variation (e.g. standard deviation) or associated estimates of uncertainty (e.g. confidence intervals) |
| ☐ | ☒ | For null hypothesis testing, the test statistic (e.g. *F*, *t*, *r*) with confidence intervals, effect sizes, degrees of freedom and *P* value noted *Give P values as exact values whenever suitable.* |
| ☒ | ☐ | For Bayesian analysis, information on the choice of priors and Markov chain Monte Carlo settings |
| ☒ | ☐ | For hierarchical and complex designs, identification of the appropriate level for tests and full reporting of outcomes |
| ☒ | ☐ | Estimates of effect sizes (e.g. Cohen's *d*, Pearson's *r*), indicating how they were calculated |

*Our web collection on statistics for biologists contains articles on many of the points above.*

## Software and code

Policy information about availability of computer code

| Data collection | PET data were collected using a nanoScan PET/MR system with 3T field strength and two PET rings (Mediso Medical Imaging Solutions, Budapest, Hungary). The scanner was operated using the Nucline NanoScan software (ver. 3.04.025.0000). |
|---|---|
| Data analysis | Reconstructed PET data were analyzed using the Inveon Research Workplace software (ver. 4.2; Siemens Medical Solutions, Knoxville, TN) by 3D isocontour set at 50% of the maximum intensity voxel, spheres with a diameter of 10 to 40 pixels or by 3D volume of interest (VOI) calculating %ID/g or activity. |
| | Various data were analyzed and plotted using Prism software (ver. 9.3.1; GraphPad, San Diego, CA). |
| | Histology data were visualized using Aperio ImageScope software (ver. 12.4; Leica Biosystems). The positive cell fraction of IHC(P) results was analyzed using the QuPath software (ver. 0.3.2). |
| | Flow cytometry results were analyzed using the FlowJo software (ver. 10.8.1; Becton Dickinson). |
| | xCELLigence data were analyzed using the RTCA software pro (ver. 2.0.0.1301; ACEA Bioscience). |
| | Chemical structures were visualized using ChemDraw (ver. 21.0.0; PerkinElmer) |
| | Radiation dose was modelled using MIRDcell software (ver. 3.13; MIRDsoft) |

For manuscripts utilizing custom algorithms or software that are central to the research but not yet described in published literature, software must be made available to editors and reviewers. We strongly encourage code deposition in a community repository (e.g. GitHub). See the Nature Portfolio guidelines for submitting code & software for further information.

# Data

Policy information about availability of data

All manuscripts must include a data availability statement. This statement should provide the following information, where applicable:
- Accession codes, unique identifiers, or web links for publicly available datasets
- A description of any restrictions on data availability
- For clinical datasets or third party data, please ensure that the statement adheres to our policy

The authors declare that the main data supporting the findings of this study are available within the publication and its Supplementary Information files. The corresponding author will make raw data and step-by-step protocols available upon request. However, the source data for all figures is available.

# Research involving human participants, their data, or biological material

Policy information about studies with human participants or human data. See also policy information about sex, gender (identity/presentation), and sexual orientation and race, ethnicity and racism.

| | |
|---|---|
| Reporting on sex and gender | not applicable |
| Reporting on race, ethnicity, or other socially relevant groupings | not applicable |
| Population characteristics | not applicable |
| Recruitment | not applicable |
| Ethics oversight | not applicable |

Note that full information on the approval of the study protocol must also be provided in the manuscript.

# Field-specific reporting

Please select the one below that is the best fit for your research. If you are not sure, read the appropriate sections before making your selection.

☒ Life sciences  ☐ Behavioural & social sciences  ☐ Ecological, evolutionary & environmental sciences

For a reference copy of the document with all sections, see nature.com/documents/nr-reporting-summary-flat.pdf

# Life sciences study design

All studies must disclose on these points even when the disclosure is negative.

| | |
|---|---|
| Sample size | The sample size used depends on the particular experiment. No sample-size calculations were performed. For in vivo PET/MR imaging studies, the sample size was limited by the number of animals that could be measured in the scanner in one day. For the AAV9 studies, n=3 mice per cohort were used, which allowed the measurement of two cohorts per day. For the CAR-T study presented, n=6 mice were used for the longitudinal cohort and n=6 animals for the endpoint cohort. Both cohorts were shifted by one day to overcome the limiting PET/MR scan time while using the same batch of tumor and CAR-T cells for this study to ensure comparability. For the CAR-T cell treatment study comparing CAR-T cells expressing DTPA-R with those expressing EGFRt, the sample size (n=5 vs. 5 + 3 control) was chosen according to institutional protocols and was limited by the maximum number of animals allowed in the initial orientation studies according to the German animal welfare regulations.<br>Sample size for in vitro assays was chosen based on literature and on established institutional protocols. In vitro assays were usually performed in triplicates and reproduced in biological and/or technical replications, as indicated in the figure legends. For flow cytometry, typically, a number of 100,000 events was analyzed per stained sample. |
| Data exclusions | PET signals caused by external contamination of the animal were not included in the data evaluation (mouse #741). Apart from this, no animals were excluded. Data from in vitro assays were not excluded except for the border wells in the xCelligence killing assay (lane A and H and column 1 and 12). |
| Replication | In vitro experiments were mainly done in duplicates, triplicates or quadruplicates (technical replication) to control for experimental variation, and experiments were repeated to show the reproducibility of the findings (biological replication). Figure panels either show results from different experiments (e.g., Fig. 1f) or from a representative experiment (e.g., Fig. 1h or Fig. 2a).<br>Animal studies were conducted with a predefined number of animals as outlined in the approved animal experimentation license. According to the German animal protection act, double or repeated animal studies are not permitted (§ 8 Abs. 3 TierSchG). |
| Randomization | Animals were randomly assigned to treatment and control groups. |
| Blinding | For the preclinical experiments, investigators were not blinded to the study groups. Due to local genetic engineering regulations, it is |

| Blinding | necessary to unmistakably label GMOs and animals carrying them. Outcome measures were primarily imaging data, which was recorded by the scanner in an unbiased fashion, survival (only for CAR-T), and ex vivo analysis (ddPCR, IHC, biodistribution analysis). The outcome parameter survival was unsuspicious to be unbiased as animals reached the predefined humane endpoint by body weight loss, which is an objective, unbiased parameter. The ex vivo analysis (ddPCR, IHC) was evaluated in a blinded manner (collaborators from different institutions). Due to the kind of study and its outcome parameters, it is highly unlikely that a non-blinded researcher consciously or unconsciously influenced the study's outcome. Quantitative data analysis and validation controls were used, minimizing the risk of introducing bias through the absence of blinding. |
|---|---|

# Behavioural & social sciences study design

All studies must disclose on these points even when the disclosure is negative.

| Study description | Briefly describe the study type including whether data are quantitative, qualitative, or mixed-methods (e.g. qualitative cross-sectional, quantitative experimental, mixed-methods case study). |
|---|---|
| Research sample | State the research sample (e.g. Harvard university undergraduates, villagers in rural India) and provide relevant demographic information (e.g. age, sex) and indicate whether the sample is representative. Provide a rationale for the study sample chosen. For studies involving existing datasets, please describe the dataset and source. |
| Sampling strategy | Describe the sampling procedure (e.g. random, snowball, stratified, convenience). Describe the statistical methods that were used to predetermine sample size OR if no sample-size calculation was performed, describe how sample sizes were chosen and provide a rationale for why these sample sizes are sufficient. For qualitative data, please indicate whether data saturation was considered, and what criteria were used to decide that no further sampling was needed. |
| Data collection | Provide details about the data collection procedure, including the instruments or devices used to record the data (e.g. pen and paper, computer, eye tracker, video or audio equipment) whether anyone was present besides the participant(s) and the researcher, and whether the researcher was blind to experimental condition and/or the study hypothesis during data collection. |
| Timing | Indicate the start and stop dates of data collection. If there is a gap between collection periods, state the dates for each sample cohort. |
| Data exclusions | If no data were excluded from the analyses, state so OR if data were excluded, provide the exact number of exclusions and the rationale behind them, indicating whether exclusion criteria were pre-established. |
| Non-participation | State how many participants dropped out/declined participation and the reason(s) given OR provide response rate OR state that no participants dropped out/declined participation. |
| Randomization | If participants were not allocated into experimental groups, state so OR describe how participants were allocated to groups, and if allocation was not random, describe how covariates were controlled. |

# Ecological, evolutionary & environmental sciences study design

All studies must disclose on these points even when the disclosure is negative.

| Study description | Briefly describe the study. For quantitative data include treatment factors and interactions, design structure (e.g. factorial, nested, hierarchical), nature and number of experimental units and replicates. |
|---|---|
| Research sample | Describe the research sample (e.g. a group of tagged Passer domesticus, all Stenocereus thurberi within Organ Pipe Cactus National Monument), and provide a rationale for the sample choice. When relevant, describe the organism taxa, source, sex, age range and any manipulations. State what population the sample is meant to represent when applicable. For studies involving existing datasets, describe the data and its source. |
| Sampling strategy | Note the sampling procedure. Describe the statistical methods that were used to predetermine sample size OR if no sample-size calculation was performed, describe how sample sizes were chosen and provide a rationale for why these sample sizes are sufficient. |
| Data collection | Describe the data collection procedure, including who recorded the data and how. |
| Timing and spatial scale | Indicate the start and stop dates of data collection, noting the frequency and periodicity of sampling and providing a rationale for these choices. If there is a gap between collection periods, state the dates for each sample cohort. Specify the spatial scale from which the data are taken |
| Data exclusions | If no data were excluded from the analyses, state so OR if data were excluded, describe the exclusions and the rationale behind them, indicating whether exclusion criteria were pre-established. |
| Reproducibility | Describe the measures taken to verify the reproducibility of experimental findings. For each experiment, note whether any attempts to repeat the experiment failed OR state that all attempts to repeat the experiment were successful. |

| Randomization | *Describe how samples/organisms/participants were allocated into groups. If allocation was not random, describe how covariates were controlled. If this is not relevant to your study, explain why.* |
| --- | --- |
| Blinding | *Describe the extent of blinding used during data acquisition and analysis. If blinding was not possible, describe why OR explain why blinding was not relevant to your study.* |

Did the study involve field work?  ☐ Yes  ☒ No

# Field work, collection and transport

| Field conditions | *Describe the study conditions for field work, providing relevant parameters (e.g. temperature, rainfall).* |
| --- | --- |
| Location | *State the location of the sampling or experiment, providing relevant parameters (e.g. latitude and longitude, elevation, water depth).* |
| Access & import/export | *Describe the efforts you have made to access habitats and to collect and import/export your samples in a responsible manner and in compliance with local, national and international laws, noting any permits that were obtained (give the name of the issuing authority, the date of issue, and any identifying information).* |
| Disturbance | *Describe any disturbance caused by the study and how it was minimized.* |

# Reporting for specific materials, systems and methods

We require information from authors about some types of materials, experimental systems and methods used in many studies. Here, indicate whether each material, system or method listed is relevant to your study. If you are not sure if a list item applies to your research, read the appropriate section before selecting a response.

## Materials & experimental systems

| n/a | Involved in the study |
| --- | --- |
| ☐ | ☒ Antibodies |
| ☐ | ☒ Eukaryotic cell lines |
| ☒ | ☐ Palaeontology and archaeology |
| ☐ | ☒ Animals and other organisms |
| ☒ | ☐ Clinical data |
| ☒ | ☐ Dual use research of concern |
| ☒ | ☐ Plants |

## Methods

| n/a | Involved in the study |
| --- | --- |
| ☒ | ☐ ChIP-seq |
| ☐ | ☒ Flow cytometry |
| ☒ | ☐ MRI-based neuroimaging |

# Antibodies

| Antibodies used | Antibodies and other reagents used for cell staining are listed in Supplementary Table 2. Protocols for their use can be found in the respective methods section.<br>Used antibodies:<br>anti-V5-tag (Bio-Rad,MCA1360, clone: SV5-Pk1, lot#: 150547, 148239, dilutions: IF: 1:500; FACS: 3.1 µg/ml; MACS: 1 µg/ml; IHC(P):1:500; WB: 1:2,000); anti-β-actin-Dylight CW680(Bio-Rad, discontinued, clone: AbD12141, lot#: 0114, dilution: 1:5,000); anti-mouse IgG-IRDye 800CW (LI-COR, 926-32210, lot#: C91210-09, dilution: 1:20,000); anti-mouse IgG [F(ab')2]-AF488 (Invitrogen, A21204, lot#: 2155587, dilution: 1:20,000); anti-human EGFR(t)-PE (BioLegend, 352904, clone: AY13, lot#: B336514, dilution: 1:2,000); anti-V5-tag-PE (Life Technologies, 12-6796-42, clone: TCM5, lot#: 2301156, dilution: 1:500); anti-human CD3-APC (Life Technologies, 17-0038-42, clone: UCHT1, lot#: 2376138, dilution: 1:200); anti-human CD8-APC-efluor780 (eBioscience, 47-0086-42, clone: OKT8, lot#: 2611767, dilution: 1:100); anti-human CD45-krome-orange (Beckman Coulter, B36294, clone: J33, lot#: 200097, dilution: 1:50); Streptavidin-efluor450 (Life Technologies, 48-4317-82, lot#: 2527387, dilution: 1:50); Streptavidin-FITC (Biolegend, 405201, lot#: B309228, dilution: 1:400); anti-CD8-PE (eBioscience, 12-0086-42, clone: OKT8, lot#: 2504398, dilution: 1:50); anti-CD8-APC (BioLegend,301049, clone: RPA-T8, lot#: B368721, dilution: 1:200); anti-CD8-APC.eflour780 (eBioscience, 47-0086-42, clone: OKT8, lot#: 2611767, dilution: 1:100); anti-CD8- pacific orange ~ KO (Life Technologies, MHCD0830, clone: 3B5, lot#: 2375611, dilution: 1:50); anti-CD8-efluor450 (eBioscience, 48-0086-42, clone: OKT8, lot#: 2410932, dilution: 1:100); anti-CD69-APC (BioLegend, 310910, clone: FN50, lot#: B337763, dilution: 1:50); anti-CD3-AF488 (BioLegend, 300319, clone: HIT3a, lot#: B278329, dilution: 1:20); anti-CD4-AF488 (BioLegend, 317419, clone: OKT4, lot#: B292040, dilution: 1:20); anti-CXCR3 / CD183-AF488 (BioLegend, 353709, clone: G025H7, lot#: B264198, dilution: 1:20); anti-human CD19 (Cell Signaling Technology,  90176S , clone: D4V4B, lot#: 1, dilution: 1:600) |
| --- | --- |
| Validation | The DTPA-R reporter gene comprises the V5-tag, which can be bound by the anti-V5-tag antibody SV5-Pk1. The ability of this antibody to detect the DTPA-R reporter protein and DTPA-R labeled cells has been demonstrated for various techniques, including immune fluorescence (Fig. 1e), IHC(P) (Fig. 4j), MACS (Fig. 1h), and FACS (Fig. 1f).<br>The FACS panel for CAR-T cell characterization as well as the anti-CD19 antibody for IHC(P) have been used in previous studies and were validated with appropriate samples and controls.<br>The following antibodies were tested by the vendors. |

Vendors tested for Flow cytometry, Immunofluorescence, Western Blot, Immunohistochemistry: anti-V5-tag (Bio-Rad)
Vendors tested for Western Blot: anti-β-actin (Bio-Rad) and anti-mouse IgG (LI-COR)
Vendors tested for Immunofluorescence: anti-mouse IgG [F(ab')2] (Invitrogen)
Vendors tested for Flow cytometry: anti-human EGFR(t) (BioLegend), anti-V5-tag (Life Technologies), anti-human CD3 (Life Technologies), anti-human CD8 (eBioscience), Streptavidin (Life Technologies), Streptavidin (Biolegend), anti-CD8 (eBioscience), anti-human CD45 (Beckman Coulter), anti-CD8 (BioLegend), anti-CD8 (eBioscience), anti-CD8 (Life Technologies), anti-CD8 (eBioscience), anti-CD69 (BioLegend), anti-CD3 (BioLegend), anti-CD4 (BioLegend), anti-CXCR3 / CD183 (BioLegend)
Vendors tested for Immunohistochemistry: anti-human CD19 (Cell Signaling Technology)

# Eukaryotic cell lines

Policy information about cell lines and Sex and Gender in Research

| | |
|---|---|
| Cell line source(s) | Jurkat T cell line obtained from Prof. Bernhard Küster, TU Munich (American Type Culture collection (ATCC), Manassas, VA; TIB-152), identity confirmed by Multiplex human Cell line Authentication Test on 20.08.2023).<br>Prostate carcinoma cell line PC3 (ATCC, catalogue number: CRL-1435).<br>Raji-GFP-fLuc cells expressing green fluorescent protein (GFP) and firefly luciferase (fLuc); obtained from Prof. Stanley Riddell, Fred Hutchinson Cancer Center Seattle, ATCC: CCL-86 transduced with GFP-fLuc (Hudecek, M. et al. 2015) identity confirmed by Multiplex human Cell line Authentication Test on 20.08.2023.<br>NALM6-GFP-fLuc cells obtained from Prof. Stanley Riddell, Fred Hutchinson Cancer Center Seattle, NALM6 ATCC: RL-3273 transduced with GFP-flLuc.<br>Human embryonic kidney (HEK293T) cells obtained from Prof. Gil Westmeyer, TU Munich (Sigma-Aldrich: ECACC 12022001) identity confirmed by Multiplex human Cell line Authentication Test on 20.08.2023.<br>HEK293CD19 cells (obtained from Prof. Stanley Riddell, ATCC_CRL-1573 transduced with CD19 identity confirmed by Multiplex human Cell line Authentication Test on 20.08.2023. |
| Authentication | Cell lines obtained from academic sources were authenticated by SNP profiling (Multiplexion, Heidelberg, Germany). The SNP profiles matched known profiles. |
| Mycoplasma contamination | Eucaryotic cells were regularly tested by PCR and found to be free of mycoplasma contamination. |
| Commonly misidentified lines (See ICLAC register) | The cell lines used were compared to the Register of Misidentified Cell Lines (ver. 12).<br>Only HEK is considered a misidentified cell line. The HEK293T and HEK293-CD19 cell lines have been authenticated by SNP profiling, and the profiles obtained matched. |

# Palaeontology and Archaeology

| | |
|---|---|
| Specimen provenance | *Provide provenance information for specimens and describe permits that were obtained for the work (including the name of the issuing authority, the date of issue, and any identifying information). Permits should encompass collection and, where applicable, export.* |
| Specimen deposition | *Indicate where the specimens have been deposited to permit free access by other researchers.* |
| Dating methods | *If new dates are provided, describe how they were obtained (e.g. collection, storage, sample pretreatment and measurement), where they were obtained (i.e. lab name), the calibration program and the protocol for quality assurance OR state that no new dates are provided.* |

☐ Tick this box to confirm that the raw and calibrated dates are available in the paper or in Supplementary Information.

| | |
|---|---|
| Ethics oversight | *Identify the organization(s) that approved or provided guidance on the study protocol, OR state that no ethical approval or guidance was required and explain why not.* |

Note that full information on the approval of the study protocol must also be provided in the manuscript.

# Animals and other research organisms

Policy information about studies involving animals; ARRIVE guidelines recommended for reporting animal research, and Sex and Gender in Research

| | |
|---|---|
| Laboratory animals | Mice were purchased from Charles River Laboratories and used for experiments: C57BL/6 (C57BL/6NCrl, strain code 027), CD1-nude (Crl:CD1-Foxn1nu, strain code 086), and NSG (NOD.Cg-PrkdcSCIDIl2rgtm1Wjl/SzJ, strain code 614). Mice were 6 to 10 weeks old with a body weight of 15 to 25 g at the start of the experiments. Animals were kept at 45-60% humidity and 20-24°C. |
| Wild animals | The study did not involve wild animals. |
| Reporting on sex | Female animals were used for most experiments to decrease biological variation, except for the biodistribution study investigating the sex-specificity of [18F]F-DTPA biodistribution. |
| Field-collected samples | The study did not involve samples that were collected in the field. |

| Ethics oversight | Animal experiments were conducted in accordance with animal welfare regulations in Germany and with permission from the District Government of Upper Bavaria (approvals ROB-55.2-2532.Vet_216-15, Vet_21-127, Vet_02-21-41 and ROB-55.2-2532_Vet_02-18-162). The animal protocol was reviewed by the commission defined by §15 of the German animal protection act and received approval. |
|---|---|

Note that full information on the approval of the study protocol must also be provided in the manuscript.

# Clinical data

Policy information about clinical studies
All manuscripts should comply with the ICMJE guidelines for publication of clinical research and a completed CONSORT checklist must be included with all submissions.

| Clinical trial registration | *Provide the trial registration number from ClinicalTrials.gov or an equivalent agency.* |
|---|---|
| Study protocol | *Note where the full trial protocol can be accessed OR if not available, explain why.* |
| Data collection | *Describe the settings and locales of data collection, noting the time periods of recruitment and data collection.* |
| Outcomes | *Describe how you pre-defined primary and secondary outcome measures and how you assessed these measures.* |

# Dual use research of concern

Policy information about dual use research of concern

## Hazards

Could the accidental, deliberate or reckless misuse of agents or technologies generated in the work, or the application of information presented in the manuscript, pose a threat to:

| No | Yes | |
|---|---|---|
| ☐ | ☐ | Public health |
| ☐ | ☐ | National security |
| ☐ | ☐ | Crops and/or livestock |
| ☐ | ☐ | Ecosystems |
| ☐ | ☐ | Any other significant area |

## Experiments of concern

Does the work involve any of these experiments of concern:

| No | Yes | |
|---|---|---|
| ☐ | ☐ | Demonstrate how to render a vaccine ineffective |
| ☐ | ☐ | Confer resistance to therapeutically useful antibiotics or antiviral agents |
| ☐ | ☐ | Enhance the virulence of a pathogen or render a nonpathogen virulent |
| ☐ | ☐ | Increase transmissibility of a pathogen |
| ☐ | ☐ | Alter the host range of a pathogen |
| ☐ | ☐ | Enable evasion of diagnostic/detection modalities |
| ☐ | ☐ | Enable the weaponization of a biological agent or toxin |
| ☐ | ☐ | Any other potentially harmful combination of experiments and agents |

# Plants

| Seed stocks | *Report on the source of all seed stocks or other plant material used. If applicable, state the seed stock centre and catalogue number. If plant specimens were collected from the field, describe the collection location, date and sampling procedures.* |
|---|---|
| Novel plant genotypes | *Describe the methods by which all novel plant genotypes were produced. This includes those generated by transgenic approaches, gene editing, chemical/radiation-based mutagenesis and hybridization. For transgenic lines, describe the transformation method, the number of independent lines analyzed and the generation upon which experiments were performed. For gene-edited lines, describe the editor used, the endogenous sequence targeted for editing, the targeting guide RNA sequence (if applicable) and how the editor was applied.* |
| Authentication | *Describe any authentication procedures for each seed stock used or novel genotype generated. Describe any experiments used to* |

| Authentication | *assess the effect of a mutation and, where applicable, how potential secondary effects (e.g. second site T-DNA insertions, mosiacism, off-target gene editing) were examined.* |
|---|---|

# ChIP-seq

## Data deposition

☐ Confirm that both raw and final processed data have been deposited in a public database such as GEO.

☐ Confirm that you have deposited or provided access to graph files (e.g. BED files) for the called peaks.

| Data access links
*May remain private before publication.* | *For "Initial submission" or "Revised version" documents, provide reviewer access links. For your "Final submission" document, provide a link to the deposited data.* |
|---|---|
| Files in database submission | *Provide a list of all files available in the database submission.* |
| Genome browser session
(e.g. UCSC) | *Provide a link to an anonymized genome browser session for "Initial submission" and "Revised version" documents only, to enable peer review. Write "no longer applicable" for "Final submission" documents.* |

## Methodology

| Replicates | *Describe the experimental replicates, specifying number, type and replicate agreement.* |
|---|---|
| Sequencing depth | *Describe the sequencing depth for each experiment, providing the total number of reads, uniquely mapped reads, length of reads and whether they were paired- or single-end.* |
| Antibodies | *Describe the antibodies used for the ChIP-seq experiments; as applicable, provide supplier name, catalog number, clone name, and lot number.* |
| Peak calling parameters | *Specify the command line program and parameters used for read mapping and peak calling, including the ChIP, control and index files used.* |
| Data quality | *Describe the methods used to ensure data quality in full detail, including how many peaks are at FDR 5% and above 5-fold enrichment.* |
| Software | *Describe the software used to collect and analyze the ChIP-seq data. For custom code that has been deposited into a community repository, provide accession details.* |

# Flow Cytometry

## Plots

Confirm that:

☒ The axis labels state the marker and fluorochrome used (e.g. CD4-FITC).

☒ The axis scales are clearly visible. Include numbers along axes only for bottom left plot of group (a 'group' is an analysis of identical markers).

☒ All plots are contour plots with outliers or pseudocolor plots.

☒ A numerical value for number of cells or percentage (with statistics) is provided.

## Methodology

| Sample preparation | Samples were either obtained from cell culture experiments (activation or proliferation assays, MACS) or collected from animal tissue or blood. Antibodies and other reagents used for cell staining are listed in Supplementary Table 2. |
|---|---|
| Instrument | LSR-Fortessa (Becton Dickinson) or CytoFLEX S (Beckman Coulter) |
| Software | FlowJo software (ver. 10.8.0 & 10.8.1; Becton Dickinson) |
| Cell population abundance | The exemplary gating strategy for CD8-positive CAR-T cells depicted in Extended Data Fig. 8c indicates the following cell abundance after gating.
0.819*0.989*0.858*0.627*0.648*0.999*0.352=0.099
The abundance of the exemplary CD8$^+$ CAR-T cells is ~10% which can be reliably detected and quantified. |

| Gating strategy | A representative gating strategy for CD8-positive CAR-T cells starting from samples collected from animal studies can be seen in Extended Data Fig. 8c.<br>In short, 1) the target cell population and counting beads were identified in a SSC-H/FSC-H plot, 2) doublets were excluded using a SSC-H/FSC-W plot, 3) dead cells that were positively stained by propidium iodide were excluded (live cell gating), 4) huCD45-pos cells were gated (leukocyte gating), 5) endogenous GFP expression in Raji tumor cells and CD3 staining of T cells was used to identify respective cell populations, 6) Streptavidin (binding to the Strep-tag within the CAR) and antibodies against the V5-tag or the EGFRt allowed the characterization of the CAR-T population, finally 7) CD8 antibody stain allowed the differentiation between CD4 and CD8 T cells. |
|---|---|

☒ Tick this box to confirm that a figure exemplifying the gating strategy is provided in the Supplementary Information.

# Magnetic resonance imaging

## Experimental design

| Design type | *Indicate task or resting state; event-related or block design.* |
|---|---|
| Design specifications | *Specify the number of blocks, trials or experimental units per session and/or subject, and specify the length of each trial or block (if trials are blocked) and interval between trials.* |
| Behavioral performance measures | *State number and/or type of variables recorded (e.g. correct button press, response time) and what statistics were used to establish that the subjects were performing the task as expected (e.g. mean, range, and/or standard deviation across subjects).* |

## Acquisition

| Imaging type(s) | *Specify: functional, structural, diffusion, perfusion.* |
|---|---|
| Field strength | *Specify in Tesla* |
| Sequence & imaging parameters | *Specify the pulse sequence type (gradient echo, spin echo, etc.), imaging type (EPI, spiral, etc.), field of view, matrix size, slice thickness, orientation and TE/TR/flip angle.* |
| Area of acquisition | *State whether a whole brain scan was used OR define the area of acquisition, describing how the region was determined.* |

Diffusion MRI ☐ Used ☐ Not used

## Preprocessing

| Preprocessing software | *Provide detail on software version and revision number and on specific parameters (model/functions, brain extraction, segmentation, smoothing kernel size, etc.).* |
|---|---|
| Normalization | *If data were normalized/standardized, describe the approach(es): specify linear or non-linear and define image types used for transformation OR indicate that data were not normalized and explain rationale for lack of normalization.* |
| Normalization template | *Describe the template used for normalization/transformation, specifying subject space or group standardized space (e.g. original Talairach, MNI305, ICBM152) OR indicate that the data were not normalized.* |
| Noise and artifact removal | *Describe your procedure(s) for artifact and structured noise removal, specifying motion parameters, tissue signals and physiological signals (heart rate, respiration).* |
| Volume censoring | *Define your software and/or method and criteria for volume censoring, and state the extent of such censoring.* |

## Statistical modeling & inference

| Model type and settings | *Specify type (mass univariate, multivariate, RSA, predictive, etc.) and describe essential details of the model at the first and second levels (e.g. fixed, random or mixed effects; drift or auto-correlation).* |
|---|---|
| Effect(s) tested | *Define precise effect in terms of the task or stimulus conditions instead of psychological concepts and indicate whether ANOVA or factorial designs were used.* |

Specify type of analysis: ☐ Whole brain ☐ ROI-based ☐ Both

| Statistic type for inference<br>(See Eklund et al. 2016) | *Specify voxel-wise or cluster-wise and report all relevant parameters for cluster-wise methods.* |
|---|---|
| Correction | *Describe the type of correction and how it is obtained for multiple comparisons (e.g. FWE, FDR, permutation or Monte Carlo).* |

# Models & analysis

| n/a | Involved in the study |
|-----|----------------------|
| ☐ | ☐ Functional and/or effective connectivity |
| ☐ | ☐ Graph analysis |
| ☐ | ☐ Multivariate modeling or predictive analysis |

**Functional and/or effective connectivity**

*Report the measures of dependence used and the model details (e.g. Pearson correlation, partial correlation, mutual information).*

**Graph analysis**

*Report the dependent variable and connectivity measure, specifying weighted graph or binarized graph, subject- or group-level, and the global and/or node summaries used (e.g. clustering coefficient, efficiency, etc.).*

**Multivariate modeling and predictive analysis**

*Specify independent variables, features extraction and dimension reduction, model, training and evaluation metrics.*

