## [Peer Review File · Nature Biomedical Engineering]

PET-based tracking of CAR T cells and viral gene transfer using a cell surface reporter that binds to lanthanide complexes

Corresponding Author: Prof Wolfgang Weber

Version 0:

Decision Letter:

Dear Professor Weber,

Thank you again for submitting to *Nature Biomedical Engineering* your manuscript, "DTPA-Receptor – A novel reporter gene system for the specific and sensitive PET imaging of CAR-T cells and AAV transduced cells". The manuscript has been seen by 4 experts, whose reports you will find at the end of this message.

You will see that the reviewers appreciate the work. However, they express concerns about the degree of support for the claims, and provide useful suggestions for improvement. We hope that with significant further work you can address the criticisms and convince the reviewers of the merits of the study. In particular, we would expect that a revised version of the manuscript provides:

* Evidence of whether cells transduced with the reporter gene system are non-immunogenic, as queried by Reviewers #1 and #4.

* Evidence of any functional effects on the expression of the reporter gene on CAR-T cells, as per the questions from Reviewers #1, #2 and #3.

* Clarification, supported by evidence, of the reason for the unexpected higher retention, in the kidney, of the CAR T cells expressing the anticalin-based reporter, as per the concerns of Reviewers #1 and #2.

When you are ready to resubmit your manuscript, please upload the revised files, a point-by-point rebuttal to the comments from all reviewers, the [reporting summary](https://www.nature.com/authors/policies/ReportingSummary.pdf), and a cover letter that explains the main improvements included in the revision and responds to any points highlighted in this decision.

Please follow the following recommendations:

* Clearly highlight any amendments to the text and figures to help the reviewers and editors find and understand the changes (yet keep in mind that excessive marking can hinder readability).

* If you and your co-authors disagree with a criticism, provide the arguments to the reviewer (optionally, indicate the relevant points in the cover letter).

* If a criticism or suggestion is not addressed, please indicate so in the rebuttal to the reviewer comments and explain the reason(s).

* Consider including responses to any criticisms raised by more than one reviewer at the beginning of the rebuttal, in a section addressed to all reviewers.

* The rebuttal should include the reviewer comments in point-by-point format (please note that we provide all reviewers will the reports as they appear at the end of this message).

* Provide the rebuttal to the reviewer comments and the cover letter as separate files.

We hope that you will be able to resubmit the manuscript within 20 weeks from the receipt of this message. If this is the case,

you will be protected against potential scooping. Otherwise, we will be happy to consider a revised manuscript as long as the significance of the work is not compromised by work published elsewhere or accepted for publication at *Nature Biomedical Engineering*.

We hope that you will find the referee reports helpful when revising the work. Please do not hesitate to contact me should you have any questions.

Best wishes,

Liqian

Dr Liqian Wang
Associate Editor, *Nature Biomedical Engineering*

Reviewer #1 (Report for the authors (Required)):

Summary:

This manuscript described a new genetic reporter system for PET imaging. The advantages of this new system are the (presumed) lack of immunogenicity, compatibility with ¹⁸F, lack of endogenous expression, and rapid blood clearance. The method is demonstrated both in the context of CAR-T therapy and gene therapy. The results are encouraging and suggest that DTPA-R is a sensitive reporter gene method for tracking therapeutic cells and genes.

Degree of advance

The technology is at an advanced stage and was demonstrated under realistic conditions in small-animal models.

Implications of the findings:

DTPA-R may become a valuable new tool for reporter gene imaging in research and, perhaps in the future, in the clinic.

Specific comments (minor and major):

1. (major) Are there any data or experiments showing that the DTPA-R reporter is non-immunogenic?
2. (minor) Could the removal of EGFRt from the CAR-T cells limit the utility of these CAR-T cells since the ablation functionality is lost?
3. (major) Are dead CAR T cells able to bind ¹⁸F-DTPA, and could the presence of dead CAR T cells in a tissue bias the imaging findings?
4. (minor) Fig 2 - What is the IC50 for the final [¹⁸F]F-DTPA agent, and how does it compare to IC50 measured in Fig 2a?
5. (minor) What is the rationale for loading the DTPA chelator with radioactive ¹⁷⁷Lu in Fig 3f? There is no mention of this point in the manuscript. Why not use stable ¹⁷⁵Lu? The presence of ¹⁷⁷Lu could interfere with the 511 keV gamma from ¹⁸F and reduce image quality while also worsening dosimetry.
6. (major) In Fig. 4, the PET signal in the kidneys appears notably different in (b) and (c), in terms of %ID/g. Is there an explanation for why the kidney signal would be higher when CAR-T cells express DTPA-R as opposed to EGFRt? Could it be that the two series of images are shown on different intensity scale?
7. (minor) Microscopic images (i.e. Fig 4j-l) should include scale bars.
8. (major) T cells are known for being highly susceptible to low doses of radiation. Could the author confirm that the exposure of the CAR T cells to ¹⁸F=DTPA in vivo does not affect their effector functions? This could be potentially tested in vitro using low concentration of ¹⁸F-DTPA representative of in vivo conditions.
9. (minor) Fig. 4d: Are the kidneys and bladder included in this analysis? Could the authors show the region of interest that was used for whole-body quantitation?
10. (minor) Fig. 4ef: Are the cell numbers normalized according to the amount of tissue? Larger tissue specimens will contain more cancer and CART cells so it seems that some normalization should be applied.
11. (minor) Fig 6g: Add the control animal (0 AAV9 dose) to the dose-response curve.
12. (minor) Fig 6h: "kideny" kidney

13. (minor) L 384: Isomorph MRT: Please explain what you mean by this term.

14 (minor) Fig S8E: What do the dashed/solid lines represent?

Reviewer #2 (Report for the authors (Required)):

Morath et. al. submitted an original research paper entitled "DTPA-Receptor – A reporter gene system for the specific and sensitive PET 1 imaging of CAR-T cells and AAV transduced cells". The study presented a novel genetic reporter system and its validation for non-invasively monitoring of the infused CAR-T cell (as an example of the advanced cell therapy) and AAV (as an example of viral gene transfer vector). The reporter-tracer pair is novel and highly functional for both sensitivity and specificity of detection. The results are so far one of the best for the genetic reporter systems developed for this purpose and has a good potential for clinical translation. The overall design, execution, and written presentation of the work are adequate, and the manuscript provides details of the work.

Major Points

1. The effect of the DPTA-R expression on CAR-T cells on their in vivo efficacy should be tested. Comparison of the in vivo efficacy between mock T cells, α CD19-CAR T (or α CD19-CAR TEGFRt), and α CD19-CAR TDTPA-R should be performed and presented.
2. The retention of the dubbed [18F]F-DTPA in kidney at 6h is higher than that of the PC3DTPA-R tumor. This needs be addressed for the possibility of specific binding of the dubbed [18F]F-DTPA in kidney (Fig. 3c, d). BioD study with radiometal chelated F-DTPA over a longer time period should address this issue. Statements in lines 47 and 465-472 regarding the inertness of the radioligand should be reconsidered in the light of the results from the suggested experiment. Regardless of the outcome of the proposed experiment, the kidney would have a strong PET signal at the time of clinical imaging (e.g., PET scanning in 1-2 hr after tracer injection). This will be a limitation and should be discussed in the manuscript.
3. It is not clear whether Fig. 3f presents the stability of [18F]F-DTPA (no metal chelation) or the [18F]F-Nic-D-Glu2-PEG4-CHX-A''-DTPA•TbIII (referred to as [18F]F-DTPA). If it was for the [18F]F-DTPA (no metal chelation), the stability of the [18F]F-Nic-D-Glu2-PEG4-CHX-A''-DTPA•TbIII (referred to as [18F]F-DTPA) should be tested and presented.

Minor Points

1. Details of the retroviral vector for each reporter should be described. The manuscript just states "To this end, RD114 cells were transiently transfected with the expression cassette via the CaCl₂-precipitation method." in line 525.
2. The source of human PBMC must be provided, line 527.
3. It is not clear whether the data shown in Fig. 1f were performed with the cells before or after the V5-Tag MACS sorting; please verify. In addition, the transduction efficiency presented in Fig. 1h for the DTPA-R-mRuby3 is very low. The transduction efficiency of all other reporters should be provided.
4. In Fig. S4, IHC staining of all tumors from non-transduced control cells should be presented as negative controls.

Reviewer #3 (Report for the authors (Required)):

Summary: This is a well written contribution that describes to the best of my knowledge, a first-of-its kind anticalin-based gene reporter for molecular imaging. Anticalin proteins, generated from combinatorial design from natural lipocalins, offer promising alternative to antibodies and antibody fragments. In this contribution, the coauthors adopted a metal-DTPA specific anticalin developed by others (CL31d) and they configure it with T-cell surface glycoprotein CD4 transmembrane domain linked by a V5-epitope tag to generate the first of its kind gene reporter for imaging. They name this gene reporter DTPA-(Receptor) or (DTPA-R), although there appears to be no functional activity of the expressed reporter "receptor". The authors compared retroviral transduction of DTPA-R and, as a control, to anticalin D6.4(Q77E) specific to colchicine (called Colchi-R) in human Jurkat cells, HEK293T, and PC3 cells. The team characterized the gene reporter "receptor", the PET labeled "ligand" to DTPA-R, and then provided examples of tracking CAR T-cells and AAV2/9 transduction. For characterization, they showed that fluorescent proteins encoded in the cytoplasmic portion of the CD4 reduced overall gene reporter expression and therefore did not use fluorescent protein reporter construct in their vector. They showed there was no difference in doubling times between activated transduced or activated WT Jurkat cells. They showed that the V5-tag could be used as a target for IHC and as others have shown, for magnetic-activated cell sorting. While 90Y(III) and Tb(III) metal complexes with DTPA-ligand have a higher affinity to CL31d, for diagnostic imaging they designed an 18F ligand based upon a trimethylamine, single-step radiofluorination using natural Tb as the metal chelated to DTPA. The ligands for both DTPA-R and Colchi-R had low, 20% radiochemical yield. The ligands were tested in DTPA-R and Colchi-R transfected PC3 cells and were stable (>97% intact) in two mice after 3 hrs of injection.

PET imaging studies were conducted of PC3 s.c. xenografts showing lack of non-specific uptake in cancer studies and favorable biodistribution of ligand and, as could be expected, low signal associated with the Colchi construct. The authors then tested the DTPA-4 on the CD-19 targeting 2nd generation CAR-T using scFV FMC63. Herein they replaced the EGFRt with the CD4-V5-CL31d vector in the expression cassette and showed equivalent transduction in PBMCs, similar numbers of receptors in CD3, CD4, and CXCR3 expressing T-cells, , and equivalent cell lysis. The authors

demonstrated PET imaging of CAR T-cells in animals engrafted with systemic Raji tumor, and estimated as few as 500 cells detected. There was no assessment of in vivo functionality of the CAR-T DTPA-R cells.

Finally, the authors used the DTPA-R reporter gene under the control of the CMV promoter to image AAV2/9 gene therapy via PET. Transduction levels imaged by PET were validated from ddPCR.

To this reviewer's knowledge, this is the first time that the anticalin technology (although it's been around for some time) has been effectively translated as a nuclear gene reporter. The contribution is exciting and could be strengthened by addressing a few points:

MAJOR POINTS:

(1) I was disappointed over the selection of materials for the supplemental section and the inclusion of less exciting characterization for the main body. The authors talk about characterization of DTPA-R in PC3 and Jurkat cell lines. Yet given that one of the two applications presented is imaging CAR-T cells, Figure S8 is probably most important since it characterizes DTPA-R in CAR-T cells. Hence this data should be presented within the contribution and is not supplementary.

(2) Along these lines, the number of DTPA-R receptors/cell are represented for the total CD3+ population, the subset CD4+ population, but not the CD8+ population. Could this data be added? Most importantly, while the authors show that the no of receptors in CXCR3 expressing cells is not different between CAR-T (DTPA-R) and CAR-T(EGFRt) cells, does the percentage of CXCR3+ expressing cells change with transduction with the DTPA-R gene reporter ?

(3) Lines 430-437 are somewhat overenthusiastic as the data presented is in no means comprehensive for evaluating biological activity, interference, and toxicity. I think the authors' message is that these preliminary data are exciting and are suggestive for future studies. For example, it remains to be seen that the incorporation of DTPA-R in a CD8+ CAR-T cells does not impact functionality.

MINOR POINTS:

(1) What does NT stand for in Fig 1g?

(2) Perhaps it is in the text, but could the figure captions describe the time of imaging after 18F-F-DTPA ?

(3) Line 157 – an incomplete sentence.

Reviewer #4 (Report for the authors (Required)):

This manuscript reports a novel engineered PET reporter system for quantitative monitoring of cell trafficking and gene therapy in vivo. The system comprises an engineered anticalin protein fused in frame to a transmembrane spanning scaffold as a receptor for a modified F-18 labeled DTPA ligand to enable PET imaging. The study is interesting and well crafted, but there is a key issue that needs to be addressed.

1) Fundamentally, the investigators are correct to outline the need for an ideal reporter gene/protein that should be small, bio-orthogonal, perhaps use F-18 for ease of use and non-immunogenic. The investigators have crafted a nice study with convincing data describing a third-generation extracellular anticalin (that binds DTPA-metal complexes with high affinity) now fused in frame to V5 (to facilitate isolation or cellular analysis), and a CD4 transmembrane domain (for cell surface localization), thus generating a genetically-encoded PET reporter fusion construct. Detailed characterization of the construct and its applications are reasonable and informative. Building upon the cognate affinity ligand, a DTPA-metal (Tb) complex that was previously used to seed the development of the anticalin, the investigators have also developed a modified PET radioligand that incorporates a scaffold for facile F-18 radiolabeling. Overall, the genetically-encoded reporter builds on prior work from this group wherein each component is not novel per se, but the fusion construct is compelling and an interesting advance. Same for the radioligand.

2) The most important critique relates to the goals for this long line of investigation, which requires, as stated by the investigators themselves, that the construct be non-immunogenic in order to provide a significant advance over many other genetically-encoded PET reporters described in the past (and even translated into the clinic). Herein, most studies in vivo were performed with immuno-compromised mice strains. In this regard, importantly, in Figure 6, the investigators reported that immunocompetent C57Bl/6 mice showed nearly two-fold less expression levels of the AAV9 vector compared to immuno-compromised CD-1 nude mice. Indeed, while interpretation of these data might be open to debate, these data might be pointing to immuno-reactivity derived from the reporter in the AAV9. Fundamentally, while anticalins are derived from an endogenous gene product, the lipocalins, the specific anticalin used herein is a derived protein product, screened from a targeted random mutagenesis library for binding to DPTA-metal complexes, and thus, not an endogenous protein product. Overall, there are three potentially immunogenic reporter domains of concern in vivo: the anticalin, V5, and where applied, the RFP variant. The investigators are compelled to rigorously demonstrate that the new reporter is indeed 'non-immunogenic' in the context of use as a reporter for cell trafficking or gene therapy in vivo. And while it is not always clear as applied herein, the selection marker per se will also represent a foreign gene product that may also be immunogenic per se and must be included in the analysis. Claims of "low immunogenic potential" are not sufficient as applied herein. At the very least, the lifetime in vivo in immunocompetent mice of cells transduced with the full reporter in comparison to non-transduced native cells should be determined by orthogonal methods (likely conventional non-imaging approaches). Characterizing

seroconversion may also be a reasonable approach. Characterizing the specific TCR repertoire induced by the reporter (and selection marker) is probably not required. But some assessment of overall transduced cell survival in native and sensitized mice should be undertaken. Any other in vitro tests of immune activation would be complementary, but this reviewer is most focused on the practical aspect of long-term survival in vivo of cells expressing the reporter construct compared to genetically matched native cells as a final test of validity.

3) Are the investigators interested in applying Tb-161 for therapy with variants of the ligand?

4) Lines 185-189. What about the purity of the colchicine analogue as contributory to the lower yields in the radiolabeling regime?

5) Lines 363-370. For the general reader, briefly explain the expected organ tropism of the AAV9 after systemic injection.

Version 1:

Decision Letter:

Dear Prof Weber,

Thank you for your revised manuscript, "DTPA-Receptor – A novel reporter gene system for the specific and sensitive PET imaging of CAR-T cells and AAV transduced cells". Having consulted with the original reviewers, I am pleased to write that we shall be happy to publish the manuscript in *Nature Biomedical Engineering*.

We will be performing detailed checks on your manuscript, and in due course will send you a checklist detailing our editorial and formatting requirements. You will need to follow these instructions before you upload the final manuscript files.

Best wishes,

Filipe

Dr Filipe Almeida

Senior Editor, <http://www.nature.com/nbme> > *Nature Biomedical Engineering*

Reviewer #1 (Report for the authors (Required)):

Summary: The manuscript was revised with significant new data presented. In particular, the therapeutic efficacy of the modified CAR-T cells was demonstrated in vivo, cell dosimetry estimates were provided, and additional information was included to elucidate the change in kidney uptake when the reporter was expressed in vivo. With these changes, I believe the manuscript is suitable for publication. A large amount of information is provided to characterize the performance and demonstrate the application of this new reporter system. The only missing piece is the demonstration that the reporter system does not induce a strong immune response in vivo, though I concur with the authors' determination that such study is beyond the scope of this manuscript.

Reviewer #2 (Report for the authors (Required)):

Authors carefully addressed all the concerns and comments raised by the reviewer with appropriate evidence and scientific reasoning. The revised manuscript is ready for publication as is.

Reviewer #3 (Report for the authors (Required)):

Concern addressed and the manuscript is now suited for publication.

Reviewer #4 (Report for the authors (Required)):

My concerns have been addressed.

Response to Reviewers' comments

Reviewer 1

Comment 1 (major) “Are there any data or experiments showing that the DTPA-R reporter is non-immunogenic?”

A definitive assessment of the immunogenicity of DTPA-R in humans will require human studies and without those, one cannot claim that the DTPA-R reporter is non-immunogenic. However, there are several lines of evidence that the potential for immunogenicity in humans is limited:

1. DTPA-R is based on the human Lipocalin 2 protein.
2. An *in-silico* analysis (IEDB.org, see section in supplementary information starting on line 730) revealed that only two peptide fragments of DTPA-R (peptides 'rank 15' and 'rank 17' in Fig. S1) were predicted to be potentially immunogenic.
3. Eight Anticalin proteins have been investigated in clinical studies (clinicaltrials.gov and Rothe et al.¹), and there has been no evidence that an immune response was generated.

Nevertheless, we agree with the reviewer that immunogenicity in humans and potential detrimental effects on reporter gene imaging or CAR-T function in humans cannot be excluded at this point in time. This is now specifically acknowledged in the discussion lines 482-492.

Should immunogenicity in humans be problematic, it is feasible to design a less immunogenic DTPA-binding Anticalin version because the structure of CL31d (DTPA-binding Anticalin within DTPA-R) in complex with Tb•DTPA has previously been solved by protein crystallography². Specifically, the risk for immunogenicity of peptide 'rank 15' could be reduced by returning to the Lcn2 sequence, especially as the residue is located at the side of the calyx, pointing outwards, and is not directly involved in ligand binding. Deimmunization of the second peptide (peptide 'rank 17') will require more complex protein engineering to replace single amino acid residues to decrease potential immunogenicity while preserving or even increasing affinity for CHX-A'-DTPA•metal complexes.

Comment 2 (minor) “Could the removal of EGFRt from the CAR-T cells limit the utility of these CAR-T cells since the ablation functionality is lost?”

Breyanzi is an approved CAR-T cell therapy product that features the EGFRt as a second membrane protein besides the CAR. However, to our knowledge EGFRt targeted therapy is not used in clinical routine to treat side effects of Breyanzi. Most other CAR-T cell therapies do not contain EGFRt for cell ablation. Therefore, we do not consider the removal of EGFRt to present a problem. As pointed out by Reviewer 4 in point 3, the ability to deliver therapeutic radiometals (such as Terbium-161•DTPA) is an excellent alternative mechanism for *in vivo* cell ablation. We

are currently investigating such ablation approaches, but we think that their study is beyond the scope of this publication. We have addressed the relevance of a suicide gene for CAR-T cell ablation in lines 272-274 of the revised manuscript.

Comment 3. (major). “Are dead CAR T cells able to bind 18F-DTPA, and could the presence of dead CAR T cells in a tissue bias the imaging findings?”

In principle, the binding of [¹⁸F]F-DTPA to the DTPA-R does not require a living cell. Nevertheless, membrane proteins on dead cells will quickly lose function over time, and the immune system rapidly removes dead cells. Cell death does not lead to an immediate signal loss in a PET scan, but the minimal imaging interval of [¹⁸F]F-PET measurements is one day, which is sufficient for the immune system to reliably remove most dead cells. For example, apoptotic cells are removed from the circulation with a half-life of 10 min³. We have clarified this in the discussion, lines 509 to 513, of the revised manuscript.

Comment 4. (minor). “Fig 2 - What is the IC₅₀ for the final [18F]F-DTPA agent, and how does it compare to IC₅₀ measured in Fig 2a?”

We have measured the IC₅₀ of ¹⁹F-Glu₂-PEG₄-CHX-A''-DTPA•^{nat}Tb using the radioactive ligand, as requested by the reviewer. The IC₅₀ was found to be 0.3±0.15 nM, which is even lower than the 1.4 nM reported previously for the DTPA•Y complex and comparable to the NH₂-CHX-A''-DTPA•Tb (see new Fig. 2e, Fig. S9 and Table S4).

To further confirm the high affinity of DTPA-R for ¹⁹F-Glu₂-PEG₄-CHX-A''-DTPA•Tb we performed surface plasmon resonance (SPR) spectroscopy and real-time interaction cytometry (RT-IC). Both methods offer not only the equilibrium constant (K_D) but also the on- and off-rates. While SPR is the well-established standard for kinetic measurements *in vitro*, RT-IC is an innovative approach offering kinetic affinity measurements on life cells. As depicted in Figure 2e, all these affinity measurements resulted in an affinity in the range of 0.2 to 0.6 nM (see new Fig. 2e, Fig. S9 and Table S4).

Comment 5. (minor) “What is the rationale for loading the DTPA chelator with radioactive 177Lu in Fig 3f? There is no mention of this point in the manuscript. Why not use stable 175Lu? The presence of 177Lu could interfere with the 511 keV gamma from 18F and reduce image quality while also worsening dosimetry.”

All compounds in this experiment were either labeled by F-18 or Lu-177, and we did not perform labeling of one substance with two radioisotopes. The figure legend has been modified accordingly.

The [^{18}F]F-DTPA radioligand is composed of chemical building blocks connected by peptide bonds. We have included respective fragment reference substances to demonstrate that the size-exclusion chromatography can separate fragments of [^{18}F]F-DTPA by delayed elution. One reference was prepared by the acidic hydrolysis of the radioligand ([^{18}F]F-nicotinic acid is detected here). The other relevant reference was the NH_2 -CHX-DTPA moiety, which we labeled with a lanthanide radio-metal (Lu-177) that is regularly available in our department and is known to form stable complexes with the CHX-DTPA chelator. We could also have used stable Lu-175, but Lu-177 was preferable because there is an offset between the absorbance and the radioactivity detector of the HPLC, which we could avoid by also detecting the radioactivity signal for all samples. The term "hydrolysis fragments" was explained in more detail to clarify the figure legend.

Comment 6. (major) "In Fig. 4, the PET signal in the kidneys appears notably different in (b) and (c), in terms of %ID/g. Is there an explanation for why reason why the kidney signal would be higher when CAR-T cells express DTPA-R as opposed to EGFRt? Could it be that the two series of images are shown on different intensity scale?"

Thank you for pointing out this unexpected finding. First of all, we can confirm that the images were all scaled to the same maximum %ID/ml and that there is a real increase in kidney activity concentration in mice treated with CAR-T cells expressing DTPA-R (see also new supplemental Fig. 16).

We have therefore reevaluated kidney uptake in all mice imaged with [^{18}F]F-DTPA ligand and found that there was a significant correlation between kidney uptake and the binding of [^{18}F]F-DTPA to CAR-T^{DTPA-R} cells or host organs transduced by AAVs to express DTPA-R (see new supplemental Fig. S16a-b). However, V5-tag immunohistochemistry showed only small amounts of CAR-T cells in the kidneys (see Fig. S11). There was also only little transduction of renal cells following injection of AAV9 vectors encoding DTPA-R. Measurement of activity in skeletal muscles (an organ that is not infiltrated by CAR-T cells) showed no differences between animals with a low or high expression level of DTPA-R (see Fig. S16c). This suggests that there was no major impairment of renal function.

However, visual and quantitative analysis of the PET images revealed that mice harboring DTPA-R expressing cells retained significant amounts of radioactivity in the renal cortex, whereas the radioactivity was quickly cleared to the pelvis in control animals (Fig. S16d,g,h).

It is well established that Anticalin proteins labeled with radiometals and used as PET tracers are rapidly accumulated and retained in the renal cortex of mice^{4, 5}. Following i.v. injection typical activity concentrations are in the order of 100%ID/ml kidney volume with only slow clearance over time. We therefore hypothesized that the significant correlation between renal activity and the number of DTPA-R expressing cells is due to binding of [^{18}F]F-DTPA to shedded DTPA-R or the shedding of DTPA-R ectodomains with bound [^{18}F]F-DTPA. The shedded DTPA-R/[^{18}F]F-DTPA would be rapidly cleared from the blood pool by the kidneys and would remain there at the time of imaging (90 min p.i.).

We therefore investigated if cells transduced with DTPA-R shed the ectodomain protein into the culture media. Using Western blot analysis, we were indeed able to detect DTPA-R fragments in

the culture media (Fig. S16k). As the band in the Western blot shifted by ~3 kDa, we estimated that the cleavage site was within the CD4 transmembrane domain and the DTPA-R fragments thus retained their ability to bind [¹⁸F]F-DTPA as confirmed by size-exclusion chromatography (Fig. S16l). From these data we conclude, that shedded DTPA-R ectodomain/[¹⁸F]F-DTPA complexes most likely explain the increased radioactivity concentration in the cortex of mice harboring DTPA-R expressing cells.

This has been added in the supplemental information starting at line 969.

Comment 7. (minor) “Microscopic images (i.e. Fig 4j-l) should include scale bars.”

Scale bars have been added to the IHC panels in Fig. 5j-l.

Comment 8. (major) “T cells are known for being highly susceptible to low doses of radiation. Could the author confirm that the exposure of the CAR T cells to 18F-DTPA in vivo does not affect their effector functions? This could be potentially tested in vitro using low concentration of 18F-DTPA representative of in vivo conditions.”

While lymphocytes are highly susceptible to ionizing radiation, a positron emitter with a short physical half-life like Fluorine-18 only transfers a low amount of energy to a CAR-T cell that has been labeled with the radiopharmaceutical. Zanzonico et al.⁶ have extensively studied the effects of radiolabeling on the function of lymphocytes. They found no significant effects on lymphocyte function for radiation doses of up to 8.3 Gy from ¹³¹I-FIAU. For external beam radiotherapy, toxic effects have been observed at much lower doses (see review by Harald Paganetti)⁷. However, these studies have used much higher dose rates than the ones produced by radionuclides used in nuclear medicine.

We have used our data on the correlation between the *in vivo* PET signal and the number of CAR-T cells (Fig. 7) to determine the amount of radioactivity bound to the CAR-T cells and to calculate the resulting radiation doses. We have assumed conservatively that the activity is not cleared biologically but that it decays entirely on the surface and/or in the cytoplasm of the CAR-T cell. Using the MIRDcell software, we calculated the radiation dose that a CAR-T cell receives from the radioactivity it has accumulated. This radiation dose is 0.037 Gy and thus far below the level at which to expect biological effects. However, radiation doses to clusters of CAR-T cells can be substantially higher due to a “cross-fire-effect”, i.e. the irradiation of cells by radioactivity bound by neighboring cells. We have therefore also evaluated the radiation dose to spherical clusters of CAR-T cells, ranging from 0.1 mm to 1 mm. The median radiation dose per CAR-T cell in these clusters were significantly higher (0.086-0.714 Gy) but still substantially lower than the radiation dose levels at which to expect an impact on lymphocyte function. The details of these radiation dose calculations and their results are summarized in new supplementary information, in lines 919 to 967.

Comment 9. (minor) "Fig. 4d: Are the kidneys and bladder included in this analysis? Could the authors show the region of interest that was used for whole-body quantitation?"

Tissues and organs that contain unspecifically accumulated [¹⁸F]F-DTPA were not included in the "cumulative specific signal". Boxes with dashed lines indicating the region analyzed were added to one of the last panels (now Fig. 5b). The figure legend has also been improved to state which kind of signals were not included (lines 342-343).

Comment 10. (minor) "Fig. 4ef: Are the cell numbers normalized according to the amount of tissue? Larger tissue specimens will contain more cancer and CART cells so it seems that some normalization should be applied."

For blood samples, the numbers were normalized to 100 µl of blood. For spleen samples, numbers have now been normalized to 50 mg of tissue, which approximates the weight of the spleen of a NSG mouse. Regarding the bone marrow, we deliberately decided not to normalize to the amount of tissue. This decision was made because it is challenging (and error-prone) to determine the amount of bone marrow tissue (containing the CAR-T cells) flushed out using a syringe. For this reason, we believe the representation as the total / absolute number of CAR-T cells is more sound.

Comment 11. (minor) "Fig 6g: Add the control animal (0 AAV9 dose) to the dose-response curve."

Data from these animals has been added to the new Fig. 7 (previously Fig 6).

Comment 12. (minor) "Fig 6h: "kideny" kidney"

We have corrected this mistake.

Comment 13. (minor) "L 384: Isomorph MRT: Please explain what you mean by this term."

You are right, we wanted to write "isotropic MRT" instead of "isomorph MRT". We have deleted the term from the main text and have moved the corrected term to the M&M section in lines 537-539 of the supplementary information.

Comment 14 (minor) "Fig S8E: What do the dashed/solid lines represent?"

The different lines in Fig. S8e (now Fig. 4d) represent the mean of n=3 wells for each condition. The 95% confidence interval is depicted as a shaded area. The figure legend has been extended to make this clear.

Reviewer 2

Major Points

Comment 1. "The effect of the DTPA-R expression on CAR-T cells on their *in vivo* efficacy should be tested. Comparison of the *in vivo* efficacy between mock T cells, α CD19-CAR T (or α CD19-CAR T^{EGFRt}), and α CD19-CAR T^{DTPA-R} should be performed and presented."

The proposed experiment has been conducted and confirmed the comparable anti-tumor effect of α CD19-CAR-T^{EGFRt} and α CD19-CAR-T^{DTPA-R} cells (please see Fig. 4f-l). We compared these CAR-T cell therapies in an i.v.-injected NALM-6 xenograft model in which we followed the eradication of the NALM-6 tumor cells by bioluminescence imaging and body weight measurements. The individual BLI results are shown in Fig. 4i, and a biostatistical analysis is shown in Fig. 4j. This additional *in vivo* experiment is described in the manuscript in lines 302-307.

Comment 2. "The retention of the dubbed [18F]F-DTPA in kidney at 6h is higher than that of the PC3DTPA-R tumor. This needs be addressed for the possibility of specific binding of the dubbed [18F]F-DTPA in kidney (Fig. 3c, d). BioD study with radiometal chelated F-DTPA over a longer time period should address this issue. Statements in lines 47 and 465-472 regarding the inertness of the radioligand should be reconsidered in the light of the results from the suggested experiment. Regardless of the outcome of the proposed experiment, the kidney would have a strong PET signal at the time of clinical imaging (e.g., PET scanning in 1-2 hr after tracer injection). This will be a limitation and should be discussed in the manuscript."

Thank you for making this important observation. We have investigated the kidney uptake in more detail as described in response to reviewer 1, who commented similarly.

Since the maximum observed activity concentration at the time of imaging was 16.3% ID/ml, we do not consider this to represent a clinically relevant limitation. For example, kidney uptake of the prostate-specific membrane antigen (PSMA) radioligand ¹⁸F-PSMA-1007⁸ in mice at the time of

PET imaging is about 84-142 %ID/g. Nevertheless, this ligand is now broadly used clinically for imaging of prostate cancer patients. Other clinically used radiotracers show a similar kidney uptake, see list below. Furthermore, this high renal uptake only occurred when there was a very large amount of radioactivity bound to CAR-T cells or AAV-transduced tissues (more than 10% of the injected activity). This scenario is unlikely to occur in humans. Therefore, the renal activity concentrations will be substantially lower in most scenarios.

Kidney uptake in mice of common PET radioligand types:

- 3.25 %ID/g kidney uptake at t=60 min for [⁶⁸Ga]Ga-DOTA-TATE⁹
- 46 %ID/g kidney uptake at t=75 min for [¹⁸F]F-Folic_acid¹⁰
- 84-142 %ID/g kidney uptake at t=60 min for ¹⁸F-PSMA-1007⁸
- 80-90 %ID/g kidney uptake at t=24 h for [¹¹¹In]In-Exendin4¹¹
- 15.7-23.6 %ID/g kidney uptake at t=45 min for [¹¹C]choline¹²
- Furthermore, nearly all protein PET tracers (antibodies, nanobodies, scaffold proteins) have far higher kidney accumulation compared to [¹⁸F]F-DTPA.

Comment 3. "It is not clear whether Fig. 3f presents the stability of [18F]F-DTPA (no metal chelation) or the [18F]F-Nic-D-Glu2-PEG4-CHX-A"-DTPA•TbIII (referred to as [18F]F-DTPA). If it was for the [18F]F-DTPA (no metal chelation), the stability of the [18F]F-Nic-D-Glu2-PEG4-CHX-A"-DTPA•TbIII (referred to as [18F]F-DTPA) should be tested and presented."

As you described, [¹⁸F]F-DTPA refers to the [¹⁸F]F-radioligand in complex with cold terbium if not stated otherwise. The experiment presented in Fig. 3f has been performed with the complex (= [¹⁸F]F-DTPA). We have improved the figure legend to make this clearer.

Minor Points

Comment 1. "Details of the retroviral vector for each reporter should be described. The manuscript just states "To this end, RD114 cells were transiently transfected with the expression cassette via the CaCl₂-precipitation method." in line 525."

A schematic map of the respective lentiviral constructs is shown in Fig. 4a,b. For DTPA-R and Colchi-R, a sequence map of the open reading frame is included in the supplementary information. We have added more references on the source of the constructs and have also expanded the information on the retroviral plasmid. Please see lines 80-83 of the supplementary information.

Comment 2. "The source of human PBMC must be provided, line 527."

A sentence with more information on the human PBMC source has been included in the M&M section in line 85 of the supplementary information.

Comment 3. "It is not clear whether the data shown in Fig. 1f were performed with the cells before or after the V5-Tag MACS sorting; please verify. In addition, the transduction efficiency presented in Fig. 1h for the DTPA-R-mRuby3 is very low. The transduction efficiency of all other reporters should be provided."

The different Jurkat lines were created by retroviral transduction, and the highest 10% of transduction events were isolated by FACS. The Figure legend has been expanded to improve clarity (lines 117-118). In Fig. 1h, we did not perform transduction, but we used a mixture of the two stable cell lines in a ratio of 95:5%. It was intended to purify a small (5%) cell population from a larger quantity of negative cells (95%) to demonstrate the purity and efficiency of the MACS procedure. The figure legend has been modified to increase clarity in line 122. A statement on the transduction efficiency (~25% for PBMCs) has been added to the M&M section in line 96 of the supplementary information. The transduction efficiency of one experiment with PBMCs is additionally shown in Fig. 4f now.

“Comment 4. In Fig. S4, IHC staining of all tumors from non-transduced control cells should be presented as negative controls.”

The predominantly used subcutaneous xenograft tumor model was PC3 cells expressing the DTPA-R or Colchi-R reporter gene above the right and left shoulder, respectively. The IHC was performed to characterize these tumor models and to demonstrate ongoing DTPA-R/Colchi-R expression of the cells after weeks and months *in vivo*. Control tumors without a reporter gene were not inoculated into animals because there was no need from an experimental point of view at that time. The anti-V5 IHC method was established using Jurkat and Jurkat^{DTPA-R} cells grown *in vitro* that were FFPE embedded. The IHC was clearly negative for Jurkat and clearly positive for Jurkat^{DTPA-R} cells. The specificity of the V5 IHC staining with the SV5-Pk1 antibody on FFPE tissue can be seen in Fig. S11 and Fig. 7i and for tumor stroma in Fig. S5g,i,j,l. Due to strict 3R regulations and the limited gain in scientific insight in our opinion, we would prefer not to sacrifice additional animals just for another negative control.

Reviewer 3

Comment 1. “I was disappointed over the selection of materials for the supplemental section and the inclusion of less exciting characterization for the main body. The authors talk about characterization of DTPA-R in PC3 and Jurkat cell lines. Yet given that one of the two applications presented is imaging CAR-T cells, Figure S8 is probably most important since it characterizes DTPA-R in CAR-T cells. Hence this data should be presented within the contribution and is not supplementary.”

We agree on the importance of the data presented in the old Fig. S8. We included this figure (strengthened by additional functional *in vivo* studies, as suggested in your major point 3) as new Fig. 4.

Comment 2. “Along these lines, the number of DTPA-R receptors/cell are represented for the total CD3+ population, the subset CD4+ population, but not the CD8+ population. Could this data be added? Most importantly, while the authors show that the no of receptors in CXCR3 expressing cells is not different between CAR-T (DTPA-R) and CAR-T(EGFRt) cells, does the percentage of CXCR3+ expressing cells change with transduction with the DTPA-R gene reporter?”

The figure S8c (new Fig. 4c) shows the surface density of relevant receptors on primary human CAR-T cells co-expressing EGFRt or DTPA-R. We have chosen CD3, CD4, CXCR3 as examples to illustrate that the exchange of EGFRt in the CAR-T-2A-EGFRt expression cassette for DTPA-R does not change the expression levels of important receptors on a T cell. We have divided the graph S8c into separate panels to explain which measurements are depicted on the y-axis.

Comment 3. “Lines 430-437 are somewhat overenthusiastic as the data presented is in no means comprehensive for evaluating biological activity, interference, and toxicity. I think the authors’ message is that these preliminary data are exciting and are suggestive for future studies. For example, it remains to be seen that the incorporation of DTPA-R in a CD8+ CAR-T cells does not impact functionality.”

An *in vivo* study characterizing the functionality of anti-CD19 CAR-T^{DTPA-R} cells has been requested by Reviewer 2, too (major point 1). We have confirmed experimentally that CAR-T^{DTPA-R} and CAR-T^{EGFR-T} cells show comparable functionality (Fig. 4f-l; see lines 298-307 of the revised manuscript).

Minor points

Comment 1. What does NT stand for in Fig 1g?

We changed “NT” (which meant “not transduced”) to “wild type” to make it clearer.

Comment 2. Perhaps it is in the text, but could the figure captions describe the time of imaging after 18F-F-DTPA ?

Thank you for the remark. We additionally included the imaging time point in the figure legends in lines 339 and 388 of the revised manuscript.

Comment 3. Line 157 – an incomplete sentence.

We apologize for this oversight. We have changed the wording and extended the sentence (now in lines 163-165 of the revised manuscript).

Reviewer 4

Comment 1: “This manuscript reports a novel engineered PET reporter system for quantitative monitoring of cell trafficking and gene therapy in vivo. The system comprises an engineered anticalin protein fused in frame to a transmembrane spanning scaffold as a receptor for a modified F-18 labeled DTPA ligand to enable PET imaging. The study is interesting and well crafted, but there is a key issue that needs to be addressed. Fundamentally, the investigators are correct to outline the need for an ideal reporter gene/protein that should be small, bio-orthogonal, perhaps use F-18 for ease of use and non-immunogenic. The investigators have crafted a nice study with convincing data describing a third-generation extracellular anticalin (that binds DTPA-metal complexes

with high affinity) now fused in frame to V5 (to facilitate isolation or cellular analysis), and a CD4 transmembrane domain (for cell surface localization), thus generating a genetically-encoded PET reporter fusion construct. Detailed characterization of the construct and its applications are reasonable and informative. Building upon the cognate affinity ligand, a DTPA-metal (Tb) complex that was previously used to seed the development of the anticalin, the investigators have also developed a modified PET radioligand that incorporates a scaffold for facile F-18 radiolabeling. Overall, the genetically-encoded reporter builds on prior work from this group wherein each component is not novel per se, but the fusion construct is compelling and an interesting advance. Same for the radioligand.

Thank you for these encouraging comments.

Comment 2. "The most important critique relates to the goals for this long line of investigation, which requires, as stated by the investigators themselves, that the construct be non-immunogenic in order to provide a significant advance over many other genetically-encoded PET reporters described in the past (and even translated into the clinic). Herein, most studies in vivo were performed with immuno-compromised mice strains. In this regard, importantly, in Figure 6, the investigators reported that immunocompetent C57Bl/6 mice showed nearly two-fold less expression levels of the AAV9 vector compared to immuno-compromised CD-1 nude mice. Indeed, while interpretation of these data might be open to debate, these data might be pointing to immuno-reactivity derived from the reporter in the AAV9. "

We agree with the reviewer that it is very likely that the human DTPA-R protein can cause an immune response in immunocompetent mice. We have added this to the discussion section of the revised manuscript, lines 482-487. In this new paragraph we also emphasize that DTPA-R is fundamentally limited for long-term studies in immunocompetent animals. DTPA-R was developed with the intention to study human cells in immunocompromised animals for translational research projects, and potentially in humans. Performing long-term studies in mice would require a murine version of DTPA-R which is beyond the scope of this manuscript.

Comment 3. "Fundamentally, while anticalins are derived from an endogenous gene product, the lipocalins, the specific anticalin used herein is a derived protein product, screened from a targeted random mutagenesis library for binding to DPTA-metal complexes, and thus, not an endogenous protein product. Overall, there are three potentially immunogenic reporter domains of concern in vivo: the anticalin, V5, and where applied, the RFP variant. The investigators are compelled to rigorously demonstrate that the new reporter is indeed 'non-immunogenic' in the context of use as a reporter for cell trafficking or gene therapy in vivo. And while it is not always clear as applied herein, the selection marker per se will also represent a foreign gene product that may also be

immunogenic per se and must be included in the analysis. Claims of “low immunogenic potential” are not sufficient as applied herein.”

We agree with the reviewer that the mutations of the human lipocalin could lead to immunogenicity in humans. The same is true for the V5-tag and the RFP variant. The V5 tag and the RFP were only introduced for the experimental validation of the DTPA-R reporter gene they are not required for PET imaging and would not be part of a clinically used reporter gene.

While DTPA-R is very likely to be immunogenic in mice (see response to comment 2) human studies are needed to determine if the Anticalin domain of DTPA-R or the fusion protein is immunogenic in humans. It is encouraging that clinical studies performed with other Anticalin proteins have shown no signs of immunogenicity. Furthermore, an *in silico* analysis showed a relatively low likelihood of a T-cell response to DTPA-R (see response to reviewer 1, comment 1). However, this is not a guarantee that an immune reaction will not occur in some patients when the Anticalin is expressed on the cell surface. Therefore, we agree with the reviewer that it is premature to claim that the DTPA-R non-immunogenic. We have addressed the issue of immunogenicity in the discussion section of the revised manuscript, lines 482-4492.

Comment 4. “At the very least, the lifetime in vivo in immunocompetent mice of cells transduced with the full reporter in comparison to non-transduced native cells should be determined by orthogonal methods (likely conventional non-imaging approaches). Characterizing seroconversion may also be a reasonable approach. Characterizing the specific TCR repertoire induced by the reporter (and selection marker) is probably not required. But some assessment of overall transduced cell survival in native and sensitized mice should be undertaken. Any other in vitro tests of immune activation would be complementary, but this reviewer is most focused on the practical aspect of long-term survival in vivo of cells expressing the reporter construct compared to genetically matched native cells as a final test of validity.”

We agree that it would be mandatory to characterize the immunogenicity of DTPA-R in the described mouse studies, if our aim were to develop a reporter gene for long-term studies of transduced cells in immunocompetent mice. However, we do not believe that the human protein DTPA-R is suitable for such studies because an immune response is very likely to occur. We have clarified this in the revised version of the manuscript (lines 482-492). However, the study presented in Fig. 7b,d that features immunocompetent C57BL/6 mice transduced with the DTPA-R should be a good model to assess potential immune reactions to the DTPA-R expressing cells and to assess if for example the formation of anti-drug-antibodies leads to functional impairment of [¹⁸F]F-DTPA binding and/or a loss of DTPA-R-labeled cells. The 4-week time frame in which we image the mice should be sufficient for the immune system to generate an adaptive immune response against the DTPA-R expressing cells. As presented, the [¹⁸F]F-DTPA signal is not lower in week 4 after transduction compared to week 2.

Please compare Reviewer 1, point (1) with the same request.

Comment 3. “Are the investigators interested in applying Tb-161 for therapy with variants of the ligand?”

DTPA-R binds CHX-A''-DTPA•Tb complexes with high affinity and CHX-A''-DTPA•Tb-161 could deliver therapeutic radiation doses to CAR-T cells expressing DTPA-R¹³. In fact, the Anticalin CL31d (the CHX-DTPA•metal binding domain of DTPA-R) has initially been developed for pre-targeted radiopharmaceutical therapy applications^{2, 14}. This could serve as a safety mechanism for CAR-T cell therapy, which allows for *in vivo* cell ablation. We have added a brief outlook for this potential application of DTPA-R to the manuscript (line 272-274).

Comment 4. “Lines 185-189. What about the purity of the colchicine analogue as contributory to the lower yields in the radiolabeling regime?”

The synthesis yields for both radioligands, ¹⁸F-DTPA and ¹⁸F-Colchicine were good. Because we focused in the end on the ¹⁸F-DTPA, mostly due to far superior excretion characteristics *in vivo*, we did not include as many details in the main text for the ¹⁸F-Colchicine (also due to the word limits). The purity of the precursor molecule (TMA-Colchicine) and the final radioligand (¹⁸F-Colchicine) was high and constituted no problem.

Comment 5. “Lines 363-370. For the general reader, briefly explain the expected organ tropism of the AAV9 after systemic injection.”

Thank you for the recommendation. A sentence explaining the expected AAV9 transduction pattern has been included in lines 408-410.

References

1. Rothe, C. & Skerra, A. Anticalin^R Proteins as Therapeutic Agents in Human Diseases. *BioDrugs* **32**, 233-243 (2018).
2. Eggenstein, E., Eichinger, A., Kim, H.J. & Skerra, A. Structure-guided engineering of Anticalins with improved binding behavior and biochemical characteristics for application in radio-immuno imaging and/or therapy. *J Struct Biol* (2013).
3. Wei, X. et al. Real-time detection of circulating apoptotic cells by *in vivo* flow cytometry. *Mol Imaging* **4**, 415-416 (2005).
4. Deuschle, F.C. et al. Development of a high affinity Anticalin^(R) directed against human CD98hc for theranostic applications. *Theranostics* **10**, 2172-2187 (2020).
5. Morath, V. et al. Molecular Design of ⁶⁸Ga- and ⁸⁹Zr-Labeled Anticalin Radioligands for PET-Imaging of PSMA-Positive Tumors. *Mol Pharm* **20**, 2490-2501 (2023).
6. Zanzonico, P. et al. ¹³¹I-FIAU labeling of genetically transduced, tumor-reactive lymphocytes: cell-level dosimetry and dose-dependent toxicity. *Eur J Nucl Med Mol Imaging* **33**, 988-997 (2006).
7. Paganetti, H. A review on lymphocyte radiosensitivity and its impact on radiotherapy. *Front Oncol* **13**, 1201500 (2023).
8. Cardinale, J. et al. Preclinical Evaluation of ¹⁸F-PSMA-1007, a New Prostate-Specific Membrane Antigen Ligand for Prostate Cancer Imaging. *J Nucl Med* **58**, 425-431 (2017).
9. Xie, Q. et al. Synthesis, preclinical evaluation, and a pilot clinical imaging study of [¹⁸F]AlF-NOTA-JR11 for neuroendocrine neoplasms compared with [⁶⁸Ga]Ga-DOTA-TATE. *Eur J Nucl Med Mol Imaging* **48**, 3129-3140 (2021).
10. Ross, T.L. et al. A new ¹⁸F-labeled folic acid derivative with improved properties for the PET imaging of folate receptor-positive tumors. *J Nucl Med* **51**, 1756-1762 (2010).
11. Trachsel, B. et al. Reducing kidney uptake of radiolabelled exendin-4 using variants of the renally cleavable linker MVK. *EJNMMI Radiopharm Chem* **8**, 21 (2023).
12. Zheng, Q.H. et al. [¹¹C]Choline as a potential PET marker for imaging of breast cancer athymic mice. *Nucl Med Biol* **29**, 803-807 (2002).
13. Müller, C. et al. A unique matched quadruplet of terbium radioisotopes for PET and SPECT and for alpha- and beta- radionuclide therapy: an *in vivo* proof-of-concept study with a new receptor-targeted folate derivative. *J Nucl Med* **53**, 1951-1959 (2012).
14. Kim, H.J., Eichinger, A. & Skerra, A. High-affinity recognition of lanthanide(III) chelate complexes by a reprogrammed human lipocalin 2. *J Am Chem Soc* **131**, 3565-3576 (2009).